# OTUD6B regulates KIFC1-dependent centrosome clustering and breast cancer cell survival

Valeria E Marotta [1,2,4], Dorota Sabat-Pośpiech[1,4], Andrew B Fielding [1,3,4✉], Amy H Ponsford [2], Amanda Thomaz [3], Francesca Querques [1], Mark R Morgan [1,2], Ian A Prior [1,2] & Judy M Coulson [1,2✉]

## Abstract

**Cancer cells often display centrosome amplification, requiring the kinesin KIFC1/HSET for centrosome clustering to prevent multipolar spindles and cell death. In parallel siRNA screens of deubiquitinase enzymes, we identify OTUD6B as a positive regulator of KIFC1 expression that is required for centrosome clustering in triple-negative breast cancer (TNBC) cells. OTUD6B can localise to centrosomes and the mitotic spindle and interacts with KIFC1. In OTUD6B-deficient cells, we see increased KIFC1 polyubiquitination and premature KIFC1 degradation during mitosis. Depletion of OTUD6B increases multipolar spindles without inducing centrosome amplification. Phenotypic rescue is dependent on OTUD6B catalytic activity and evident upon KIFC1 overexpression. OTUD6B is commonly overexpressed in breast cancer, correlating with KIFC1 protein expression and worse patient survival. TNBC cells with centrosome amplification, but not normal breast epithelial cells, depend on OTUD6B to proliferate. Indeed CRISPR-Cas9 editing results in only _OTUD6B$^{-/+}$_ TNBC cells which fail to divide and die. As a deubiquitinase that supports KIFC1 expression, allowing pseudo-bipolar cell division and survival of cancer cells with centrosome amplification, OTUD6B has potential as a novel target for cancer-specific therapies.**

**Keywords** Centrosome; Kinesin; Multipolar Spindle; OTU Deubiquitinase 6B; DUB
**Subject Categories** Cancer; Cell Cycle; Post-translational Modifications & Proteolysis

## Introduction

Centrosomes are small organelles, composed of two core centrioles surrounded by a cloud of pericentriolar material, which act as the main microtubule organising centre in metazoan cells. They undergo tightly controlled duplication during S-phase, so mitotic cells contain two mature centrosomes (Nigg and Holland, 2018) that nucleate the bipolar mitotic spindle and allow precise separation of cellular DNA into daughter cells. However, supernumerary centrosomes are prevalent in cancer cell lines, and frequently observed in solid tumours and haematological malignancies, where they are associated with poor prognosis (Chan, 2011; Marteil et al, 2018; Sabat-Pospiech et al, 2022; Sabat-Pospiech et al, 2019). Centrosome amplification may arise from de novo centrosome formation, over duplication or fragmentation of centrioles, or failure of cytokinesis (Fukasawa, 2007; Marteil et al, 2018). Increased centrosome number compromises mitotic fidelity, leading to genetic instability that fosters aneuploidy (Ganem et al, 2009; Lingle et al, 2002) and aberrant stem cell division driving cancer initiation (Basto et al, 2008; Levine et al, 2017), as well as intra-tumoral heterogeneity (Jusino et al, 2018). Centrosome amplification is also associated with cancer progression and invasion (Arnandis et al, 2018; Godinho et al, 2014) and resistance to chemotherapy (Sauer et al, 2023).

Despite the oncogenic advantages, undergoing cell division with supernumerary centrosomes carries the risk of nucleating a multipolar mitotic spindle, which almost inevitably results in cell death (Brinkley, 2001; Ganem et al, 2009). Cancer cells commonly avoid such mitotic catastrophe by clustering their excess centrosomes into two functional spindle poles, allowing pseudo-bipolar cell division (Quintyne et al, 2005; Ring et al, 1982). The proteins required for centrosome clustering therefore present attractive and specific therapeutic targets for cancer cells, which are exquisitely reliant on this process that is only rarely required in specific subsets of non-cancer cells, for example, some hepatocytes (Guidotti et al, 2003). As centrosome amplification increases low-level aneuploidy (Ganem et al, 2009) and metastasis (Godinho et al, 2014), the most aggressive cancer cells may potentially be targeted by preventing centrosome clustering.

One of the proteins required for centrosome clustering is Kinesin Family Member C1 (KIFC1/HSET), the principle hit from a _Drosophila_ screen to define mechanisms that suppress multipolar mitoses (Kwon et al, 2008). KIFC1 is a minus-end directed kinesin 14 family member that can crosslink neighbouring microtubules (Cai et al, 2009), focus microtubule minus ends at spindle poles and tether them to centrosomes (Chavali et al, 2016; She and Yang,

[1]Cellular and Molecular Physiology, Institute of Systems Molecular and Integrative Biology, University of Liverpool, Crown St, Liverpool L69 3BX, UK. [2]Molecular and Clinical Cancer Medicine, Institute of Systems Molecular and Integrative Biology, University of Liverpool, Crown St, Liverpool L69 3BX, UK. [3]Biomedical and Life Sciences, Faculty of Health and Medicine, Lancaster University, Lancaster LA1 4YG, UK. [4]These authors contributed equally: Valeria E Marotta, Dorota Sabat-Pośpiech, Andrew B Fielding.
✉E-mail: a.fielding1@lancaster.ac.uk; jcoulson@liverpool.ac.uk

2017). It is essential, although not sufficient, to cluster centrosomes (Chavali et al, 2016; Kleylein-Sohn et al, 2012). KIFC1 is overexpressed in multiple cancers that exhibit centrosome amplification, including ovarian and breast cancer (Li et al, 2015; Mittal et al, 2016), consistent with an increased requirement to mediate centrosome clustering. Centrosome amplification is associated with higher grade tumours, including triple-negative breast cancer (TNBC) (Denu et al, 2016), an aggressive and heterogenous subtype that lacks consensus targeted molecular therapies. However, KIFC1 has been established as a cancer-specific dependency in TNBC (Patel et al, 2018). The potential of KIFC1 as a tractable therapeutic target has underpinned the development of more than a dozen inhibitor compounds (Sharma et al, 2023). In common with genetic KIFC1 depletion, these compounds can induce multipolar spindles and mitotic death in cells with centrosome amplification (Kwon et al, 2008; Watts et al, 2013; Wu et al, 2013; Zhang et al, 2016). However, their pharmacological specificity for cancer over normal cells remains limited (Patel et al, 2018; Wu et al, 2013; Yukawa et al, 2018) and next-generation KIFC1 inhibitors or novel ways to target centrosome-clustering are needed.

Modulation of KIFC1 stability, and hence availability, may present an alternative strategy to reduce its activity in cancer cells. KIFC1 is known to be polyubiquitinated by the APC/C at mitotic exit, targeting it for proteasomal degradation (Min et al, 2014; Singh et al, 2014). A family of around 100 human deubiquitinases (DUBs) remove ubiquitin from proteins to reverse effects on protein stability, localisation, or activity, displaying both substrate and ubiquitin chain type specificity (Clague et al, 2013; Clague et al, 2019). DUBs are increasingly implicated in signalling pathways associated with hallmarks of cancer and are being fast-tracked as new pharmaceutical targets (Clague et al, 2012; Dewson et al, 2023). Specific inhibitors are already available for DUBs including USP7 (Turnbull et al, 2017) and, most notably USP1, with KSQ-4279 in phase 1 clinical trials (https://www.clinicaltrials.gov/study/NCT05240898). Although reversible ubiquitylation contributes to precise regulation of cell cycle progression and centrosome replication (Darling et al, 2017), no roles for DUBs have yet been delineated in centrosome clustering, or in counteracting KIFC1 ubiquitylation.

To identify DUBs required for KIFC1-dependent centrosome clustering, we performed two parallel siRNA screens in the TNBC cell line BT549, using the frequency of multipolar mitoses and KIFC1 protein level as readouts. The top DUB candidate to emerge from these screens was OTUD6B, which is amplified and associated with poor prognosis in breast cancer. We show that OTUD6B co-localises with KIFC1 at mitosis and constrains mitotic KIFC1 ubiquitylation. OTUD6B catalytic activity is required to maintain KIFC1 levels and limit multipolar spindle formation, promoting the survival of cancer cells with supernumerary centrosomes. As such, OTUD6B is an attractive new cancer-specific therapeutic target, especially in hard-to-treat cancers such as TNBC.

# Results

## DUB siRNA library screens link OTUD6B and JOSD2 to KIFC1 protein expression and centrosome clustering

We selected BT549, a triple-negative breast ductal carcinoma cell line with a high degree of centrosome amplification (Chavali et al,

2016; Kleylein-Sohn et al, 2012; Kwon et al, 2008), to screen for the effect of DUBs on multipolar spindle formation and KIFC1 expression. BT549 cells cluster their supernumerary centrosomes to survive cell division, but typically exhibit multipolar spindles in 15–20% of the mitotic cell population (Appendix Fig. S1A,B). To identify DUBs that promote centrosome clustering, we used an siRNA library consisting of a pool of four oligonucleotides targeting each of 94 DUBs and assessed de-clustering. Mitotic phenotypes were classified (Fig. 1A) and the prevalence of multipolar spindles scored following depletion of each DUB (Fig. 1B; Appendix Table S1). For the top hits in the screen, a replicate experiment using the siRNA pools confirmed a marked increase in multipolar spindle frequency upon depletion of USP10, OTUD6B, JOSD2, USP37, or USP50 (Fig. 1C).

A second siRNA screen was also performed in BT549 cells to identify DUBs that regulate KIFC1. Depletion of seven DUBs decreased KIFC1 expression by more than twofold, as determined by immunoblotting, highlighting candidates that might stabilise this kinesin (Fig. 1D; Appendix Fig. S1C and Appendix Table S1). As initial validation, transfection with these siRNA pools was repeated once more, confirming that depletion of USP3, OTUD6B, JOSD2, or UCHL1 decreased KIFC1 protein levels (Fig. 1E). Combining data from the two siRNA screens highlighted DUBs whose depletion potentially caused KIFC1-dependent centrosome de-clustering, most notably OTUD6B and JOSD2 (Fig. 1F).

To assess whether the effects we observed were likely to be on-target, deconvolution experiments were performed using the four individual siRNAs targeting the lead hits in each screen. Several initial hits from the de-clustering screen did not pass validation, for example only one USP10 siRNA had a similar effect to the pooled siRNA (Appendix Fig. S2A) suggesting an off-target effect. However, in the case of both OTUD6B and JOSD2, two independent siRNAs significantly induced a multipolar phenotype (Fig. 2A,B). In the KIFC1 screen most candidates were also eliminated through deconvolution experiments. For example, all four USP10 or USP3 siRNAs effectively depleted their target DUB, but only one siRNA reduced KIFC1 levels (Appendix Fig. S2B,C). By contrast, for both OTUD6B and JOSD2, two siRNAs significantly decreased KIFC1 protein levels to a similar extent as the siRNA pool (Fig. 2C,D). Overall, OTUD6B emerged as the lead candidate DUB that may regulate KIFC1-dependent centrosome clustering, as the same two OTUD6B siRNAs markedly induced both phenotypes (Fig. 2A,C,E). In contrast, whilst siJOSD2-3 significantly altered both readouts, siJOSD2-2 induced multipolar spindles (Fig. 2B), whereas siJOSD2-4 reduced KIFC1 expression (Fig. 2D). Interestingly, KIFC1 depletion induces OTUD6B but not JOSD2 expression, suggestive of a potential feedback mechanism where cells with low KIFC1 may upregulate OTUD6B to attempt to restore KIFC1 levels by protein stabilisation (Fig. 2C).

OTU deubiquitylase 6B (OTUD6B, DUBA5, Ensembl: ENSG00000155100, UniProtKB identifier: Q8N6M0) is a highly conserved and ubiquitously expressed cysteine protease that is a member of the OTU domain superfamily of DUBs (Clague et al, 2012; Clague et al, 2019). It has a C-terminal catalytic domain and three coiled-coil domains at the N-terminus, which may be involved in protein interactions. Whilst OTUD6B remains a relatively understudied DUB, recent reports have described new substrates and provided some insights into cellular mechanisms. These highlight context-dependent roles for OTUD6B, for example

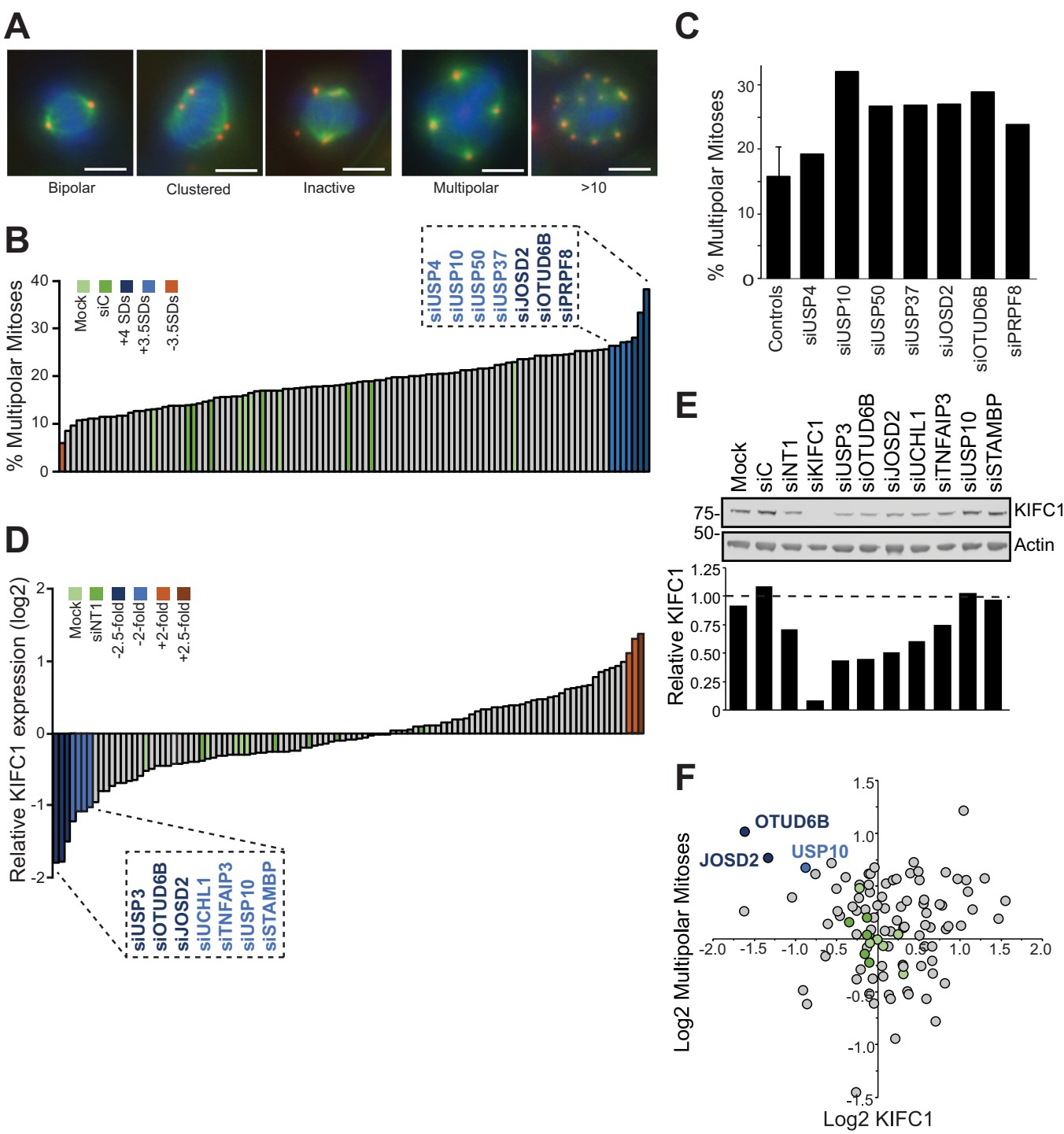

**Figure 1. DUB siRNA library screens identify potential regulators of centrosome-clustering and KIFC1 expression.**

BT549 cells were transfected with 40 nM pooled siRNA from a DUB-specific library. (A–C) Centrosome de-clustering screen. Cells were stained for pericentrin (red), alpha tubulin (green) and DNA (blue) and 36 stitched fields of view imaged. Mitotic cells were classed into phenotypes, scale bar 10 μm (A). Median of 74 mitoses scored per condition from $n = 1$ experiment, those with <30 mitotic cells excluded (USP9X, USP6 and PSMD7). Ranked percentage of multipolar mitoses for each DUB siRNA pool or control (green), standard deviation (SD) relative to control mean indicated by colour scale (B). Re-evaluation of siRNA pools for 7 candidate DUBs which increased multipolar spindle frequency by >3.5 SD in the screen, $n = 1$ experiment, error bar for controls shows SD of mean for 5 mock and 5 siC technical replicates (C). (D, E) Immunoblotting screen for KIFC1, $n = 1$ experiment. Data ranked by KIFC1 expression normalised to actin and relative to gel average (D). Re-evaluation of siRNA pools for 7 candidate DUBs that reduced KIFC1 expression by >twofold in the screen, $n = 1$ experiment (E). (F) Cross-correlation of the two screens; DUBs in top left quadrant may promote KIFC1-dependent centrosome clustering. Source data are available online for this figure.

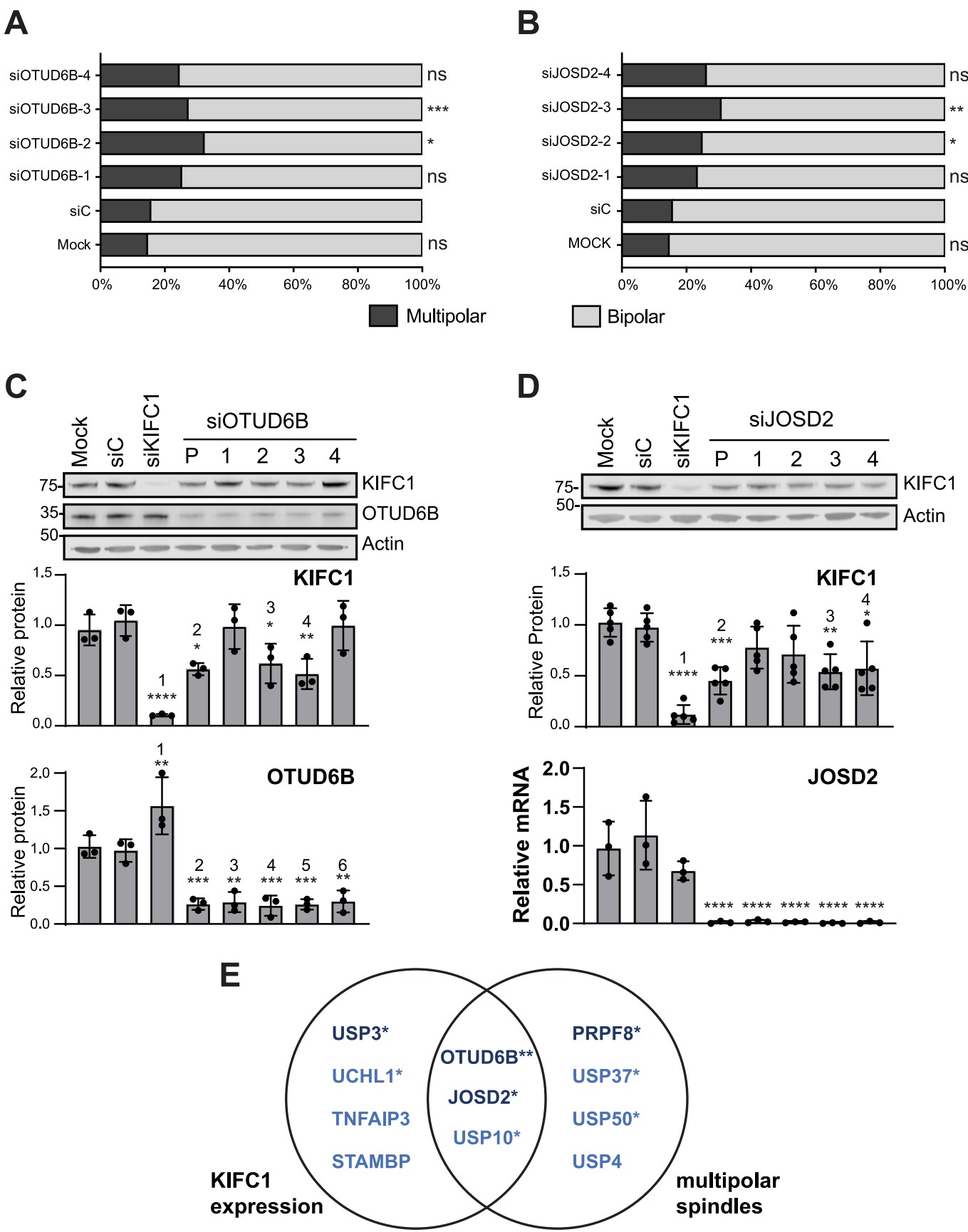

**Figure 2.  Depletion of OTUD6B or JOSD2 causes multipolar mitoses and reduces KIFC1 expression.**

Deconvolution of OTUD6B or JOSD2 siRNA pools by transfection with four individual siRNAs (40 nM). (A, B) Centrosome de-clustering was scored for >90 metaphase cells per condition across $n = 3$ biological replicates for OTUD6B (A) or JOSD2 (B) depletion; (A) ***$P = 0.0003$, *$P = 0.030$, (B) *$P = 0.026$, **$P = 0.003$ compared to siC control by one-tailed Chi-square test. (C, D) Deconvolution of OTUD6B (C) or JOSD2 (D) siRNA pools (P) for KIFC1 expression and DUB knockdown efficiency. Representative immunoblots, plots show mean expression from $n = 3$ (C), $n = 5$ (D, KIFC1) and $n = 3$ (D, JOSD2) biological replicates normalised to actin and mean of controls (JOSD2 evaluated by qRT-PCR); error bars SD, values compared to siC by one-way ANOVA with Dunnett post-hoc test. KIFC1: [1]****$P \leq 0.0001$, [2]*$P = 0.0143$, [3]*$P = 0.0323$, [4]**$P = 0.007$; OTUD6B: [1]**$P = 0.005$, [2]***$P = 0.001$, [3]**$P = 0.0014$, [4]****$P = 0.0008$, [5]***$P = 0.0009$, [6]**$P = 0.0016$ (C) KIFC1: [1]****$P \leq 0.0001$, [2]***$P = 0.0008$, [3]**$P = 0.0057$, [4]*$P = 0.0119$; JOSD2: ****$P \leq 0.0001$ (D). (E) Venn diagram summarizing top seven hits from each screen, *reproducible effect for siRNA pool, **both readouts altered by the same two siRNAs. Source data are available online for this figure.

in immune signalling stimulation (Wang et al, 2023) and pro-oncogenic (Ma et al, 2023; Paulmann et al, 2022; Yang et al, 2024) or anti-oncogenic pathways (Li et al, 2023; Liu et al, 2020), including conflicting roles for the main isoform and a shorter isoform in cell growth (Sobol et al, 2017).

## OTUD6B but not JOSD2 is overexpressed in clinical breast cancer

As our data suggest that OTUD6B may also be required to cluster supernumerary centrosomes in cancer cells, we next examined how OTUD6B expression relates to KIFC1 expression, centrosome amplification and patient outcomes. We first examined a panel of breast derived cell lines. With the notable exception of MCF7, the percentage of cells with amplified centrosomes was higher in breast cancer than in normal mammary epithelium cell lines (Fig. EV1A) and correlated with both KIFC1 and OTUD6B levels (Fig. EV1B–D). BT549 and MDA-MB-231 TNBC cell lines, with centrosome amplification in 32% or 23% of the cell population, respectively, were selected for subsequent functional analyses. We also interrogated the Breast Invasive Carcinoma TCGA PanCancer Atlas (Berger et al, 2018; Data ref: Berger et al, 2018) and CPTAC Breast Cancer (Mertins et al, 2016; Data Ref: Mertins et al, 2016) datasets. Strikingly, 31% of patients display genetic alterations in OTUD6B (Appendix Fig. S3A). Gain of OTUD6B copy number closely correlated with mRNA overexpression, which was evident in 28% of cases (Appendix Fig. S3B,C) and was associated with worse overall survival (Fig. EV1E).

Disaggregating the TCGA data into the major breast cancer subtypes showed that both KIFC1 and OTUD6B were more highly expressed in luminal B cases, but most significantly overexpressed in the TNBC basal-like subtype (Fig. EV1F,G). Correlation between the TCGA and CPTAC mass spectrometry data for invasive breast carcinomas confirmed that OTUD6B amplification was associated with increased protein levels (Appendix Fig. S3D). Importantly, where data were available for both proteins, OTUD6B and KIFC1 levels were positively correlated (Fig. EV1H), reflecting our observations from cell lines. Taken together, these data suggest a potential oncogenic role for OTUD6B and support the idea that OTUD6B maintains KIFC1 protein levels in cancer cells. In contrast, our second hit from the screens, JOSD2, was only altered in 5% of breast cancer cases and was not associated with survival or with KIFC1 protein levels (Appendix Fig. S4), so was not investigated further.

## OTUD6B modulates KIFC1 level and centrosome clustering in panel of breast cell lines

In subsequent experiments using lower concentrations of siRNA in BT549 cells, we found that the effect on KIFC1 correlated with

knockdown efficiency for the four OTUD6B siRNAs from the library (Fig. EV2A–D). In addition, a fifth siRNA sequence, specific for the transcript variant encoding the major OTUD6B isoform-1 (NM_016023.5), also substantially reduced KIFC1 expression in BT549 (Fig. EV2C). Importantly, OTUD6B depletion could reduce KIFC1 expression in other cancer cell lines: MDA-MB-231, another TNBC cell line with moderate centrosome amplification, U2OS osteosarcoma cells that were originally used to demonstrate KIFC1 ubiquitination by the APC/C (Min et al, 2014), as well as in hTERT-HME1 as a model of immortalised normal breast epithelium (Fig. EV2E–G). Multiple OTUD6B siRNAs also increased the frequency of multipolar spindles in the MDA-MB-231 and BT549 TNBC cell lines (Fig. EV2H–J).

## OTUD6B depletion mimics the KIFC1-deficient centrosome phenotype

We next sought to investigate whether the requirement for OTUD6B to limit multipolar spindle formation was linked to centrosome clustering, rather than to the maintenance of centrosome number or to centriole stability. Importantly, depletion of OTUD6B with multiple siRNAs did not induce centrosome amplification in the immortalised normal breast epithelium cell line hTERT-HME1 (Fig. EV3A) or in MDA-MB-231 TNBC cells (Fig. EV3B). To investigate the centrioles within amplified centrosomes that form multipolar spindles, we compared centrin staining in cells treated with OTUD6B or KIFC1 siRNAs. In around two thirds of KIFC1-depleted BT549 cells, all pericentrin foci stained for a pair of centrioles (Fig. EV3C), consistent with previous reports (Chavali et al, 2016). Depletion of OTUD6B with either of two siRNAs also resulted in a similar proportion of cells exhibiting this normal centriole phenotype (Fig. EV3C). We also found that OTUD6B depletion did not change centriole or centrosome number in interphase BT549 cells, nor induce centriole splitting, as two centrioles per pericentrin foci were observed in cells with both normal and amplified numbers of centrosomes (Fig. EV3D,E). Together these data suggest that multipolar spindles are predominantly formed in OTUD6B-deficient cells by centrosome de-clustering, rather than through centriole fragmentation or formation of acentrosomal poles.

## OTUD6B catalytic activity is required to rescue KIFC1 levels and bipolar spindles

We next examined the effect of OTUD6B depletion on KIFC1 levels in individual cells. Consistent with immunoblotting data (Fig. 2C),

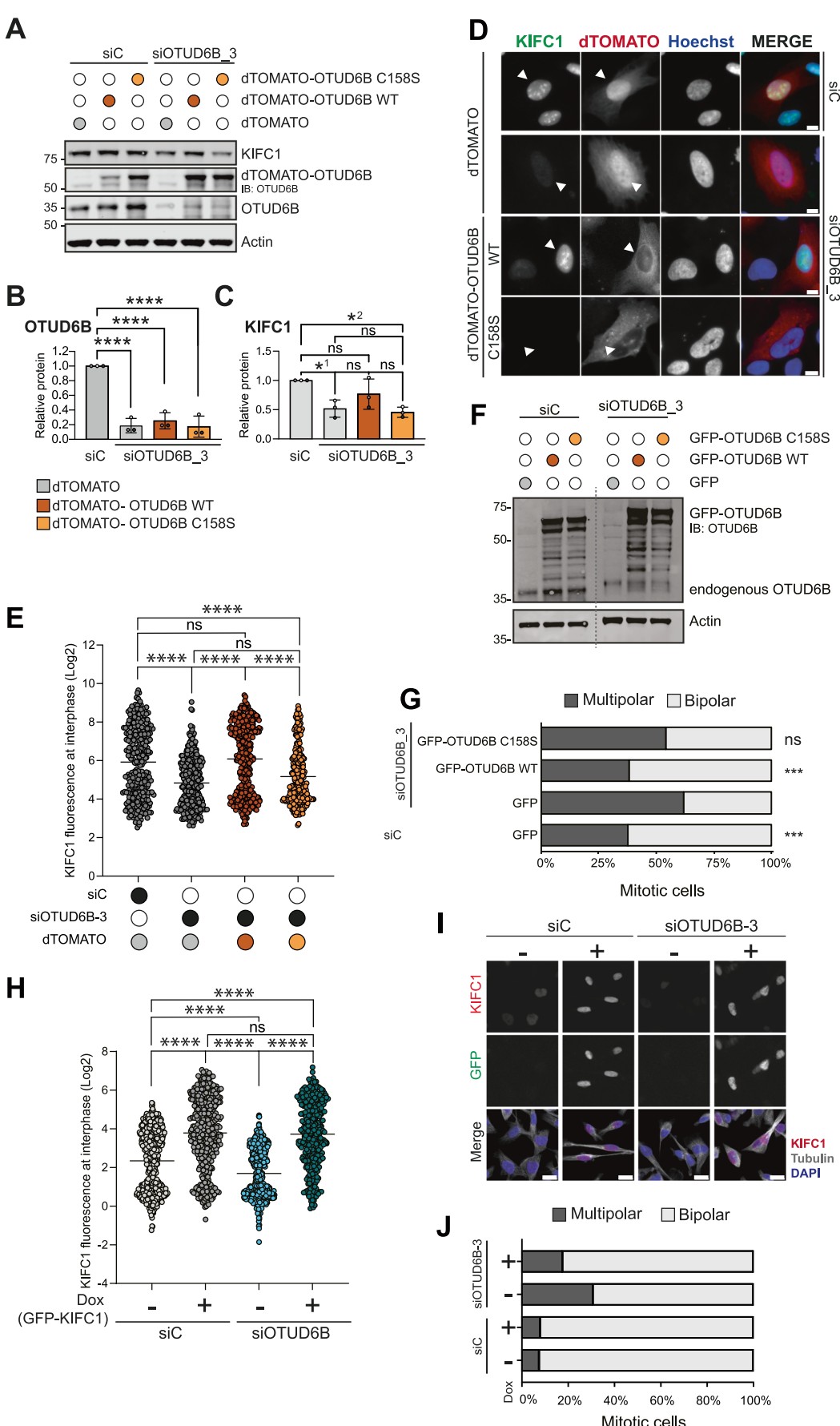

◄ **Figure 3. OTUD6B catalytic activity is required to rescue KIFC1 levels and prevent multipolar spindle formation.**

(A–E) Overexpression of dTOMATO-OTUD6B but not catalytic dead dTOMATO-OTUD6B-C158S rescues KIFC1 levels in OTUD6B-depleted cells. U2OS cells were transfected with plasmids (48 h) and siRNAs (10 nM, 72 h) then immunoblotted (A). Plots show mean expression normalised to actin from $n = 3$ biological replicates for endogenous OTUD6B (B) and KIFC1 (C), error bars SD. ****$P \leq 0.0001$ (B), [1]*$P = 0.0214$, [2]*$P = 0.0112$, ns, not significant (C), compared to siC by one-way ANOVA with Tukey's multiple comparison test. In parallel, samples were co-stained for KIFC1 and DNA (Hoechst), representative images acquired with Nikon Eclipse Ti fluorescent microscope with, scale bar 10 µm. White arrows indicate examples of transfected cells, which were used for quantification (D). Mean pixel intensity scored for KIFC1 after background subtraction was measured for >250 cells per condition from $n = 3$ biological replicates; ns, not significant; ****$P \leq 0.0001$ by Kruskal–Wallis with Dunn's multiple comparison test (E). (F, G) GFP-OTUD6B but not catalytic-dead GFP-OTUD6B-C158S, rescues centrosome clustering. BT549 cells were transfected with 10 nM siRNAs for 72 h and plasmids for the final 24 h. Immunoblot shows expression of siOTUD6B_3 resistant GFP-OTUD6B, dotted line indicates break in membrane, $n = 1$ experiment (F). Cells were stained for tubulin and DAPI (DNA) and visualised using a Nikon Eclipse Ti fluorescent microscope. The percentage of bipolar and multipolar spindles were scored in 100 GFP-positive mitotic cells per condition from $n = 1$ experiment, ***$P = 0.0005$ by one-tailed Chi-square test (G). (H–J) KIFC1 overexpression can rescue the siOTUD6B_3 induced multipolar spindle phenotype. MDA-MB-231 cells, either non-induced (−Dox) or induced (+Dox) to express GFP-KIFC1siRes, were transfected with 10 nM siRNAs and images captured using a Zeiss LSM880 confocal microscope. Cells were stained for KIFC1, α-tubulin, and DNA (DAPI) 72 h post-transfection, and KIFC1 fluorescence intensity quantified for >330 cells per condition from $n = 1$ experiment; ****$P \leq 0.0001$ by Kruskal–Wallis with Dunn's multiple comparison test (H) representative images, scale bar 10 µm (I). Cells were stained for pericentrin, α-tubulin, and DNA (DAPI) 48 h post-transfection, and the percentage of multipolar spindles calculated from all visible mitoses across multiple fields of view; $n > 40$ cells per condition from $n = 1$ experiment (J). Source data are available online for this figure.

OTUD6B depletion significantly reduced the mean KIFC1 immunofluorescence signal in individual interphase cells (Appendix Fig. S5A–C). To explore whether restoring OTUD6B could rescue this phenotype, we generated dTOMATO-tagged OTUD6B that was resistant to the siRNA siOTUD6B-3. Transient overexpression of wild type (WT) or catalytic dead (C158S) dTOMATO-OTUD6B was seen on a background of endogenous OTUD6B depletion (Fig. 3A,B). Despite the immunoblot representing a heterogenous population of cells, some with and some without overexpression of dTOMATO-OTUD6B, WT but not C158S partially rescued endogenous KIFC1 expression (Fig. 3A,C). To provide a more accurate readout of this rescue, KIFC1 levels were quantified at a single cell level in individual dTOMATO-transfected cells (Fig. 3D); WT dTOMATO-OTUD6B significantly rescued KIFC1 levels, but the C158S form did not (Fig. 3E).

Similarly, although the baseline level of multipolar spindles observed was higher in the control transfected cells in this experiment than in untransfected cells (Fig. EV2J), overexpression of siOTUD6B-3 resistant WT GFP-OTUD6B significantly increased the number of bipolar spindles in BT549 cells, whilst the catalytic dead C158S mutant could not rescue this phenotype (Fig. 3F,G). Thus, the deubiquitinase activity of OTUD6B is required to retain cellular KIFC1 levels and avoid multipolar spindle formation. We next asked whether this OTUD6B depletion phenotype was dependent on its effects on KIFC1 levels, and therefore could be rescued by overexpressing KIFC1. To address this, we used an MDA-MB-231 cell line where KIFC1 could be induced by doxycycline and examined individual cells by microscopy. We confirmed that OTUD6B depletion reduced KIFC1 in the population of uninduced cells, but that this was overcome by acute induction of KIFC1 with doxycycline (Fig. 3H,I). Under these conditions of OTUD6B knockdown and KIFC1 overexpression, we also scored mitotic cells for the presence of multipolar spindles. As in our previous experiments, OTUD6B depletion induced multipolar spindle formation, and this could be partially rescued by induction of KIFC1 (Fig. 3J). Overall, these data suggest that cancer cells with centrosome amplification depend upon the catalytic activity of OTUD6B to safely execute cell division, at least in part through the role of OTUD6B in protecting KIFC1, which facilitates centrosome clustering.

## OTUD6B and KIFC1 co-localise during mitosis, and can interact in cells

KIFC1 is nuclear in interphase TNBC cells and localises to the mitotic spindle at prometaphase, remaining enriched at the minus end of microtubules around centrosomes during mitosis (Appendix Fig. S5D). OTUD6B depletion did not alter this localisation, as residual KIFC1 was still evident at the mitotic spindle during metaphase (Appendix Fig. S5E) following knockdown of OTUD6B with three siRNAs that efficiently reduce KIFC1 levels in BT549 cells (Fig. EV2A–D). Importantly though, the amount of KIFC1 present at the mitotic spindle was reduced in OTUD6B-depleted cells (Appendix Fig. S5F,G). A systematic survey of exogenously expressed DUBs in HeLa cells assigned OTUD6B as a cytosolic DUB (Urbé et al, 2012). However, evaluation of GFP-OTUD6B expression in interphase U2OS cells showed that, whilst distributed throughout the cytoplasm, OTUD6B also tightly co-localised with pericentrin (Fig. 4A,C). Notably, following nuclear membrane breakdown at prometaphase, GFP-OTUD6B remained co-localised with the pericentriolar matrix (Fig. 4A,D) and was also evident on the nascent spindle adjacent to centrosomes, co-localising with tubulin (Fig. 4A) and KIFC1 (Fig. 4B,E). This co-localisation at mitosis was even more striking when GFP-KIFC1 and dTOMATO-OTUD6B WT were exogenously co-transfected in U2OS cells (Appendix Fig. S6A). Similar localisation of OTUD6B at the nascent mitotic spindle was evident in TNBC cell line BT549 (Appendix Fig. S6B). Therefore, OTUD6B is appropriately positioned to temporally interact with KIFC1 and modulate centrosome clustering.

To determine whether the two proteins associate within a cellular environment, we expressed GFP-KIFC1 in U2OS cells and tested for interaction with OTUD6B. We found that endogenous OTUD6B co-immunoprecipitated with GFP-KIFC1, but not with GFP alone (Fig. 4F). In reverse co-immunoprecipitation experiments, GFP-OTUD6B interacted with endogenous KIFC1 (Fig. 4G), further confirming cellular association of these two proteins. On a background of endogenous OTUD6B expression in asynchronous cells, we saw no increase in endogenous KIFC1 levels upon exogenous OTUD6B overexpression, suggesting there is already sufficient OTUD6B to sustain KIFC1 levels under these conditions.

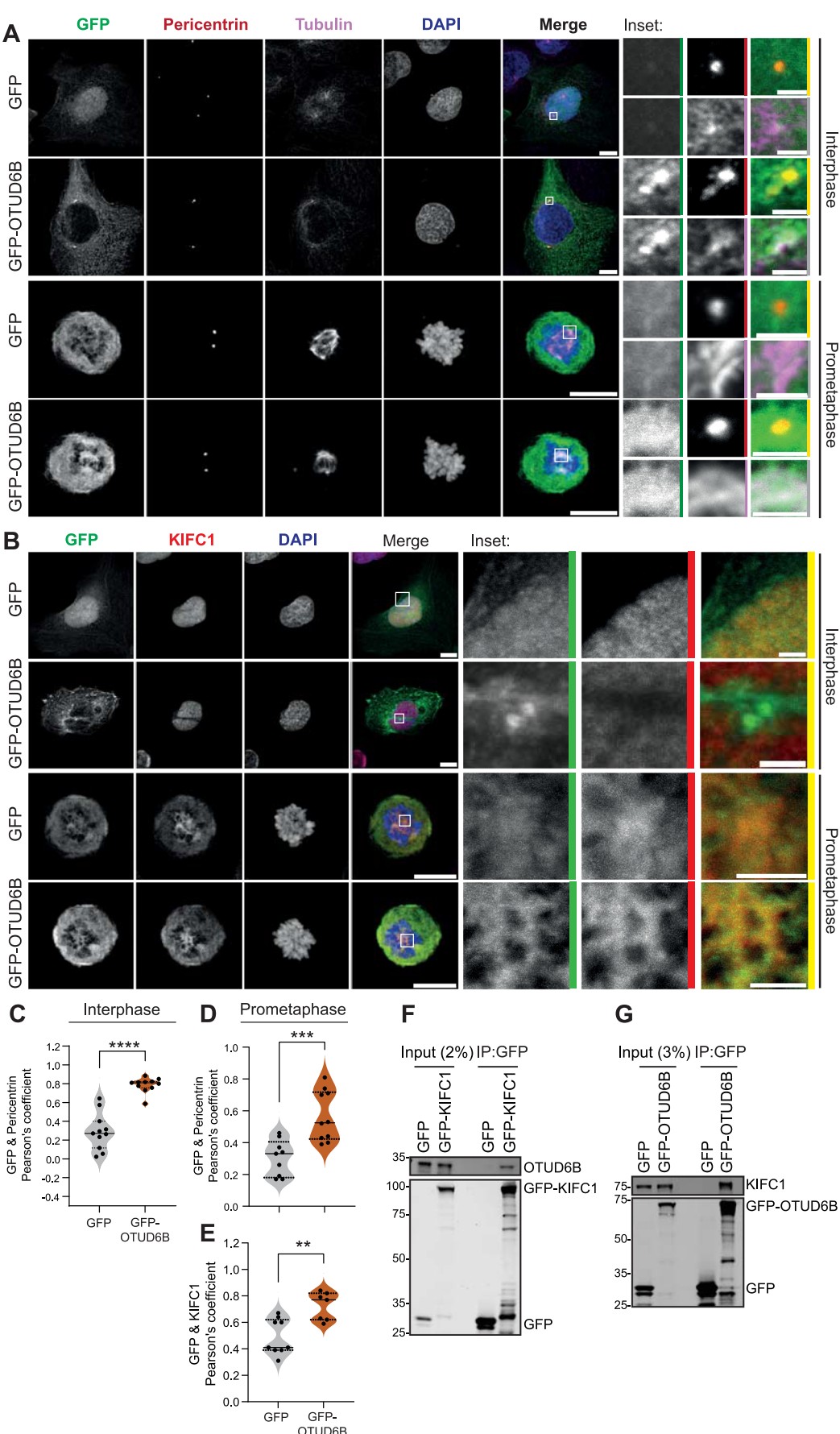

**Figure 4. OTUD6B localises to centrosomes and interacts with KIFC1.**

(A, B) OUTD6B co-localises with pericentrin in interphase cells and prometaphase cells (A) and with tubulin (A) and KIFC1 at prometaphase (B). U2OS cells transfected with GFP or GFP-OTUD6B for 48 h before staining, representative maximum-intensity projections imaged with Zeiss LSM800 confocal from $n = 2$ biological replicates; scale bar 10 μm, or 2 μm on inset. (C–E) Pearson co-localisation coefficients calculated with JACoP for GFP and GFP-OTUD6B with pericentrin at interphase (C) or prometaphase (D) and KIFC1 at prometaphase (E). Each data point represents a centrosome for pericentrin staining ($n ≥ 10$), or a cell for KIFC1 staining ($n ≥ 7$); ****$P ≤ 0.0001$ (C), ***$P = 0.0007$ (D), **$P = 0.0154$ (E) by unpaired $t$ test. (F, G) KIFC1 and OTUD6B can interact in cells. U2OS cells were transfected with GFP-KIFC1 (F), or GFP-OTUD6B (G) for 24 h and pulled-down with GFP-nanobeads to co-immunoprecipitate endogenous proteins; immunoblots representative of $n = 3$ biological replicates. Source data are available online for this figure.

To map the KIFC1 interaction domain, and explore whether interaction was dependent on catalytic competency, co-immunoprecipitation experiments were performed using OTUD6B mutants (Fig. 5A). The GFP-OTUD6B-C158S mutant, confirmed to be catalytic dead as it could no longer bind to the ubiquitin activity probe HA-Ub-PA (Fig. 5B), retained the ability to pulldown KIFC1 (Fig. 5C). Thus, interaction with KIFC1 is independent of OTUD6B catalytic activity. Using truncation mutants, we found that the C-terminal catalytic OTU domain was dispensable, whilst the N-terminal domain was required for interaction with KIFC1 (Fig. 5C).

Interestingly, whilst centrosome localisation of OTUD6B in interphase cells was independent of catalytic activity, the mutants lacking the N-terminal domain did not co-localise with pericentrin and showed nuclear accumulation (Fig. 5D,E). To investigate further, we scored cells transfected with each construct to determine whether OTUD6B was predominantly cytoplasmic or enriched within nuclei (Fig. EV4A,B). Notably, the N-terminal deleted OTUD6B constructs localised to the nucleus in 100% of cells, whereas nuclear enrichment was infrequent and less prominent for C-terminal deleted or full-length OTUD6B, albeit occurring at a slightly higher rate for catalytic dead full-length OTUD6B (Fig. EV4B). Together these data suggest that the N-terminus of OTUD6B is critical not only for interaction with KIFC1, but also for exclusion from the nucleus where KIFC1 is located during interphase. We therefore reasoned that interaction of these two proteins is likely most relevant when they co-locate to the spindle during mitosis.

## OTUD6B regulates KIFC1 deubiquitylation and stability during mitosis

These co-localisation data suggest a direct role of OTUD6B in stabilising KIFC1 and we ruled out several alternative mechanisms, as OTUD6B depletion did not alter KIFC1 subcellular localisation (Appendix Fig. S5E) or transcript levels (Fig. EV5A) in TNBC cells. Focusing our attention on KIFC1 stability, we examined this in both BT549 TNBC cells with high centrosome amplification and in U2OS cells, the cell line in which KIFC1 ubiquitination was originally demonstrated (Min et al, 2014). We found that proteasome inhibition with epoxomicin significantly increased KIFC1 levels in asynchronous U2OS cells, but not in BT549 cells (Fig. EV5B,C). This suggests that KIFC1 may be more slowly turned over in BT549 cells. Inhibition of translation with cycloheximide confirmed that this was the case (Fig. EV5D–F). However, we were unable to show any effect of OTUD6B depletion on KIFC1 stability (Fig. EV5G,H) or on KIFC1 ubiquitination in

asynchronous U2OS cells (Fig. EV5I,J). Finally, as OTUD6B (Paulmann et al, 2022) and KIFC1 (Wei and Yang, 2019), have been implicated in S-phase entry and duration, respectively, we examined cell cycle phase profiles in MDA-MB-231 cells with centrosome amplification. Whilst we confirmed that KIFC1 depletion significantly increased the proportion of cells in S-phase relative to G0/G1, we saw no effect of OTUD6B depletion (Fig. EV5K). Taken together these data argue against indirect effects of OTUD6B on KIFC1 levels and focused our attention on a direct role for OTUD6B during mitosis.

Given evidence that KIFC1 levels peak at mitosis prior to rapid degradation at mitotic exit following ubiquitination by the APC/C (Min et al, 2014; Singh et al, 2014), we reasoned that OTUD6B may stabilise KIFC1 when it is required to cluster amplified centrosomes during mitosis. To better understand the timing and mechanism of OTUD6B activity, we replicated the synchronisation method used by Min et al, 2014, arresting cells at prometaphase with the Eg5 (KIF11) inhibitor STLC, before releasing with the Aurora B inhibitor ZM447439 to monitor pseudo-mitotic exit. As expected, KIFC1 expression levels peaked at prometaphase before KIFC1 protein levels rapidly decreased between 30 to 60 min after release in U2OS (Fig. 6A) and in TNBC cells (Appendix Fig. S7A–C). Critically, OTUD6B depletion blunted this dynamic KIFC1 regulation throughout mitosis (Fig. 6A). We also confirmed that substantial polyubiquitylation of GFP-KIFC1 was induced during pseudo-mitotic exit (Appendix Fig. S7D). Importantly, although we were unable to demonstrate an effect of OTUD6B on KIFC1 ubiquitination in asynchronous cells (Fig. EV5I,J), the level of KIFC1 polyubiquitylation increased in OTUD6B-depleted cells arrested at prometaphase and this persisted 30 to 45 min into pseudo-mitotic exit in the absence of proteasome inhibition (Fig. 6B; Appendix Fig. S7E). This observation is consistent with OTUD6B promoting KIFC1 centrosome clustering activity early in mitosis to form a pseudo-bipolar spindle, along with an ongoing requirement to maintain KIFC1 levels and prevent de-clustering during anaphase. In the converse experiment, we found that overexpression of wild-type, but not catalytic dead, OTUD6B could decrease polyubiquitination of GFP-KIFC1 during mitotic exit (Fig. 6C). These data are consistent with OTUD6B acting to constrain ubiquitylation of KIFC1 during mitosis, and given that KIFC1 had been identified as an APC/C substrate (Min et al, 2014; Singh et al, 2014), we investigated further.

The APC/C appends chains containing K11-linked ubiquitin to substrates and consequently ubiquitin chains containing K11 are highly upregulated during mitosis (Matsumoto et al, 2010; Meyer and Rape, 2014). Therefore, we next examined GFP-KIFC1 polyubiquitination using K11-only and K63-only ubiquitin

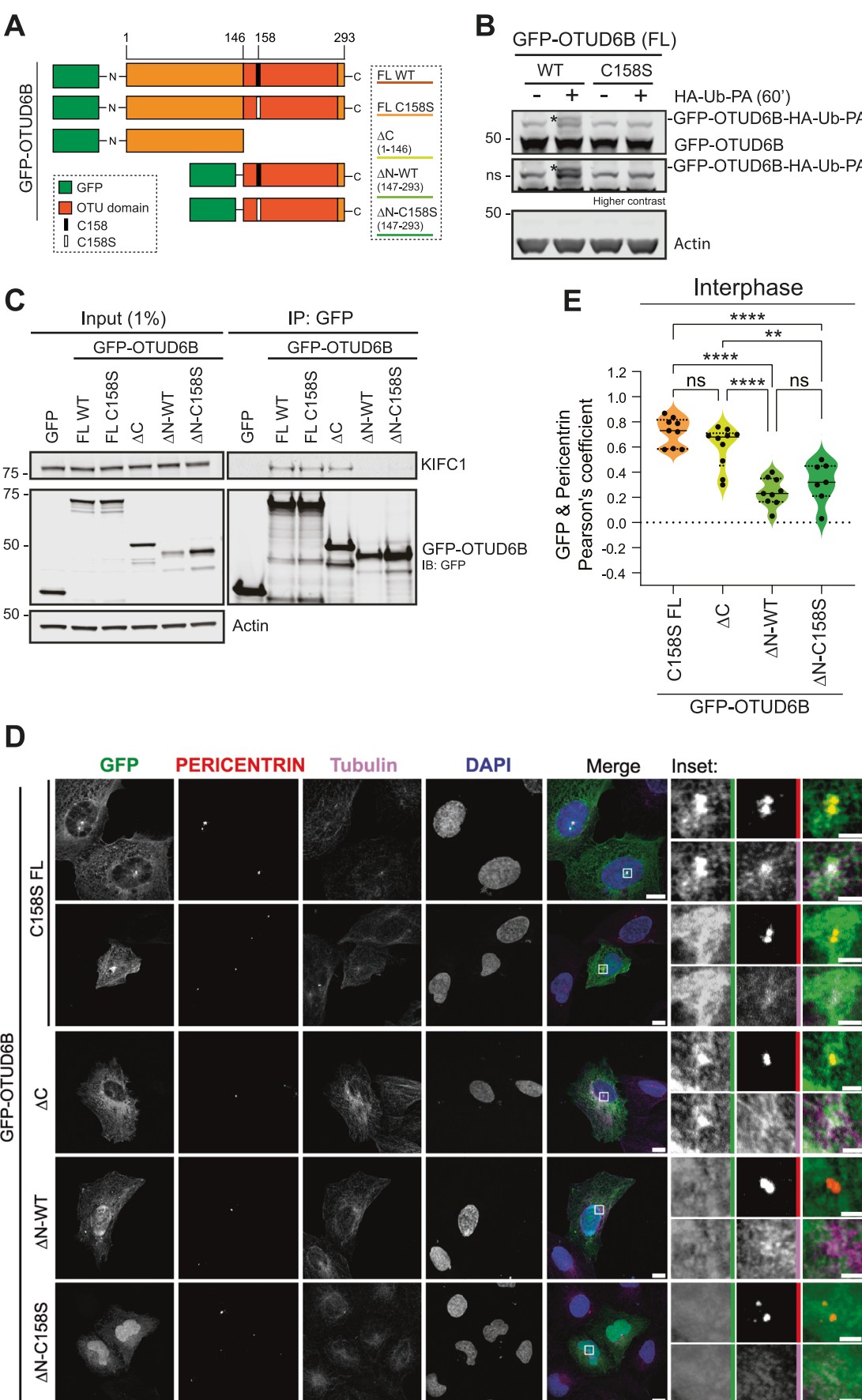

**Figure 5. The N-terminus of OTUD6B is required for centrosome localisation and interaction with KIFC1.**

(**A**) Schematic of GFP-OTUD6B mutants. (**B**) OTUD6B demonstrates catalytic activity in cells, which is abrogated by mutation of cysteine 158. U2OS cells were transfected with plasmids for 48 h, lysed and incubated with Ub-PA for 60 min then immunoblotted for OTUD6B, $n = 1$ experiment, *denotes Ub-PA conjugation causing GFP-OTUD6B gel shift. (**C**) KIFC1 interacts with the N-terminus of OTUD6B. U2OS were transfected with plasmids for 24 h and GFP-OTUD6B pulled-down with GFP-nanobeads to co-immunoprecipitate endogenous KIFC1; Immunoblot representative of $n = 2$ biological replicates. (**D, E**) Localisation of GFP-OTUD6B mutants in U2OS cells co-stained for centrosome (pericentrin), spindle (tubulin), and DNA (DAPI) markers. Representative images from $n = 1$ experiment presented as maximum-intensity projections imaged with Zeiss LSM800 confocal; scale bar 10 μm, or 2 μm on inset. FL, full length (**D**). Pearson co-localisation coefficients calculated with JACoP for GFP-OTUD6B with pericentrin at interphase for >7 cells from $n = 1$ experiment; **$P = 0.0018$, ****$P \leq 0.0001$, ns not significant by one-way ANOVA with Tukey's multiple comparison test (**E**). Source data are available online for this figure.

mutants. GFP-KIFC1 was heavily polyubiquitinated with K11 but not K63 chains during mitotic exit, and interestingly endogenous OTUD6B was co-immunoprecipitated with GFP-KIFC1 from mitotic cells when wild-type or K11-only, but not K63-only, ubiquitin was overexpressed (Fig. 6D). Importantly, the poly-ubiquitination of GFP-KIFC1 in the presence of K11-only ubiquitin was robustly increased by OTUD6B depletion (Fig. 6E). We therefore concluded that OTUD6B likely regulates KIFC1 through antagonising its ubiquitylation by the APC/C during mitosis. Although this effect on KIFC1 ubiquitination begins in prometaphase ahead of the spindle assembly checkpoint, it continues into anaphase, and this timing could be consistent with antagonism of the APC/C, which has several early substrates such as cyclin A and Nek2 (Di Fiore and Pines, 2010; Meyer and Rape, 2011, 2014).

Given the evidence for OTUD6B regulation of mitotic KIFC1 ubiquitination, we used live cell imaging of Venus-tagged KIFC1 (Min et al, 2014) to assess how KIFC1 degradation progressed in OTUD6B-depleted cells during mitosis. Whilst wild-type KIFC1-Venus degraded over 120 min after anaphase onset, the KIFC1-Venus-Dmut with mutation (R5A, L8A) of the N-terminal D-box recognised by the APC/C did not degrade (Appendix Fig. S7F,G). Furthermore, stability of KIFC1-Venus-Dmut was unaffected by OTUD6B depletion (Fig. 6F,G). In contrast, degradation of wild type KIFC1-Venus began around anaphase onset in control cells, but in OTUD6B-depleted cells degradation was evident prior to anaphase onset (Fig. 6F). OTUD6B depletion also resulted in significantly lower levels of KIFC1 at anaphase onset (Fig. 6G), which persisted until 16–24 min into anaphase (Fig. 6F), however these data suggest that OTUD6B is less able to oppose the APC/C once the spindle assembly checkpoint is satisfied. Thus, OTUD6B may counteract the APC/C early in mitosis, so that in the absence of OTUD6B this may position KIFC1 as a relatively early APC/C substrate, more akin to cyclin A and Nek2 (Meyer and Rape, 2011), which can be degraded prior to satisfying the spindle assembly checkpoint. These degradation data are consistent with the ubiquitination results (Fig. 6B) and suggest that OTUD6B could help maintain KIFC1 stability between prophase and anaphase, as centrosomes need to become, and remain, clustered to maintain a pseudo-bipolar spindle.

## OTUD6B represents a potential cancer-specific target

As OTUD6B may represent a legitimate target to promote cancer-specific cell death by inducing multipolar spindles, we tested the requirement of OTUD6B for viability of centrosome-amplified

TNBC cells. Depletion of either KIFC1 or OTUD6B significantly decreased growth of BT549 cells based on estimation of metabolically viable cells (Fig. 7A). We obtained similar if more modest results using live cell imaging to monitor growth of MDA-MB-231 cells that have more moderate centrosome amplification (Fig. 7B; Appendix Fig. S8A,B). Hence, two TNBC cell lines with heterogenous genetic backgrounds and differing degrees of centrosome amplification, both rely on centrosome clustering to proliferate. We then compared the effect of KIFC1 and OTUD6B depletion on colony formation over a longer time course. The immortalised breast epithelial cell line hTERT-HME1 was used as a normal tissue comparator that has low centrosome amplification (Fig. EV1A) but, as in TNBC, OTUD6B depletion leads to reduced KIFC1 levels (Fig. EV2G). Whilst depletion of either KIFC1, or OTUD6B by two independent siRNAs, significantly decreased colony formation in BT549 cells that have high centrosome amplification, colony number was unaffected in hTert-HME1 cells (Fig. 7C–F). Likewise, in the MDA-MB-231 TNBC cell line which has an intermediate level of centrosome amplification (Fig. EV1A), OTUD6B depletion also reduced colony formation (Fig. 7G,H). Thus, the effect of OTUD6B deficiency on cell survival was restricted to the TNBC cells, consistent with the idea that it is a cancer-specific target.

Based on these results, we predicted that knockout of *OTUD6B* using CRISPR/Cas9 may prove lethal to TNBC cells with amplified centrosomes. Indeed, in we were only able to generate a few heterozygous *OTUD6B*[-/+] BT549 clones (Fig. 7I) and no homo-zygous knockouts. In contrast, using a workhorse cell line, we could generate homozygous *OTUD6B*[-/-] HEK-293T clones using the same guide pair (Appendix Fig. S8C). HEK-293T lack centrosome amplification (Bose et al, 2021) and the *OTUD6B*[-/-] clones demonstrated no growth defects (Appendix Fig. S8D; Movies EV1 and EV2), despite lacking OTUD6B and having reduced KIFC1 expression (Appendix Fig. S8E) that further confirmed a requirement for OTUD6B to maintain KIFC1 levels. The BT549 *OTUD6B*[-/+] clones remained extremely sparse over 4 weeks (Fig. 7J) with too few cells to quantify protein levels or growth, but displayed evidence of failed mitosis and in some cases subsequent cell death (Movies EV3–6). This phenotype is consistent with mitotic failure and catastrophe, common outcomes in cells with centrosome amplification that fail to cluster their supernumerary centrosomes during mitosis (Ganem et al, 2009). Overall, these CRISPR/Cas9 data reflect our findings for OTUD6B siRNA depletion in TNBC cells, suggesting that OTUD6B is essential for long-term survival, and demonstrating *OTUD6B*[-/+] BT549 phenotypes consistent with failed centrosome clustering.

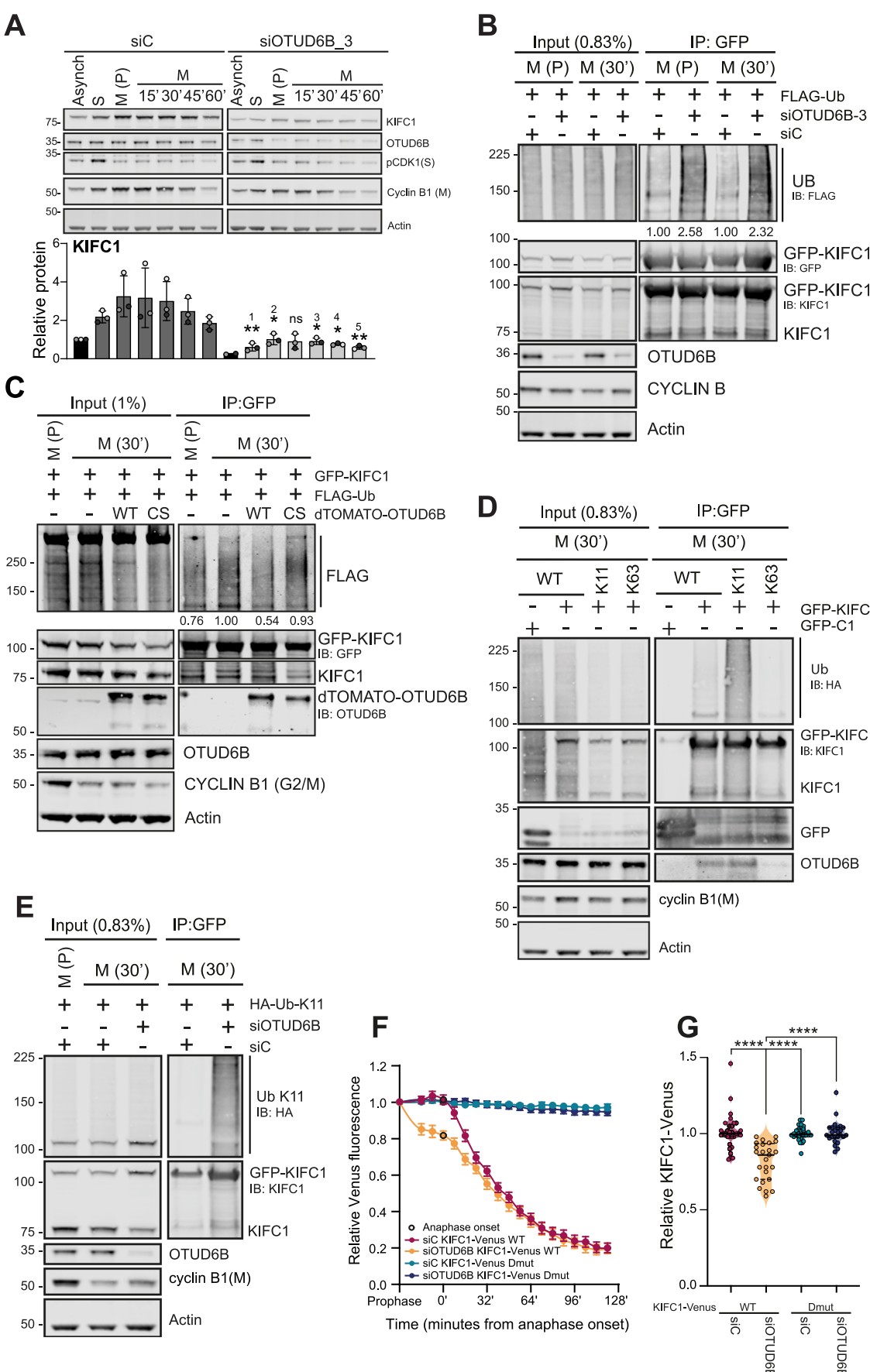

◀ **Figure 6. Regulation of mitotic KIFC1 deubiquitination requires OTUD6B catalytic activity.**

(A) KIFC1 levels are regulated through the cell cycle. U2OS cells were transfected with 10 nM siRNAs for 48 h then synchronised using thymidine/STLC/ZM447439. Representative immunoblots and KIFC1 normalised to asynchronous cells, mean of $n = 3$ biological replicates (or $n = 2$ biological replicates for siOTUD6B_3 asynchronous cells), error bars SD; ns not significant, [1]**$P = 0.0018$, [2]*$P = 0.0247$, [3]*$P = 0.0240$, [4]*$P = 0.0116$, [5]**$P = 0.0034$ by unpaired $t$ test relative to the corresponding siC sample. S S-phase, M mitosis, P prometaphase; min (') after release from prometaphase are indicated. (B–E) U2OS cells were transfected with plasmids (48 h) and siRNAs (10 nM, 72 h), synchronised at mitotic entry (prometaphase) or collected 30 min into mitotic exit (M30') then immunoprecipitated with GFP-nanobeads; $n = 1$ experiment; M45' data also shown in Appendix Fig. S7E. Numbers beneath immunoprecipitation blots indicate FLAG-ubiquitin signal normalised to GFP-KIFC1 pulldown detected by GFP antibody. OTUD6B depletion increases KIFC1 ubiquitylation at mitotic entry and mitotic exit (B). Overexpression of dTOMATO-OTUD6B but not catalytic dead dTOMATO-OTUD6B-C158S decreases KIFC1 ubiquitylation during mitosis (C). KIFC1 can be polyubiquitinated with wild-type or K11-only chains and interacts with endogenous OTUD6B during mitosis (D). OTUD6B depletion increases K11 ubiquitylation of KIFC1 during mitosis (E). (F, G) KIFC1 degradation is affected by OTUD6B depletion. KIFC1-Venus levels quantified in individual mitotic cells after release from double-thymidine block; >24 individual mitotic cells were scored per condition from $n = 2$ biological replicates. In vivo degradation curves plotted as a function of prophase, mean indicated, error bars SEM (F). Scatterplot representing KIFC1-Venus levels in individual cells at anaphase onset relative to prophase; ****$P \leq 0.0001$; relative to siC by Kruskal–Wallis test with Dunn's multiple comparison test (G). Source data are available online for this figure.

## Discussion

Through parallel focused screens we reveal roles for OTUD6B in supporting KIFC1 abundance and pseudo-bipolar spindle formation in TNBC cells. We found that OTUD6B co-localises with KIFC1 on the mitotic spindle, interacts with KIFC1 via its N-terminal substrate binding domain, and can modulate KIFC1 polyubiquitylation and protein levels in early and late mitosis. Importantly, phenotypic rescue of KIFC1 levels and ubiquitination, as well as multipolar spindle formation, are dependent on the catalytic activity of OTUD6B. The timing, and ability to limit K11-ubiquitin chains, suggest OTUD6B opposes the activity of the APC/C towards KIFC1. Notably, OTUD6B is amplified and overexpressed in breast cancer, and we show is required for survival of TNBC cell lines with either high (BT549) or moderate (MDA-MB-231) levels of amplified centrosomes. Together, these findings highlight OTUD6B as a DUB that maintains KIFC1 levels to ensure supernumerary centrosomes remain clustered, allowing cancer cells to avoid mitotic catastrophe and complete cell division.

Amongst the 94 DUBs screened, we found that depletion of only OTUD6B or JOSD2 could both reduce cellular KIFC1 levels and increase multipolar spindles in TNBC cells (Figs. 1 and 2). Whilst no DUBs have previously been reported to regulate KIFC1, candidates from genome-wide screens for centrosome clustering regulators in *Drosophila* or oral squamous cell carcinoma, included orthologs of USP8 (CG5798) and USP43 (CG30421) (Kwon et al, 2008) and the pseudo-DUB USP54 (Leber et al, 2010). However, roles for these DUBs have not been pursued, and they had no notable effect in our DUB-focused TNBC screens (Appendix Table S1). Of our two candidates, we prioritised OTUD6B for mechanistic investigation, as it is frequently altered in clinical breast cancer and correlates with KIFC1 protein levels (Fig. EV1; Appendix Fig. S3).

The ability of OTUD6BB depletion to reduce KIFC1 levels and increase multipolar spindles was evident using several OTUD6B siRNAs and cell lines (Fig. EV2), whilst rescue of both phenotypes required OTUD6B catalytic activity (Fig. 3). A lack of activity for OTUD6B towards di-ubiquitin in vitro had led to speculation about its catalytic capability and precluded determination of its ubiquitin chain specificity (Mevissen et al, 2013; Ritorto et al, 2014). However, in cells OTUD6B binds ubiquitin active site-directed probes (Catic et al, 2007; Pinto-Fernandez et al, 2019) and exhibits

enzymatic activity towards Ub-Met-β-gal that is reliant upon the catalytic cysteine (Xu et al, 2011), confirming it as an active DUB.

In the context of KIFC1 regulation, OTUD6B activity was most evident during mitosis. Whilst modulating OTUD6B had little effect on KIFC1 stability or ubiquitination in interphase cells (Fig. EV5I,J), acute ubiquitination of KIFC1 during mitosis (Appendix Fig. S7D) was decreased by overexpression of catalytically active OTUD6B (Fig. 6C) and increased by OTUD6B depletion (Fig. 6B) particularly for K11 polyubiquitin (Fig. 6E). Recent identification of several OTUD6B substrates has provided more insight into its cellular activity and chain specificity. Although OTUD6B regulates pVHL independently of its DUB activity (Liu et al, 2020), the catalytic cysteine of OTUD6B is required to limit ubiquitination of β-TrCP (Li et al, 2023) and LIN28B (Paulmann et al, 2022), as we found for KIFC1 (Fig. 6C). In common with these two substrates, KIFC1 interacts with the N-terminus of OTUD6B (Fig. 5C) (Li et al, 2023; Paulmann et al, 2022). OTUD6B also exhibits some chain specificity in the context of its substrates, being able to directly remove K48-branched ubiquitin from LIN28B (Ma et al, 2023), and K33 or K11 chains from IRF3 (Wang et al, 2023). The *toxoplasma gondii* OTUD6B ortholog can specifically process K11- and K63-linked ubiquitin (Wilde et al, 2023), although K63 chain removal from irf3 is independent of the catalytic cysteine in *Zebrafish* (Zhou et al, 2021).

This evidence for catalytic activity of OTUD6B towards K11-linked ubiquitin in other contexts is consistent with its ability to reduce mitotic KIFC1 polyubiquitination (Fig. 6) as the APC/C appends K11-linked and branched chains to substrates (Jin et al, 2008; Meyer and Rape, 2014). Early mitotic APC/C activity is limited to substrates like cyclin A and Nek2 until the spindle assembly checkpoint (SAC) is satisfied, when APC/C^Cdc20 degrades Cyclin B1 and Securin, triggering CDK1 inhibition and mitotic exit. This leads to activation of APC/C^Cdh1, which degrades Aurora A and CDC20, regulating anaphase spindle dynamics and cytokinesis (Greil et al, 2022; van Leuken et al, 2009). KIFC1 was shown to be targeted by APC/C^Cdh1 during mitotic exit (Singh et al, 2014). The timing and selectivity of OTUD6B interaction and activity towards K11-ubiquitinated KIFC1 during mitotic exit (Fig. 6D), together with its effect on stability of KIFC1 but not a D-box mutant (Fig. 6G), infer it may directly oppose APC/C^Cdh1. We speculate that such a mechanism may enable cancer cells to maintain a check on KIFC1 ubiquitylation during anaphase and telophase, to ensure

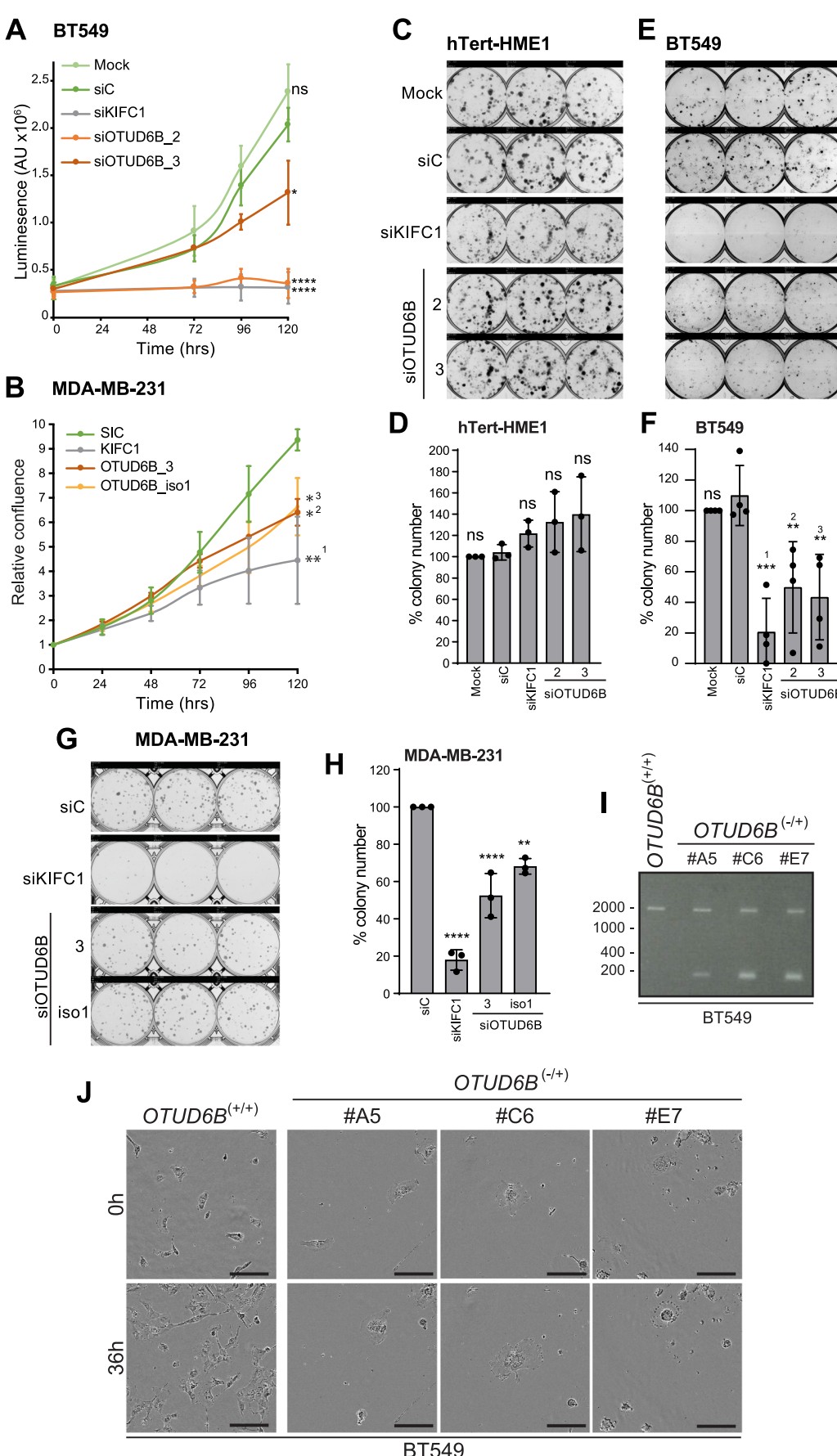

**Figure 7.   OTUD6B is required for breast cancer cell survival.**

(A, B) Growth of breast cancer cells with centrosome amplification is reduced by KIFC1 or OTUD6B depletion. Cell viability and growth were monitored from 72 h to 120 h post-transfection with siRNAs using an ATP-dependent assay for BT549 (40 nM siRNA) (A) or live cell imaging of MDA-MB-231 (10 nM siRNA) (B) in n = 3 biological replicates; mean indicated, error bars SD; ns, not significant, *P = 0.0135, ****P ≤ 0.0001 (A), [1]**P = 0.002, [2]*P = 0.0241, [3]*P = 0.0241 (B), compared to siC by one-way ANOVA with Dunnett's post-hoc test. (C–H) Clonogenic assays in cells transfected with siRNAs. Example image of colonies for hTERT-HME1 or BT549 (40 nM siRNA) (C, E) and MDA-MB231 (10 nM siRNA) (G) quantified using an automated colony counter (D, F, H) from n = 3 biological replicates, error bars SD; ns, not significant, [1]***P = 0.0002, [2]**P = 0.0065, [3]**P = 0.0029 (F), ****P ≤ 0.0001, **P = 0.0012 (H), compared to siC by one-way ANOVA with Dunnett's post-hoc test. (I, J) Genetic editing of *OTUD6B* impairs the growth of TNBC cells. Genomic DNA was extracted from parental or three *OTUD6B* CRISPR-edited BT549 clones (A5, C6, E7) from two gene editing experiments. Endpoint PCR for full length OTUD6B (1972 bp) or CRISPR editing with the sgRNA guide pair ( ~ 190 bp), indicates heterozygous knockout clones (I). Parental BT549 and three CRISPR-edited *OTUD6B*[+/−] clones (A5, C6, E7) 4 weeks post-sgRNA transfection imaged over 36 h to monitor cell division and growth, scale bar 200 μm (J). Source data are available online for this figure.

that supernumerary centrosomes remain clustered so that chromosomes are accurately divided into daughter cells.

However, we found that OTUD6B depletion could increase KIFC1 polyubiquitination not only at mitotic exit, but also in prometaphase arrested cells (Fig. 6B) when it had the strongest effect on KIFC1 degradation. This is consistent with a model where OTUD6B modulates KIFC1 accumulation in early mitosis, when supernumerary centrosome clustering is initially required for spindle formation. Thus, under conditions of low OTUD6B, KIFC1 degradation kinetics (Fig. 6F,G) are shifted towards those of early APC/C substrates like Cyclin A or Nek2, raising the possibility it may also be susceptible to APC/C$^{Cdc20}$ activity. However, APC/C$^{Cdh1}$ is also active in G2 allowing DNA-repair prior to mitotic entry (Wasch et al, 2010), whilst untimely APC/C activity may occur in cancer cells, where the SAC is often defective (Teixeira et al, 2014) and the presence of supernumerary centrosomes may cause prolonged mitotic delay (Karki et al, 2017; Kwon et al, 2008). Thus, premature KIFC1 ubiquitination during G2 or mitotic entry may also be attributable to the APC/C and opposed by OTUD6B. This may also account for the observation that WT OTUD6B re-expression can rescue KIFC1 levels in interphase cells depleted of OTUD6B (Fig. 3D,E). Alternatively, OTUD6B might antagonise another ubiquitin E3 ligase. One potential candidate is TRIM8, which interacts with KIFC1 and Eg5, localises at the mitotic spindle and plays a role in centrosome separation in early mitosis (Venuto et al, 2020). However, rather than degradative K48 or K11 chains, TRIM8 generates K63 (Li et al, 2011), K6 and K33 (Ye et al, 2017) chains.

Although the APC/C is highly processive, two other DUBs were previously reported to oppose its activity. USP37 antagonises APC/C$^{Cdh1}$ to stabilise Cyclin A at S-phase entry (Huang et al, 2011), whilst OTUD7B (Cezanne) exhibits K11-linkage specificity (Mevissen et al, 2013) and deubiquitinates both Cyclin B1 and Aurora A (Bonacci et al, 2018). However, these DUBs appear to lack specificity for KIFC1, as OTUD7B depletion showed no effect in either of our screens, and although USP37 depletion moderately increased multipolar spindle formation this was not associated with decreased KIFC1 levels (Appendix Table S1).

As endogenous OTUD6B levels do not change through the cell cycle (Fig. 6A) (Paulmann et al, 2022), the mitotic activity of OTUD6B might be regulated through post-translational modification and/or sub-cellular recruitment. Indeed, alongside cytoplasmic distribution of OTUD6B (Li et al, 2023; Urbé et al, 2012; Wang et al, 2023), we saw specific enrichment at interphase centrosomes, and on microtubules at mitotic spindle poles (Fig. 4A,D). Thus whilst OTUD6B appears to be largely spatially segregated from

nuclear KIFC1 during interphase, the two proteins co-localise at mitotic entry (Fig. 4B) as KIFC1 focuses microtubules at spindle poles for centrosome tethering (Chavali et al, 2016; She and Yang, 2017). Intriguingly deletion of the N-terminal OTUD6B domain re-localised it from the cytoplasm into the nucleus (Fig. EV4) and ablated OTUD6B at centrosomes (Fig. 5D), suggesting that this domain contains a nuclear exclusion signal and is crucial for centrosome enrichment. There are no consensus centrosome localisation signals to allow bioinformatic prediction, but higher confidence nuclear export signals map to this N-terminal region. Interestingly, a shorter OTUD6B isoform in NSCLC that is proposed to have opposing properties to isoform 1 (Sobol et al, 2017), on which we focus here, may lack these targeting signals.

One initial driver for this study was the idea that identifying a DUB for KIFC1 might present an alternative way to therapeutically target centrosome clustering in breast cancer, particularly TNBC. We found that OTUD6B is frequently amplified or overexpressed leading to elevated protein levels in the TCGA breast cancer cohort (Appendix Fig. S3), where it is associated with worse survival (Fig. EV1E) and basal-like TNBC (Fig. EV1G), and positively correlates with KIFC1 protein levels (Fig. EV1H). Interestingly, OTUD6B protein expression was previously associated with breast cancer in a different context, as OTUDB auto-antibodies were identified in the TgMMTV-neu murine model of breast cancer development suggesting their utility for early detection of breast cancer (Mao et al, 2014). Together these data suggest OTUD6B is a legitimate protein target that is overexpressed in breast cancer. Furthermore, we found that, like KIFC1, OTUD6B is a cancer-specific dependency, as its depletion reduced cell viability and colony formation in two TNBC cancer cell lines with centrosome amplification, but not in cells derived from normal breast (Fig. 7A–H). Although OTUD6B knockdown has a milder effect than KIFC1 knockdown, this likely reflects its longer half-life of 71 h (Schwanhausser et al, 2011), compared with 9.4 h for KIFC1 that it is acutely degraded each mitosis.

The outcome of targeted CRISPR/Cas9 for *OTUD6B* further confirm that it is a fitness gene in TNBC cells with a phenotype indicative of mitotic catastrophe (Fig. 7I,J). Although OTUD6B was not identified as essential in BT549 in the Dependency Map Consortium (https://depmap.org/portal/, (Tsherniak et al, 2017)) genome-wide CRISPR dataset (Behan et al, 2019), these screens are non-saturating (Dede and Hart, 2023) and may miss progressive physiologically relevant effects. Given the population frequency of centrosome amplification (32% in BT549) and its dynamic accumulation (Lee et al, 2014), cells would be predicted to only progressively become susceptible over time if fitness was tightly

linked to the ability to cluster centrosomes. Although targeting other proteins that play roles in centrosome clustering (Rhys and Godinho, 2017; Sabat-Pospiech et al, 2019) can induce multipolar spindles and reduce cancer cell growth, such as Hsp72, Nek6 (Sampson et al, 2017) or STAT3 (Morris et al, 2017), no therapeutic approaches have yet reached clinical trials. Therefore, exploring novel modulators of KIFC1 remains an important area of research. As kinesins are relatively difficult to target pharmaceutically, and complete inhibition of KIFC1 may have non-cancer cell specific effects (Xiao and Yang, 2016), targeting OTUD6B has the potential to be developed as a more tolerable therapeutic route.

In conclusion, our work identifies OTUD6B as a breast cancer-associated deubiquitinase that stabilises KIFC1 during mitosis, to support pseudo-bipolar spindle formation and cell survival.

# Methods

**Reagents and tools table**

| Reagent/resource | Reference or source | Identifier or catalog number |
|---|---|---|
| **Experimental models** | | |
| *Human cell lines:* | | |
| BT549 | ATCC | RRID:CVCL_1092 |
| MDA-MB-231 | ATCC | RRID:CVCL_0062 |
| MCF7 | ATCC | RRID:CVCL_0031 |
| T47D | ATCC | RRID:CVCL_0553 |
| U2OS | ATCC | RRID:CVCL_0042 |
| HEK293T | ATCC | RRID:CVCL_0063 |
| hTert-HME1 | ATCC | RRID:CVCL_3383 |
| MCF10A | ATCC | RRID:CVCL_0598 |
| **Recombinant DNA** | | |
| pDONOR223-OTUD6B (isoform 1) | Gift from Prof Sylvie Urbé (University of Liverpool, UK) | NM_016023.5 |
| eGFP-KIFC1 | Gift from Prof Steve Royle (Warwick Medical School, UK) | NM_002263.4 |
| FLAG-Ub | Gift from Prof Sylvie Urbé (University of Liverpool, UK) | |
| pLVX-TetOne-Puro | Gift from Dr. Leonie Unterholzner (Lancaster University, UK) | |
| HA-Ub K11 only | AddGene | #22901 |
| HA-Ub K63 only | AddGene | #17606 |
| KIFC1-Venus WT and Dmut plasmids | Gift from Prof Catherine Lindon (University of Cambridge, UK) | |

| Reagent/resource | Reference or source | Identifier or catalog number |
|---|---|---|
| GFP-OTUD6B WT | Gift from Prof Sylvie Urbé (University of Liverpool, UK) | NM_016023.5 |
| GFP-OTUD6B CS | Generated in house | |
| GFP-OTUD6B ΔC (1-146) | Generated in house | |
| GFP-OTUD6B ΔN (147-293) WT | Generated in house | |
| GFP-OTUD6B ΔN (147-293) CS | Generated in house | |
| dTOMATO | Generated in house | |
| dTOMATO-OTUD6B WT | Generated in house | |
| dTOMATO-OTUD6B CS | Generated in house | |
| lentiCRISPRv2 | AddGene | #52961 |
| **Antibodies** | | |
| Mouse anti-β-actin | Abcam | ab6276, RRID:AB_2223210 |
| Mouse anti-α-tubulin | Sigma | DM1A, RRID:AB_477583 |
| Mouse anti-centrin | Millipore | 04-1624, RRID:AB_10563501 |
| Mouse anti-cyclin B1 | Millipore | 05-373, RRID:AB_309701 |
| Mouse anti-ubiquitin | Enzo | PW8805, RRID:AB_10541434 |
| Mouse anti-P53 | Santa Cruz | sc-126, RRID:AB_628082 |
| Rabbit anti-actin | Sigma | A2066, RRID:AB_476693 |
| Rabbit anti-pericentrin | Abcam | ab4448, RRID:AB_304461 |
| Rabbit anti-KIFC1 | Abcam | ab72452, RRID:AB_1269257 |
| Rabbit anti-OTUD6B | Proteintech | 25430-1-AP, RRID:AB_2880075 |
| Rabbit anti-USP10 | Abcam | ab72486, RRID:AB_1271412 |
| Rabbit anti-CDK1 | Cell signalling | 9111, RRID:AB_331460 |
| Rabbit anti-NRF2 | Abcam | ab62352, RRID:AB_944418 |
| Rabbit anti-FLAG | Sigma | F7425, RRID:AB_439687 |
| Rabbit anti-HA | Biolegend | 931301, RRID:AB_291262 |
| Rat anti-tubulin | Sigma | MAB1864, RRID:AB_2210391 |
| Sheep anti-GFP | Gift from Prof Ian Prior, University of Liverpool, UK | Polyclonal affinity-purified. |
| **Oligonucleotides and other sequence-based reagents** | | |
| *siRNA sequences:* | | |
| siKIFC1_5 | Qiagen | SI02653210 |
| siOTUD6B_1 | Qiagen | SI00344939 |
| siOTUD6B_2 | Qiagen | SI04172420 |

| Reagent/resource | Reference or source | Identifier or catalog number |
| --- | --- | --- |
| siOTUD6B_3 | Qiagen | SI04185034 |
| siOTUD6B_4 | Qiagen | SI04233915 |
| si_OTUD6B_iso1 | Qiagen | custom design specifically targeting OTUD6B isoform_1: 5'-GAGACGGGAAAAGAAAGCT-3' |
| siO6B_untCON | Qiagen | custom design control complementary to untranscribed sequence immediately 5' to the OTUD6B gene: 5'-TCGTCTTTGCAGTGGCTTT-3' |
| siJOSD2_1 | Qiagen | SI03075541 |
| siJOSD2_2 | Qiagen | SI02648009 |
| siJOSD2_3 | Qiagen | SI00150535 |
| siJOSD2_4 | Qiagen | SI00150528 |
| siUSP3_1 | Qiagen | SI00089432 |
| siUSP3_2 | Qiagen | SI00089439 |
| siUSP3_3 | Qiagen | SI00089446 |
| siUSP3_4 | Qiagen | SI03071432 |
| siUSP10_1 | Qiagen | SI00072989 |
| siUSP10_2 | Qiagen | SI00302113 |
| siUSP10_3 | Qiagen | SI00073003 |
| siUSP10_4 | Qiagen | SI00073010 |
| siC (All-Stars negative control) | Qiagen | SI03650318 |
| siNT1 (non-targeting) | Dharmacon | D-001810-10-05 |
| *Cloning & mutagenesis primers:* | *This study* | *Below* |
| OTUD6B_siRNA res (For) | | 5'-TGCATTGGAAAAGGAACGGGAGGAGCGGATAGCTGAAGCTG-3' |
| OTUD6B_CatMut (For) | | 5'-CCATCTGATGGCCACTCTATGTATAAAGCCATTG-3' |
| GFP-OTUD6B ΔC (1-146) (For) | | 5'-CTCAAGCTTCGAATTATGGAGGCGGTATTGACC-3' |
| GFP-OTUD6B ΔC (1-146) (Rev) | | 5'-GTCGACTGCAGAATTTCACTGTCTAGCTGCCAATATTTG-3' |
| GFP-OTUD6B ΔN (147-293) WT (For) | | 5'-CTCAAGCTTCGAATTTTAGAAATTAAACAGATTCCATCT-3' |
| GFP-OTUD6B ΔN (147-293) WT (Rev) | | 5'-GTCGACTGCAGAATTTCAGCTGCAATTTTCAGT-3' |
| GFP-OTUD6B ΔN (147-293) CS (For) | | 5'-CTCAAGCTTCGAATTTTAGAAATTAAACAGATTCCATCT-3' |
| GFP-OTUD6B ΔN (147-293) CS (Rev) | | 5'-GTCGACTGCAGAATTTCAGCTGCAATTTTCAGT-3' |
| dTOMATO (For) | | 5'-TGAACCGTCAGATCCGCTAGCACCATGTCAGTGAGCAAGGG-3' |
| dTOMATO (Rev) | | 5'-GAAGCTTGAGCTCGATTACTTGTACAGCTCGTCCATGC-3' |

| Reagent/resource | Reference or source | Identifier or catalog number |
| --- | --- | --- |
| dTOMATO-OTUD6B WT and CS (For) | | 5'-GCTGTACAAGATGGAGGCGGTATTGACC-3' |
| dTOMATO-OTUD6B WT and CS (Rev) | | 5'-AATTCGAAGCTTGAGCTCGACTAGCTGCAATTTTCAGTAACTATG-3' |
| GFP-KIFC1_siRNA res (For) | | 5'-CTGGGACCTTAAGGGTCAGTTATGTGACCtaaatgcagaac-3' |
| GFP-KIFC1_siRNA res (Rev) | | 5'- acccttAAGGTCCCAGGCTGGACGTTTGCTGG -3' |
| *CRISPR guides & PCR primers:* | *This study* | *Below* |
| OTUD6B sgRNA_1 | | 5'-ATTGACCGAAGAGCTTGATGAGG-3' |
| OTUD6B sgRNA_2 | | 3'-TACCTCACAGTGTGTTACTGAGG-5' |
| OTUD6B CRISPR seq (For) | | 5'-AGGTTTCTTCTAGCG-3' |
| OTUD6B CRISPR seq (Rev) | | 3'-TGGCAATCAACAGCTG-5' |
| **PCR primers** | **This study** | **Below** |
| ACTB (For) | | 5'-CACCTTCTACAATGAGCTGCGTGTG-3' |
| ACTB (Rev) | | 5'-ATAGCACAGCCTGGATAGCAACGTAC-3' |
| GAPDH (For) | | 5'-CAATGACCCCTTCATTGACC-3' |
| GAPDH (Rev) | | 5'-GACAAGCTTCCCGTTCTC-3' |
| USP3 (For) | | 5'-TGCCATCCTTCAGTCACTC-3' |
| USP3 (Rev) | | 5'-TTCCTGCTGTTTTCCCATTC-3' |
| JOSD2 (For) | | 5'-ACGCCCTCAACAACGTTCTG-3' |
| JOSD2 (Rev) | | 5'-GCGGCCATGATCACATTGAC-3' |
| KIFC1 (For) | | 5'-TAGAGGAGAAGGAGAGGAGG-3' |
| KIFC1 (Rev) | | 5'-TTCCCGCTCAGTCAGTAAG-3' |
| OTUD6B (For) | | 5'-GAGCTTGATGAGGAAGAGCAG-3' |
| OTUD6B (Rev) | | 5'-GTCATTCTTGGGAACAGCATTC -3' |
| **Chemicals, enzymes and other reagents** | | |
| RNAiMax | Life Technologies | 13778075 |
| Genejuice | Sigma | 70967 |
| Gateway cloning system | Invitrogen | 12535-019 |
| In-Fusion cloning | Takara | 638945 |
| Cycloheximide | Sigma | C4859 |
| Epoxomicin | Sigma | 134381-21-8 |
| Thymidine | Sigma | T1895-5G |
| S-Trityl-L-cysteine | Tocris | 2191 |
| ZM447439 | Tocris | 2458 |
| SiR700-DNA | Spirochrome | SC015 |
| HA-Ub-PA | UbiQ | UbiQ-078 |

| Reagent/resource | Reference or source | Identifier or catalog number |
|---|---|---|
| **Software** | | |
| Image J. Fiji with the plug in JACoP | | RRID:SCR_003070 RRID:SCR_025164 |
| FlowJo v10.8.1 software | | RRID:SCR_008520 |
| Prism v6.00 for Mac | GraphPad | RRID:SCR_002798 |
| Image Studio 6.0 | LI-COR Biosciences | RRID:SCR_015795 |
| **Other** | | |
| Eclipse Ti microscope (with CFI Super Plan Fluor ELWD ADM 20XC N.A. 0.45) | Nikon | RRID:SCR_021242 |
| LSM800 confocal microscope (with Plan Apochromat 63X NA 1.4 OIL objective) | Zeiss | RRID:SCR_015963 |
| Odyssey CLX system | LI-COR Biosciences | RRID:SCR_014579 |
| CFX real-time PCR detection system | Bio-Rad | RRID:SCR_018064 |
| IncuCyte S3 Live Cell Analysis System | Essen Bioscience | RRID:SCR_023147 |
| GloMax Explorer Multimode Plate Reader | Promega | RRID:SCR_015575 |
| GelCount colony analyser | Oxford Optronics | RRID:SCR_023219 |
| Attune NxT system | ThermoFisher | RRID:SCR_019590 |

## Cell culture

All cell lines were obtained from the ATCC, authenticated by STR profiling, verified as mycoplasma free and cultured for limited passage numbers. BT549 cells were cultured in RPMI (Invitrogen) supplemented with 10% fetal bovine serum (FBS) and 72 ng/ml Insulin (Sigma). MDA-MB-231, MCF7, T47D, U2OS and HEK239T cells were cultured in DMEM (Invitrogen) supplemented with 10% FBS and 1% non-essential amino-acids (NEAA, Gibco). hTert-HME1 cells were cultured in MEGM plus supplements, according to ATCC guidelines (MEGM BulletKit, CC-3150, Lonza, without addition of GA-1000). MCF10A cells were cultured in DMEM F12 (Invitrogen) supplemented with 10% FBS, 5% horse serum (Invitrogen), 20 ng/ml EGF (Invitrogen), 10 ng/ml cholera toxin (Sigma), 50 ng/ml hydrocortisone (Invitrogen) and 10 μg/ml insulin. All cells were cultured in a humidified incubator at 37 °C and 5% $CO_2$.

## DUB family siRNA screens

A custom designed DUB siRNA library consisting of four pooled oligos for each of 94 human DUBs (Qiagen) was used for the RNAi screens. For the KIFC1 immunoblotting screen, BT549 cells were seeded at $2.1 \times 10^5$ cells per 6 cm dish and 24 h later transfected with 40 nM siRNA using Lipofectamine RNAiMax (Life Technologies). 72 h after transfection, cells were lysed in hot Laemmli buffer. For the microscopy screen BT549 cells were seeded at $8 \times 10^3$ cells per well in 96-well plates and reverse transfected with 40 nM siRNA using Lipofectamine RNAiMax (Life Technologies). 72 h post-transfection, cells were fixed, stained for immunofluorescence and imaged on a Nikon Eclipse Ti microscope (CFI Super Plan Fluor ELWD ADM 20XC N.A. 0.45). For each condition, a single stitched image (6 × 6 fields of view, 7308 × 5460 pixels) was captured, and all cells were analysed for the presence of multipolar mitoses. Images were scored blind, before well numbers were matched back to the DUBs they represented in the siRNA library. The median number of cells per image was 4503, with a median of 74 mitotic cells. The same format was used for the multipolar mitosis deconvolution experiments.

## siRNA and plasmid transfection

For siRNA transfection, cells were seeded at an appropriate density and transfected the following day with 10 nM or 40 nM siRNA (as indicated in figure legends) using RNAiMax (Life Technologies); cells were analysed 72 h later, unless otherwise stated. For plasmid transfection, cells were seeded at an appropriate density and transfected with plasmid DNA using GeneJuice (Sigma-Aldrich) and analysed 48 h later, unless otherwise stated.

## Cloning and mutagenesis

pDONOR223-OTUD6B (Urbé et al, 2012) was used to sequentially generate siRNA-resistant (siOTUD6B-3) and catalytically inactive forms of OTUD6B by Quickchange site-directed mutagenesis using complementary primer pairs. pDONOR223 plasmids were sequence verified and shuttled into the destination vector pEGFP_GW using the Gateway system to add an N-terminal GFP-tag. siOTUD6B-3 resistant dTOMATO-OTUD6B WT and C158S, as well as eGFP-OTUD6B domain plasmids ΔC, ΔN-WT and ΔN-C158S were generated by In-Fusion cloning using primers designed to have a 15 bp overlap to the plasmid vector.

## Generation of inducible MDA-MB-231 GFP-KIFC1_siRNA-resistant cells

To create constructs expressing siRNA-resistant GFP-KIFC1 (siRes, to siKIFC1_5), site-directed mutagenesis was performed on a pEGFP-N1-KIFC1 plasmid and verified by sequencing. To generate the lentiviral vector expressing GFP-KIFC1siRes, GFP-KIFC1 was PCR-amplified from the pEGFP-N1-KIFC1-siRes plasmid and ligated into pLVX-TetOne-Puro vector. Lentivirus was produced by seeding HEK-293T cells in antibiotic-free medium and transfecting them with the pLVX-TetOne-Puro-GFP-KIFC1r lentiviral plasmid, along with the Gag-Pol (psPAX2) and VSV-G (pMD2.G) plasmids, using Genejuice (Sigma) according to the manufacturer's instructions. The lentivirus was harvested 24 and 48 h post-transfection. For infection, the lentivirus was mixed with polybrene (8 μg/ml) and added dropwise to MDA-MB-231 cells. Puromycin selection (1 μg/ml) began 48 h after lentiviral transduction. GFP-KIFC1siRes expression was confirmed via

immunoblotting and flow cytometry for KIFC1 after treating cells with 2 µg/ml doxycycline for 48 h.

## CRISPR cell-line generation

OTUD6B$^{-/-}$ or OTUD6B$^{-/+}$ cell-lines were generated using CRISPR/Cas9 technology. HEK-293T or BT549 cells were seeded at $1 \times 10^5$ or $7.5 \times 10^4$ respectively, in 12-well plates. 24 h later, cells were transfected with 1 µg lentiCRISPRv2 (Sanjana et al, 2014) and sgRNA guide pairs using PEI (Sigma) for 24 h. The culture media was subsequently replaced with media containing 2 µg/ml (HEK-293T) or 1.25 µg/ml (BT549) puromycin (Sigma) and cells incubated for 72 h. Cells were simultaneously seeded into 96-well plates at 0.5 cells/well for clonal selection or $5 \times 10^3$ cells/well for initial mixed population PCR screening. Genomic DNA was extracted from the mixed population of lentiCRISPRv2 transfected cells by hot lysis using 20 mM NaOH, and CRISPR editing by both sgRNAs confirmed by genomic endpoint PCR. Following expansion, individual clones were again screened for homozygous or heterozygous knockouts by genomic endpoint PCR prior to validation by immunoblot.

## Drug treatments

Cycloheximide (CHX, Sigma C4859) was added to cells at a final concentration of 10 µg/ml the day after plating, or 72 h after siRNA transfection; following incubation for 1 to 24 h cells were lysed for immunoblotting. DMSO or 50 nM epoxomicin (EPO, Sigma 134381-21-8) were added to cells the day after plating, 72 h after siRNA transfection, or 24 h after plasmid transfection, and cells lysed for immunoblotting 6 h later. U2OS cells were either synchronised with double-thymidine block to enrich cells at S-phase for live cell imaging experiments or by using a protocol adapted from (Min et al, 2014) to obtain populations enriched for S-phase, prometaphase, or mitotic exit (30, 45 or 70 min) for ubiquitination experiments. For the double-thymidine block, the day after seeding, medium supplemented with 2.5 mM thymidine (Sigma, T1895-5G) was added to the cells for 18 h. Cells were then washed and released in fresh full medium for 8 h. Thymidine-supplemented medium was added to the cells for the second block for 18 h. Cells were then washed and released for 5 h before starting imaging. To enrich cells at mitotic exit, the day after seeding, 2.5 mM thymidine (Sigma, T1895-5G) was added for 21 h (G1/S) and washed-out for 3 h (S/G2). Cells were then treated with 10 µM S-Trityl-L-cysteine (STLC, Tocris, 2191, Bristol, UK), a reversible Eg5 inhibitor, for 16 h to arrest cells in prometaphase with monopolar spindles (Skoufias et al, 2006); the prophase population was collected by mitotic shake-off. STLC was washed out and cells treated with 10 µM ZM447439 (Torcis, 2458), an aurora kinase inhibitor, to stimulate mitotic exit (Georgieva et al, 2010; Min et al, 2014). In experiments combining siRNA or plasmid transfection with synchronisation, the drug treatment cascade was initialised 24 h post-transfection with siRNA and 6 h post transfection with plasmids.

## Immunofluorescence and microscopy

Cells were either fixed with 4% paraformaldehyde, quenched with ammonium chloride and permeabilised with 0.1% triton, or fixed with −20 °C methanol, prior to blocking and primary antibody incubation. Alexa-Fluor 488, Alexa-Fluor 594 and Alexa-Fluor 647 coupled secondary antibodies (Molecular Probes) were used. Coverslips were mounted on Moviol supplemented with 4′,6-diamidino-2-phenylindole (DAPI) at 1:10,000 or cells incubated with Hoechst (Sigma) at 1:1000 for 5 min before mounting. Cells were imaged using a Nikon Eclipse Ti microscope with a CFI Plan Apochromat 40X NA 0.95 objective, or a Zeiss LSM800 confocal microscope with a Plan Apochromat 63X NA 1.4 OIL objective and ZEN 2.3.sp1 software, as indicated in figure legends. Image J. Fiji with the plug in JACoP (Bolte and Cordelieres, 2006) was used to quantify fluorescence co-localisation by Pearson's correlation coefficient. KIFC1 fluorescence intensity in interphase cells was measured in Image J. Fiji by using the DAPI-stained nuclei to define the region of interest in which to measure the KIFC1 channel mean intensity.

For live cell imaging of Venus-tagged proteins, U2OS cells were seeded in a 12-well black glass-bottom plate for super-resolution imaging (P12-1.5H-N, In Vitro Scientific, CA, USA) and transfected with 10 nM siRNA (72 h) and KIFC1-Venus WT or Dmut plasmids (Min et al, 2014) (48 h). Cells were synchronized with double-thymidine block, washed and released in medium supplemented with 50 nM siR700-dna (Spirochrome) for 5 h before starting imaging with a Nikon Eclipse Ti microscope, using CFI Plan Apochromat 40 Å~ N.A. 0.95, W.D. 0.14 mm objective, within an incubation chamber to maintain 37 °C and 5% CO$_2$. At least 8 individual positions per well per experiment were imaged for 24 h every 8 min.

## Immunoblotting

Whole cell extracts were prepared by direct addition of hot Laemmli buffer and incubation at 110 °C for 10 min with intermittent vortexing. Protein concentrations were determined after suitable dilution using Bicinchoninic Acid (BCA) assay (Thermo Scientific). Following resolution by SDS-PAGE, proteins were transferred to BiotraceNT membrane (VWR) and incubated with primary antibodies. Proteins were visualized using donkey anti-mouse or anti-rabbit secondary antibodies conjugated to the IRDyes IR680-LT, or IR800 (LI-COR) and a LI-COR Odyssey system, with images quantified in ImageStudio.

## Immunoprecipitation

Cells were lysed in NP-40 lysis buffer (0.5% NP-40, 25 mM Tris pH 7.5, 100 mM NaCl, 50 mM NaF, 2 mM MgCl$_2$, 1 mM EGTA, containing protease inhibitors and PhosStop (Sigma-Aldrich) for 20 min on ice, then pre-cleared by centrifugation. Lysates were adjusted to 1 mg/ml in lysis buffer and subject to overnight immunoprecipitation at 4 °C with GFP-NanoTrap beads (prepared in house) to pull-down GFP-tagged proteins. Immunoprecipitates were washed three times with wash buffer 1 (0.5% NP-40, 25 mM Tris pH 7.5, 100 mM NaCl, 50 mM NaF, 2 mM MgCl$_2$) and once in wash buffer 2 (10 mM Tris pH 7.5, 2 mM MgCl$_2$) before elution in 3× sample buffer (5 min at 95 °C) and analysis by immunoblotting. To assess endogenous ubiquitination, the membrane was boiled for 30 min before blocking and incubation with primary antibody.

## Ubiquitin active site-directed probe assay

Cells were lysed in non-denaturing buffer (50 mM Tris pH 7.5, 5 mM MgCl$_2$, 250 mM sucrose, 1 mM DTT, 2 mM ATP) on ice and homogenized by progressively passing through 23 G, 26 G and 30 G needles, PhosStop (Sigma-Aldrich) was added immediately afterwards. Lysates were cleared by centrifugation (20 min, 14,000 rpm, 4 °C) and a BCA assay performed to standardise protein concentrations. 15 µg protein was incubated with 30 ng HA-Ub-PA (UbiQ-078, UbiQ) at a 1:50 ratio of probe:protein, for 60 min at 37 °C with shaking at 300 rpm. 5× sample buffer (15% SDS, 312.5 mM Tris pH 6.8, 50% glycerol, 16% β-mercaptoethanol, bromophenol blue) was used to terminate the reaction at 95 °C for 5 min before SDS-PAGE.

## RNA extraction and real-time PCR

Total RNA was extracted using RNeasy columns (Qiagen) and cDNA was reverse transcribed from 1 µg RNA with RevertAid H-minus M-MuLV reverse transcriptase (Fermentas) using an oligo-dT primer (Promega). Quantitative real-time RT-PCR (qRT-PCR) was performed in triplicate using SYBR Green supermix and a CFX real-time PCR detection system (Bio-Rad). Samples underwent 2-step amplification at 94 °C (30 s) and 60 °C (60 s); melt curves were analysed after 40 cycles. The Ct values for test genes were normalised to housekeeping genes and relative expression represented as $2^{-[\Delta Ct]}$ for the cell panel or $2^{-[\Delta\Delta Ct]}$ relative to a comparator sample in experiments.

## Cell survival and clonogenic assays

CellTitre-Glo luminescent cell viability assays (Promega) or live cell imaging using an Incucyte S3 (Essen Bioscience) were used to measure cell proliferation and survival. For the CellTitre-Glo assay, optimised cell numbers were plated into white-walled 96 well plates and reverse transfection of KIFC1 or OTUD6B siRNA performed. Replicate plates were analysed at 16 h (baseline, $t = 0$), 72 h, 96 h and 120 h using a luminescence plate reader (GloMax, Promega). For live imaging, a suitable number of cells were seeded in six-well plates and siRNA transfection performed 24 h later. Cells were incubated in the Incucyte S3 and confluence monitored for 120 h after siRNA transfection. For clonogenic assays, cells were transfected with siRNAs and 72 h later re-plated into six-well plates at a low-density of 2000 cells/well (hTERT-HME1 and BT549) or 200 cells/well (MDA-MB-231), optimised to give 50–100 colonies per well in control conditions. Cells were left for 10–18 days before fixing and staining with Crystal Violet (Sigma). Colonies were counted on a GelCount colony analyser (Oxford Optronics). Following CRISPR editing, cells were imaged in the Incucyte S3 at 2 h intervals for 36 h.

## Cell cycle analysis

Suitable numbers of cells were seeded 24 h before siRNA transfection. Cells were collected after 72 h and fixed prior to 7-aminoactinomycin D (7-AAD, Invitrogen) staining to determine DNA content. The fluorescence of single cells was quantified using a Attune NxT system and analyzed using FlowJo v10.8.1 software.

## Bioinformatics and statistical analysis

The TCGA ($n = 1084$) and CPTAC ($n = 105$) datasets (Data refs: Berger et al, 2018; Mertins et al, 2016) were accessed and analysed using cBioportal in March 2022 (Cerami et al, 2012; de Bruijn et al, 2023; Gao et al, 2013). Biochemical measurements represent several thousand cells; these data are usually represented as the mean value from three independent experiments, with error bars showing standard deviation (as indicated in figure legends). No statistical method was used to predetermine sample size. For phenotypes of individual cells, where feasible, we aimed to count/score at least 100 cells per condition, unless the experiment precluded this. All statistical tests for experimental data were performed using GraphPad Prism version 6.00 for Mac; $P$ values less than 0.05 were considered significant. Fisher exact test was used to analyse differences in the frequencies of different phenotypes. Other data were analysed by $t$ test or ANOVA as appropriate (indicated in figure legends). These parametric tests are suitable for continuous data sets without off-scale measurements and assume Gaussian distribution. Pearson coefficient ($r_P$) was used to test correlation in datasets characterised by Gaussian distribution and Spearman coefficient ($r_s$) in datasets of non-Gaussian distribution, which was estimated by the D'Agostino and Pearson omnibus normality test.

## Data availability

Raw microscopy image data are available at BioImages, accession number S-BIAD1417.

The source data of this paper are collected in the following database record: biostudies:S-SCDT-10_1038-S44319-024-00361-w.

## Peer review information

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

## Acknowledgements

Wellcome Trust PhD studentship 109307/Z/15/Z to JMC, IP and DS-P. North West Cancer Research project grant CR1050 to JMC and ABF. Breast Cancer Now PhD studentship 2018NovPhD1269 to JMC, MM, IP and VM. North West Cancer Research grant NWCRLAN2024.01 to ABF and AT. This research was funded, in part, by the Wellcome Trust [109307/Z/15/Z]. For the purpose of open access, the author has applied a CC BY public copyright licence to any Author Accepted Manuscript version arising from this submission. The authors acknowledge use of the Centre for Cell Imaging (CCI) provided by Liverpool Shared Research Facilities, Faculty of Health and Life Sciences, University of Liverpool.

## Author contributions

**Valeria E Marotta**: Data curation; Formal analysis; Validation; Investigation; Visualization; Methodology; Writing—review and editing. **Dorota Sabat-Pośpiech**: Data curation; Formal analysis; Funding acquisition; Validation; Investigation; Visualization; Methodology; Writing—review and editing. **Andrew B Fielding**: Conceptualization; Data curation; Formal analysis; Supervision; Funding acquisition; Investigation; Visualization; Methodology; Writing—original draft; Writing—review and editing. **Amy H Ponsford**: Investigation; Methodology; Writing—review and editing. **Amanda Thomaz**: Investigation; Methodology; Writing—review and editing. **Francesca Querques**: Investigation; Writing—review and editing. **Mark R Morgan**: Supervision; Funding acquisition; Writing—review and editing. **Ian A Prior**: Supervision; Funding acquisition; Writing—review and editing. **Judy M Coulson**: Conceptualization; Data curation; Supervision; Funding acquisition; Visualization; Methodology; Writing—original draft; Project administration; Writing—review and editing.

Source data underlying figure panels in this paper may have individual authorship assigned. Where available, figure panel/source data authorship is listed in the following database record: biostudies:S-SCDT-10_1038-S44319-024-00361-w.

## Disclosure and competing interests statement

Andrew Fielding and Amanda Thomaz declare competing commercial interests due to intellectual property held in field related to this study. The remaining authors declare no competing interests.

# Expanded View Figures

**Figure EV1.   Relationship between OTUD6B, centrosome amplification, KIFC1 expression and patient survival.**

(A) Centrosome amplification in breast cancer and control cell lines. Cells were stained for α-tubulin (green), pericentrin (red), and DNA (DAPI, blue) and visualised using a Nikon Eclipse Ti fluorescent microscope. Centrosome number was scored in >100 mitotic cells per line. Scale bar 10 μm. (B–D) Correlation of KIFC1 and OTUD6B protein expression in a breast cell line panel. Representative immunoblot and quantitation normalised to actin shown relative to U2OS cell line, $n = 1$ experiment (B). Scatter plots comparing KIFC1 (C) or OTUD6B (D) with percentage centrosome amplification by two-tailed Pearson coefficient. MCF7 excluded from correlation as centrosome amplification low compared to other studies. (E) OTUD6B mRNA overexpression is an indicator of poor prognosis in breast cancer patients. Kaplan–Meier estimate of overall survival for all breast cancer patients stratified by high OTUD6B mRNA expression relative to diploid samples in the TCGA breast invasive carcinoma PanCancer Atlas; $n = 1084$ samples, $P = 0.00046$, long rank test. (F, G) KIFC1 and OTUD6B mRNA expression are most highly elevated in basal-like breast cancer in the TCGA PanCancer dataset. Patient data were stratified according to subtype; $n = 981$, ****$P \leq 0.0001$ Kruskal–Wallis test with Dunn's multiple comparisons test. (H) Positive correlation of KIFC1 and OTUD6B protein expression in breast cancer patient samples classified as basal-like. Scatter plot comparing expression by mass spectrometry from the CPTAC breast invasive carcinoma dataset; $n = 23$, *$P = 0.0452$ Pearson correlation.

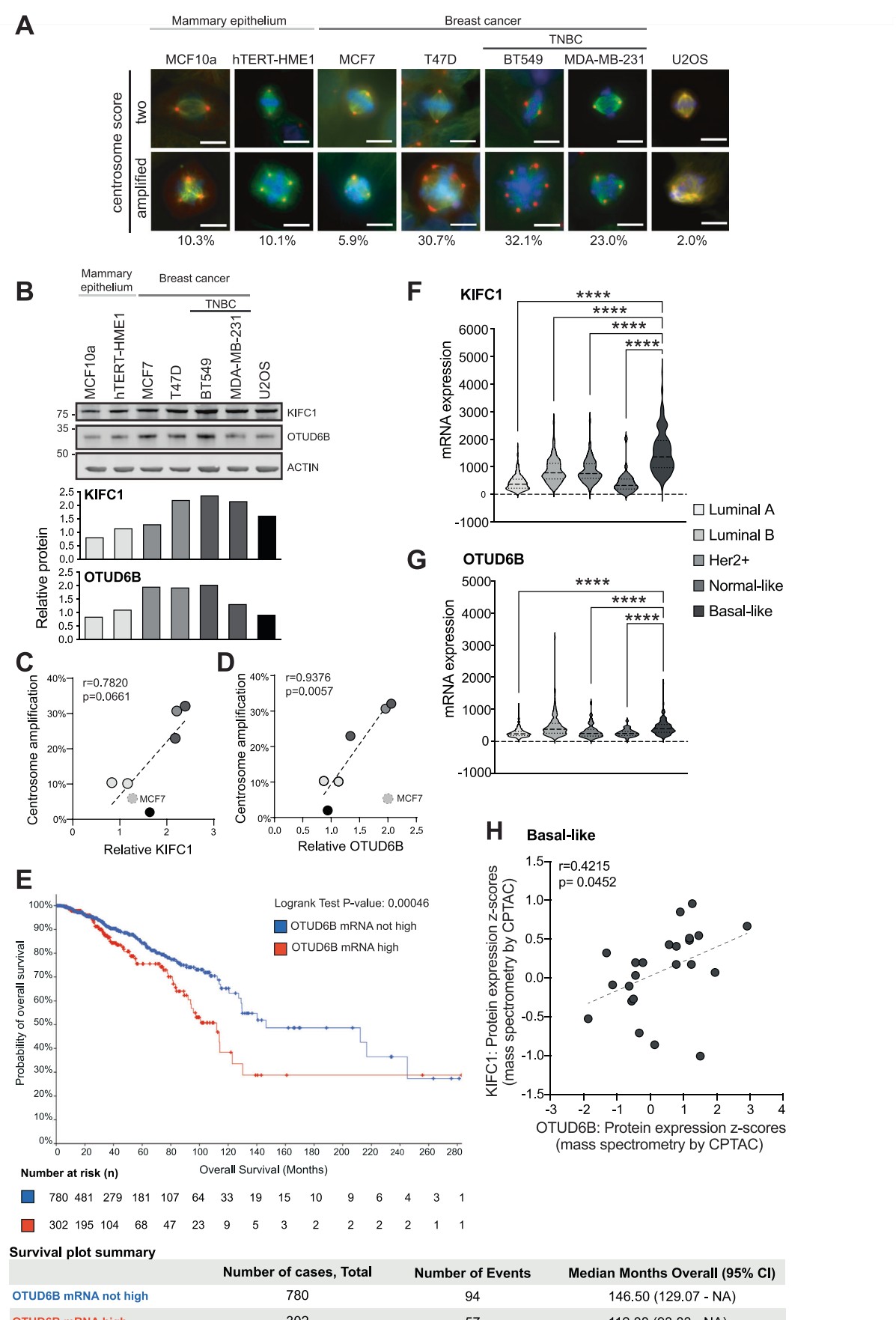

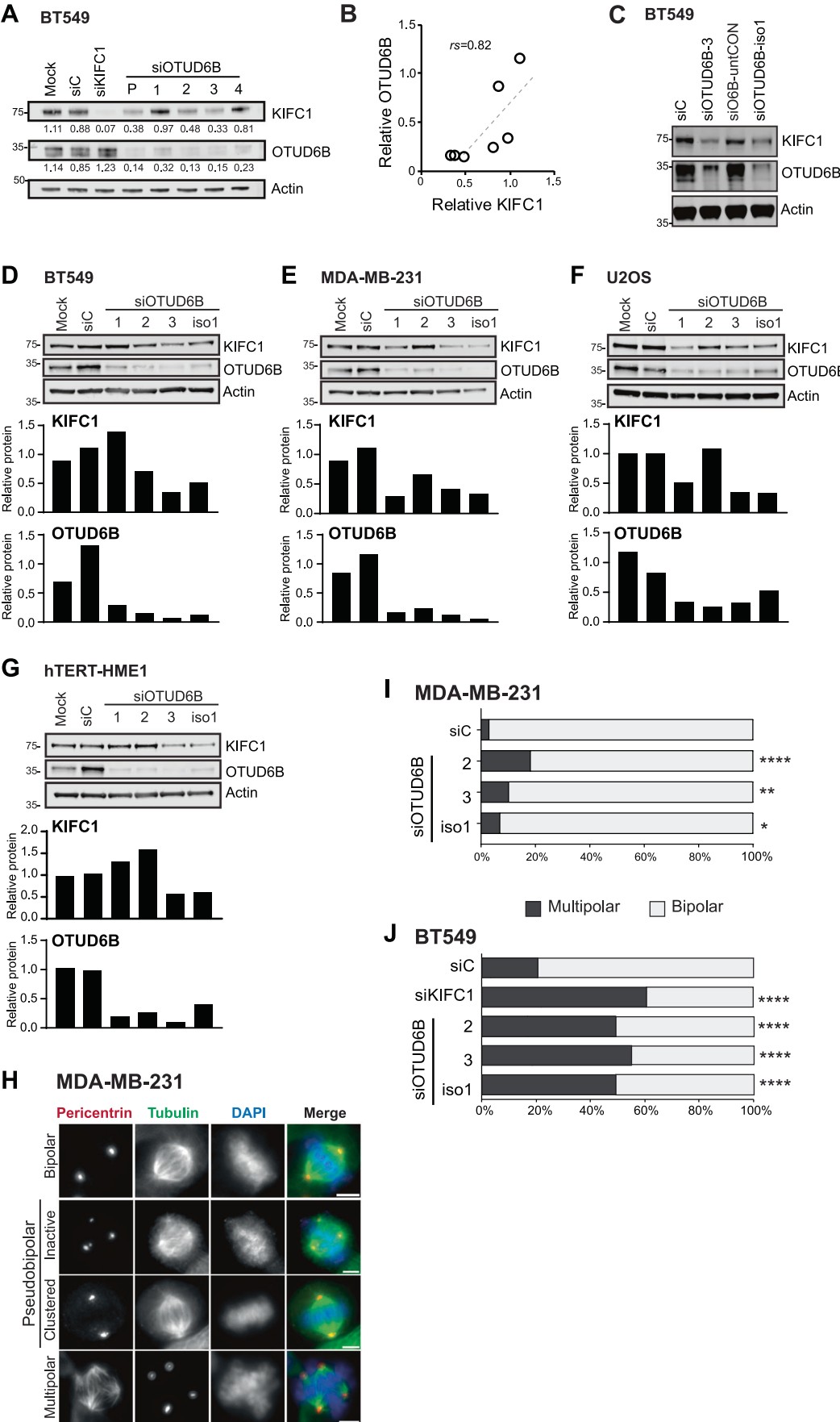

◀ **Figure EV2.** **OTUD6B knockdown reduces KIFC1 expression in various cell lines and increases multipolar spindles in breast cancer cell lines.**

(A–G) Cells were transfected with 10 nM OTUD6B or control siRNAs and analysed 72 h post-transfection; protein levels normalised to actin and mean of the controls, one experiment shown. BT549 (A–D) showing correlation between OTUD6B and KIFC1 across samples (B). An siRNA specifically targeting the major OTUD6B isoform_1 (iso1) but not an untranscribed sequence 5′ of the OTUD6B gene (siO6B_untCON) reduces KIFC1 (C). OTUD6B depletion also reduces KIFC1 levels in MDA-MB-231 (E), U2OS (F) and hTERT-HME1 (G). (H–J) Three OTUD6B siRNAs induce multipolar spindles in TNBC cell lines with centrosome amplification. Cells were co-stained for centrosomes (pericentrin), spindle (tubulin) and DNA (DAPI); scale bar 10 μm (H). >100 metaphase cells were scored in each condition in MDA-MB-231 from $n = 1$ experiment, ****$P \leq 0.0001$, **$P = 0.0027$ and *$P = 0.0119$ (I), or BT549 cells across $n = 3$ biological replicates, ****$P \leq 0.0001$ (J) by One-sided Fisher's exact test compared to siC.

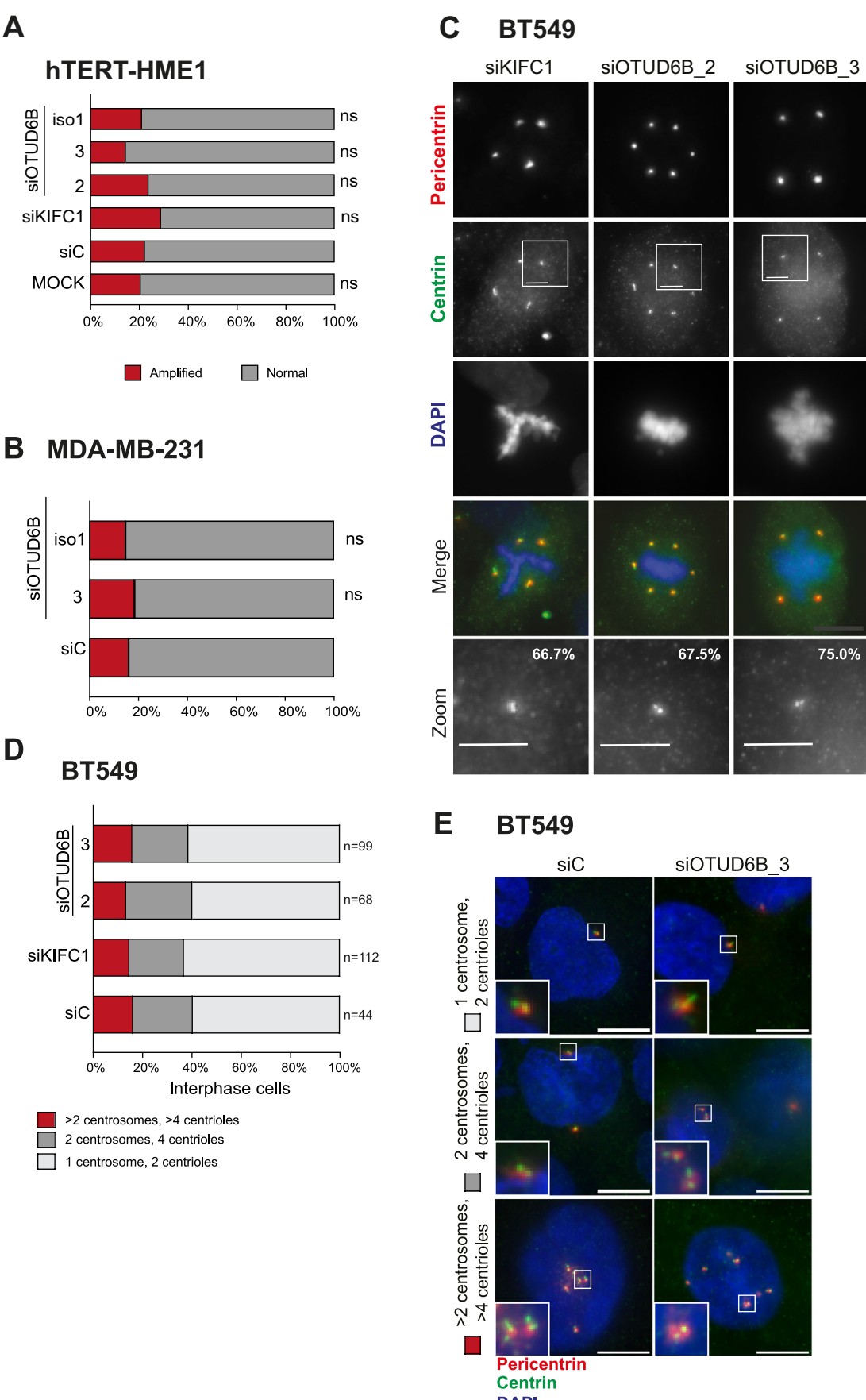

◀ **Figure EV3.** **OTUD6B depletion mimics the centrosome de-clustering phenotype of KIFC1 depletion without altering centrosome number.**

(A, B) OTUD6B depletion does not induce centrosome amplification. The hTERT-HME1 breast epithelial cell line (A) or the MDA-MB-231 TNBC cell line (B) were transfected with 10 nM siRNA for 72 h, stained with tubulin and pericentrin and imaged with Nikon Eclipse Ti fluorescent microscope. >100 cells per condition were scored from $n = 3$ biological replicates (A) or 1 biological replicate (B); ns, not significant by One-sided Fisher's exact test compared to siC. (C) OTUD6B depletion mimics centrosome de-clustering. BT549 cells were transfected with 40 nM siRNA for 72 h then co-stained for pericentrin, centrin and DNA (DAPI), $n = 1$ experiment. Z-stacks of individual mitotic cells were acquired with a Nikon Eclipse Ti fluorescent microscope, ( >15 mitotic cells per condition) and images presented as maximum-intensity projections. Insets show representative centrin foci; the percentage of multipolar spindles where all pericentrin foci contain at least two centrin foci is indicated. Scale bars, 10 μm. (D, E) OTUD6B depletion does not change centriole or centrosome number in interphase cells and does not induce centriole splitting. Experiment as described in (C) with >43 interphase cells imaged and scored for centriole and centrosome number ($n$ indicated on graph), scale bar 5 μm.

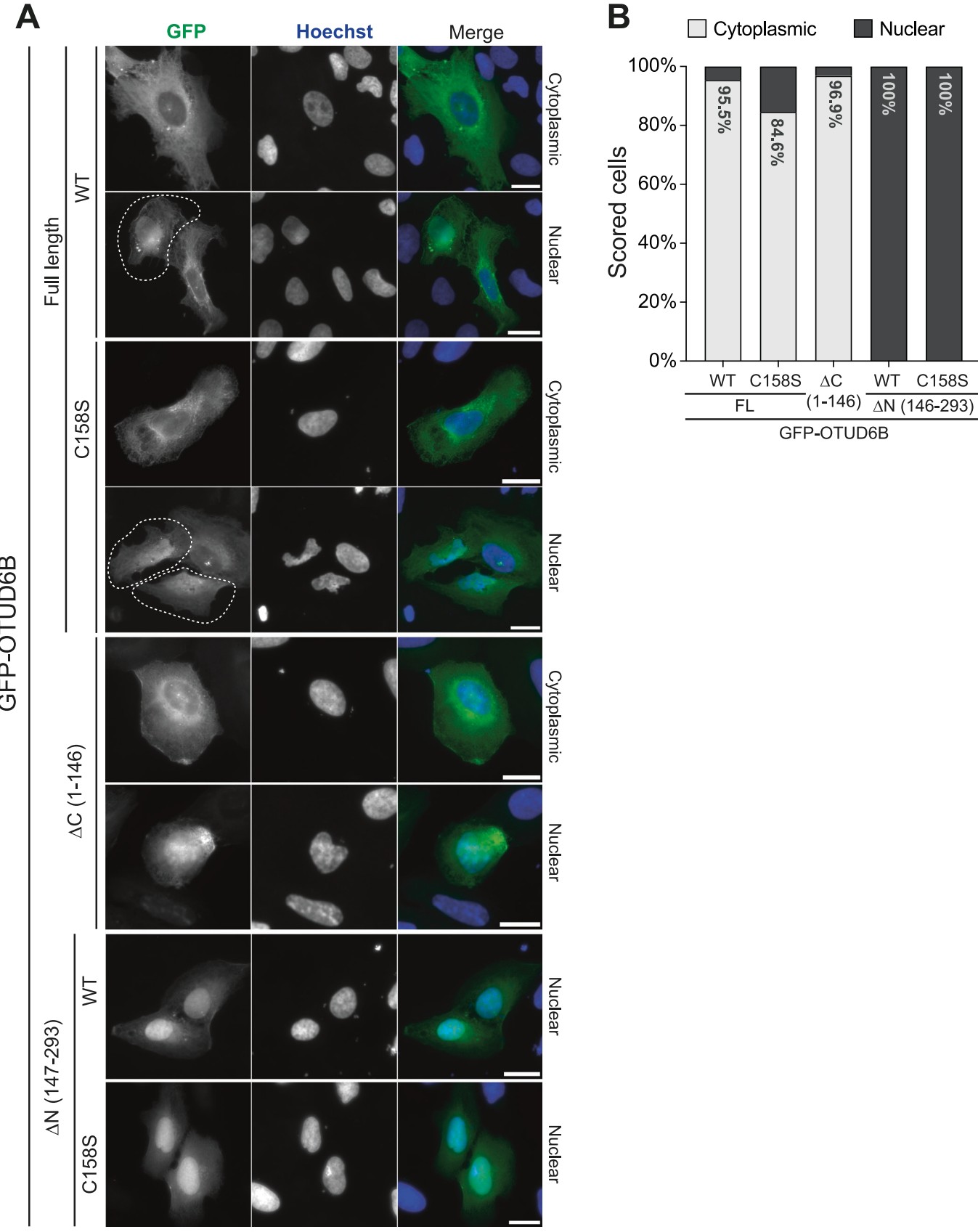

◄    **Figure EV4.   The N-terminus is required for OTUD6B nuclear exclusion.**

(**A**, **B**) U2OS cells were transfected with plasmids for 48 h then stained for KIFC1, pericentrin and DNA (Hoechst). Images were acquired with a Nikon Eclipse Ti fluorescent microscope. Representative images of cytoplasmic or nuclear distribution observed in cells transfected with each construct; scale bar 10 μm; dotted lines indicate cells scored as nuclear in the field shown (**A**). >100 interphase cells were scored per condition from one experiment (**B**).

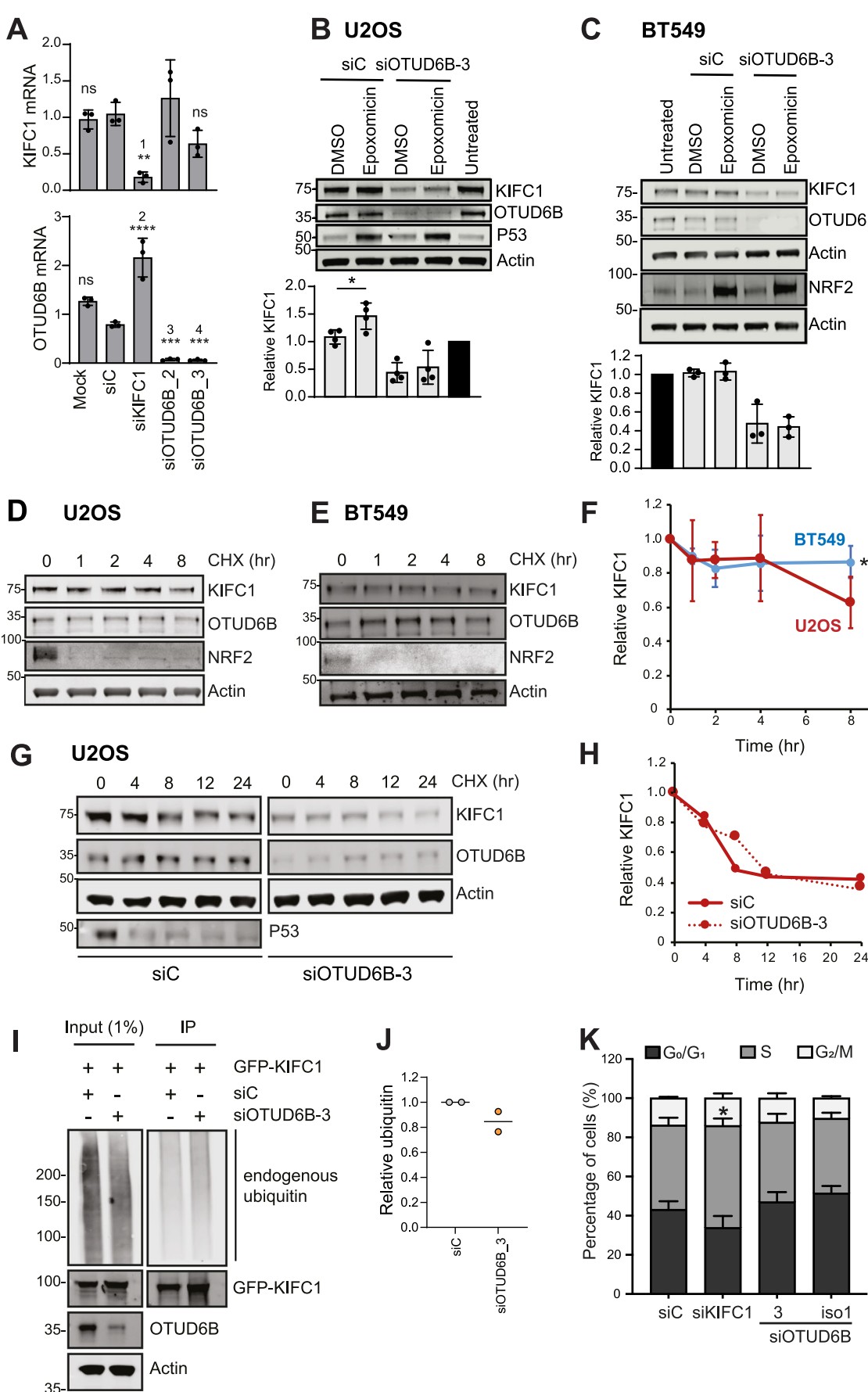

◀  **Figure EV5.   OTUD6B depletion does not affect KIFC1 transcription or global protein stability in asynchronous cells.**

(A) OTUD6B depletion does not significantly alter KIFC1 transcription. BT549 cells were transfected with 10 nM siRNA for 72 h. Expression by qRT-PCR normalised to ACTB and GAPDH, shown relative to mean of controls; mean for $n = 3$ biological replicates, error bars SD, [1]**$P = 0.0087$, [2]****$P \leq 0.0001$, [3]***$P = 0.0025$ and [4]***$P = 0.0022$, compared to siC by one-way ANOVA with Dunnett post-hoc test. (B, C) U2OS (B) or BT549 (C) cells were transfected with 10 nM siRNA for 72 h and treated with 50 nM epoxomicin or DMSO for the final 6 h. Representative immunoblot (top) and mean values for KIFC1 expression normalised to actin and shown relative to the untreated sample for $n = 4$ (B) or $n = 3$ (C) biological replicates (below); error bars SD, *$P = 0.0314$, compared to DMSO by two-tailed $t$ test. (D–F) Comparison of KIFC1 and OTUD6B half-lives in U2OS (D) or BT549 (E) cell lines treated with 10 µg/ml cycloheximide (CHX). Mean expression relative to 0 h for $n = 3$ biological replicates (F), error bars SD, *$P = 0.0416$, one-tailed t-test. (G-H) OTUD6B depletion does not reduce KIFC1 stability in asynchronous cells. U2OS cells were transfected with siRNA for 72 h, prior to CHX addition and immunoblotting (G), expression relative to 0 h, one experiment (H). (I, J) OTUD6B depletion does not increase KIFC1 ubiquitylation in asynchronous cells. U2OS cells were transfected with siRNAs for 72 h, and GFP-KIFC1 for 24 h then treated with 50n M epoxomicin for the final 6 h before immunoprecipitation with GFP nanobeads, $n = 2$ biological replicates. Representative immunoblot (I) and quantification of ubiquitin smear normalised to the total amount of GFP-KIFC1 pulled down (J). (K) KIFC1 but not OTUD6B depletion increases the proportion of cells in S-phase. MDA-MB-231 cells were transfected with 10 nM siRNAs for 72 h, stained with 7-AAD and analysed by flow cytometry; error bars SD of mean for $n = 3$ biological replicates; *$P = 0.0459$ compared to siC by one-way ANOVA with Dunnett post-hoc test.

