## [Peer Review File · EMBO Reports]

OTUD6B regulates KIFC1-dependent centrosome clustering and breast cancer cell survival

Valeria Marotta, Dorota Sabat-Pośpiech, Andrew Fielding, Amy Ponsford, Amanda Thomaz, Francesca Querques, Mark Morgan, Ian Prior, and Judy Coulson

Corresponding authors Judy Coulson (j.coulson@liverpool.ac.uk) , Andrew Fielding (a.fielding1@lancaster.ac.uk)

Review Timeline:

Submission Date:	30th Dec 23
Editorial Decision:	2nd Feb 24
Revision Received:	13th Oct 24
Editorial Decision:	8th Nov 24
Revision Received:	11th Dec 24
Accepted:	16th Dec 24

Editor: Achim Breiling

Transaction Report:

Dear Prof. Coulson,

Thank you for the transfer of your research manuscript to EMBO reports. I have now received the reports from the three referees that were asked to evaluate your study, which can be found at the end of this email.

As you will see, the referees think that the findings are of high interest. However, they have several comments, concerns, and suggestions, indicating that a major revision of the manuscript is necessary to allow publication of the study in EMBO reports. As the reports are below, and all the referee concerns need to be addressed, I will not detail them here.

Given the constructive referee comments, I would like to invite you to revise your manuscript with the understanding that all referee concerns must be addressed in the revised manuscript or in a detailed point-by-point response. Acceptance of your manuscript will depend on a positive outcome of a second round of review. It is EMBO reports policy to allow a single round of revision only and acceptance of the manuscript will therefore depend on the completeness of your responses included in the next, final version of the manuscript.

- 1) a .docx formatted version of the final manuscript text (including legends for main figures, EV figures and tables), but without the figures included. Figure legends should be compiled at the end of the manuscript text.
- 2) individual production quality figure files as .eps, .tif, .jpg (one file per figure), of main figures (up to 8) and EV figures (up to 5). Please upload these as separate, individual files upon re-submission.

For more details, please refer to our guide to authors:
<http://www.embopress.org/page/journal/14693178/authorguide#manuscriptpreparation>

Please consult our guide for figure preparation:
http://wol-prod-cdn.literatumonline.com/pb-assets/embo-site/EMBOPress_Figure_Guidelines_061115-1561436025777.pdf

See also the guidelines for figure legend preparation:
<https://www.embopress.org/page/journal/14693178/authorguide#figureformat>

4) a complete author checklist, which you can download from our author guidelines (<https://www.embopress.org/page/journal/14693178/authorguide>). Please insert page numbers in the checklist to indicate where the requested information can be found in the manuscript. The completed author checklist will also be part of the RPF.

Please also follow our guidelines for the use of living organisms, and the respective reporting guidelines:
<http://www.embopress.org/page/journal/14693178/authorguide#livingorganisms>

5) that primary datasets produced in this study (e.g. RNA-seq, ChIP-seq, structural and array data) are deposited in an

appropriate public database. If no primary datasets have been deposited, please also state this in a dedicated section (e.g. 'No primary datasets have been generated and deposited'), see below.

The accession numbers and database should be listed in a formal "Data Availability" section (placed after Materials & Methods) that follows the model below. This is now mandatory (like the COI statement). Please note that the Data Availability Section is restricted to new primary data that are part of this study. This section is mandatory. As indicated above, if no primary datasets have been deposited, please state this in this section

Data availability

8) Regarding data quantification and statistics, please make sure that the number "n" for how many independent experiments were performed, their nature (biological versus technical replicates), the bars and error bars (e.g. SEM, SD) and the test used to calculate p-values is indicated in the respective figure legends (also for potential EV figures and all those in the final Appendix). Please also check that all the p-values are explained in the legend, and that these fit to those shown in the figure. Please provide statistical testing where applicable. Please avoid the phrase 'independent experiment', but clearly state if these were biological or technical replicates. Please also indicate (e.g. with n.s.) if testing was performed, but the differences are not significant. In case n=2, please show the data as separate datapoints without error bars and statistics. See also: <http://www.embopress.org/page/journal/14693178/authorguide#statisticalanalysis>

9) Please add scale bars of similar style and thickness to microscopic images, using clearly visible black or white bars (depending on the background). Please place these in the lower right corner of the images themselves. Please do not write on or near the bars in the image but define the size in the respective figure legend.

10) Please also note our reference format:

12) We now use CRediT to specify the contributions of each author in the journal submission system. CRediT replaces the author contribution section. Please use the free text box to provide more detailed descriptions and do not provide your final manuscript text file with an author contributions section. See also our guide to authors: <https://www.embopress.org/page/journal/14693178/authorguide#authorshipguidelines>

13) We would encourage you to use 'Structured Methods', our new Materials and Methods format. According to this format, the

Materials and Methods section should include a Reagents and Tools Table (listing key reagents, experimental models, software and relevant equipment and including their sources and relevant identifiers), uploaded as separate file, followed by a Methods and Protocols section in which we encourage the authors to describe their methods using a step-by-step protocol format with bullet points, to facilitate the adoption of the methodologies across labs. More information on how to adhere to this format as well as downloadable templates (.doc or .xls) for the Reagents and Tools Table can be found in our author guidelines (section 'Structured Methods'):

14) Please order the manuscript sections like this, using these names:

Title page - Abstract - Keywords - Introduction - Results - Discussion - Materials and Methods - Data availability section - Acknowledgements - Disclosure and Competing Interests Statement - References - Figure legends - Expanded View Figure legends

I look forward to seeing a revised version of your manuscript when it is ready. Please let me know if you have questions or comments regarding the revision.

Yours sincerely,

Referee #1:

In the manuscript entitled: "OTUD6B regulates KIFC1-dependent centrosome clustering and breast cancer cell survival", Marotta et al. identified OTUD6B as a positive regulator of KIFC1 expression that is required for centrosome clustering in triple-negative breast cancer (TNBC) cells with amplified centrosomes. Cancer cells with amplified centrosomes heavily rely on a process called as centrosome clustering, which ensures bipolar cell division. Thus, novel therapeutic strategies interfering with centrosome clustering have the potential to selectively kill aggressive cancer cells with amplified centrosomes. In the current study, Marotta et al. identified OTUD6B from an siRNA screen of deubiquitinase (DUB) enzymes as a positive regulator of centrosome clustering and KIFC1 protein expression. OTUD6B depletion increased multipolar divisions and reduced cell viability as well as colony formation ability of cancer cells with amplified centrosomes, while normal cells were unaffected. Mechanistically, the authors showed that OTUD6B interacts with KIFC1 at the centrosomes and inhibits KIFC1 ubiquitination, thus, causing an increase in protein levels, while it has no effect on the mRNA expression of KIFC1. Furthermore, clinical relevance has also been tested and a higher OTUD6B expression in basal-like cancers, and a positive correlation between OTUD6B and KIFC1 protein levels were observed. Although this study identified a potentially interesting interactor of KIFC1 in centrosome amplified cancers, there are major problems in some of the experimental settings and in the conclusions being drawn based on the available data. Furthermore, the mitosis-specific activity of OTUD6B on KIFC1 is not supported by sufficient experimental data. The therapeutic benefit of OTUD6B targeting compared to other available options for centrosome amplified cancers is also not clearly stated. Overall, the following major and minor concerns outlined below are needed to be addressed.

Major Points:

1-In Fig. EV1C, the types of alterations need to be separately drawn in terms of their effects on patient survival. In the current form, patients with potentially opposing types of alterations (e.g., truncating mutations and amplifications) are included within the same group. Furthermore, other available patient datasets (e.g., METABRIC) should be analyzed to support the clinical relevance of the findings.

2-In Fig. 3E, there seems to be no overexpression of KIFC1 upon overexpressing the WT OTUD6B. What could be the reason for that?

3-In Fig. 3E, the binding of catalytically inactive OTUD6B to KIFC1 is stronger compared to WT OTUD6B. However, in Fig. 4C, FL CL158S construct only very weakly interacts with KIFC1. What is the reason for this discrepancy?

4-Although it was claimed that the activity of OTUD6B against KIFC1 is specific to mitosis, in Fig. 5B, it looks like the KIFC1 ubiquitination increases upon OTUD6B depletion, not only in M phase but also in the G2/M phase. This experiment should also include the G1-synchronized cells.

5-Related to Fig. 5C, it was mentioned that overexpressing the WT but not catalytic dead OTUD6B decreases KIFC1 polyubiquitination. However, the blots do not reflect this.

6-The proteasome-dependent degradation and protein stability testing shown in Fig. EV5 should be performed in mitotic vs. interphase-synchronized cells.

7-The quantification graph in S7B does not reflect what is shown in S7A. Also, what happens to the mitotic, centrosomal levels of KIFC1 upon OTUD6B depletion? Since the interaction between OTUD6B and KIFC1 was hypothesized to be mitosis-specific, showing the immunofluorescence signal from interphase cells is somewhat confusing.

8-A rescue experiment by overexpressing KIFC1 in the absence of OTUD6B in the mitotic- and interphase-synchronized cells and analyzing centrosome clustering and cell survival would be needed to support the hypothesis that KIFC1 is the major downstream of OTUD6B in mitotic cells.

9-The effects of depletion of OTUD6B on viability or colony formation is much less effective than KIFC1 depletion. However, the authors claim that OTUD6B could be a potentially better target for cells with centrosome amplification. Given also the moderate downregulation of KIFC1, targeting OTUD6B may not be a better option. Can authors comment on this?

10-Most experiments compare de novo centrosome amplified TNBC cell line with a non-TNBC cell line (MCF7) for the role of OTUD6B in centrosome clustering. Therefore, more controlled experiments with centrosome amplification induction using pharmacological or genomic methods are needed.

Minor Points:

1-In the Abstract, the mechanistic relation between OTUD6B and KIFC1 is not very clearly stated. It is better not to use vague terms, such as 'regulate', at least in the Abstract.

2-In Fig. 5A, the labels are missing (i.e., siC and siOTUD6B).

3-There might be a problem with the data shown in Fig. 6E in terms of labeling. It seems that OTUD6B depletion reduces multipolar mitosis while its overexpression increases it, which contradicts with what is being proposed.

4-The y-axis of Figure 2C lowest panel may be incorrect.

Referee #2:

In this manuscript Marotta et al. carry out a DUB targeted siRNA screen in a triple negative breast cancer cell line, scoring both the percentage of multipolar mitosis and the KIFC1 protein expression. KIFC1/HSET is long known as a factor promoting the clustering of amplified centrosomes and is thus required for bipolar spindle formation in the presence of an abnormally high number of centrosomes. As supernumerary centrosomes are often present in tumors, it is evident that reducing the cellular propensity to form a bipolar spindle and thus promoting catastrophic outcomes of cell division is an attractive mean to target tumor cell fitness specifically. The screen elegantly identifies two DUBs, one of which, OTUD6B, is followed up in greater detail.

While the findings reported here are interesting and novel, however, several shortcomings prevent publication without adequately addressing the following points:

Major points:

The point covered in Figure EV3 (the impact of OTU6DB knockdown on centrosome/centriole number) is important and has not been sufficiently addressed. Firstly, what has been carried out in EV3A should not be limited to hTERT-HME1 as this cell line shows no phenotype upon knockdown of the protein. Centrioles should also be counted in interphase cells of a line displaying multipolar spindles during mitosis. If the authors' hypothesis holds true, the cells should say the same penetrance of centrosome amplification +/- OTU6DB kd. Moreover, representative pictures should be shown.

Many of the microscopy pictures in Figures 3A, 3B, and 4D are not of high quality. Appreciating the differences described in the accompanying text is hard when looking at the figures. If the related phenomena are not striking, they might be quantified to allow for solid conclusions to be drawn.

At the end of page 9, the use of U2OS cells and how they are discussed is disturbing. The rationale of using this cell line is not evident, and results obtained in this line are contrasted with results obtained in breast cancer lines with centrosome amplification. Reading this section, the authors seem to credit the idea that different results can be justified by the presence/absence of amplified centrosomes, but this conclusion cannot be drawn. U2OS cells have nothing in common with the other lines covered in the manuscript, not even the tissue of origin.

The chosen strategy to synchronize cells in Figure 5 is not optimal. Thymidine-arrested cells are released in the presence of an Eg5 inhibitor. The purity of the mitotic component in those assays is not clear (figures define this data point as G2/M, which suggests a mixture; the text, on the other hand, refers to prometaphase arrest, which is not shown anywhere). Later mitotic stages are even more problematic as they are obtained by overriding the mitotic arrest promoted by the Eg5 inhibitor by pharmacologic inhibition of Aurora kinase B. Why not simply use a reversible mitotic blocker (e.g., low nocodazole), shake off the cells, and then release them in the fresh medium? This would warrant greater purity of the mitotic sample and less artefactual mitotic exit.

Data in 5B and C should be quantified as differences are not striking.

Figure 6: while dealing with a critical point of the paper, figure 6 is very confusing. Panel E shows the opposite of what one would expect. Moreover, panel B is very uninformative, and the accompanying quantification is unclear (the only clear thing is the asterisks). As for panels A and D, it is not clear to this reviewer what is the purpose of transfecting cells both with a plasmid and a siRNA targeting the mRNA expressed by the plasmid. Less data should be shown possibly reflecting multiple replicates

and with the accompanying quantification.

Minor points:

The upper part of the graphical abstract is not only superfluous but also misleading: aberrantly high centrosomes should be already present in interphase and not only in mitosis.

In page 4 of the introduction, it is stated that centrosome clustering is superfluous in non-cancer tissues. This is not always true, for instance, for hepatocytes.

How many times was the immunoblot in Figure 1E repeated? The quantification should also make the experimental variability evident (SD or SEM).

Page 6-7: "Overall, OTUD6B emerged as the lead candidate DUB that may regulate KIFC1-dependent centrosome clustering, as the same two OTUD6B siRNAs markedly induced both phenotypes (Fig 2E)." It is not clear why OTUD6B should be the lead candidate at this stage, as JOSD2 is excluded from the validation only at a later stage.

hTERT-HME are often referred to as "transformed". Shouldn't they be referred to as "immortalized"?

Figure 4C: what is the band appearing in the KIFC1 immunoblot upon overexpression of OTUD6B-deltaC? Monoubiquitylation? It seems plausible that the presence of this OTUD6B fragment hinders the activity of endogenous OTUD6B on KIFC1.

Figure 5A, the immunoblot is not accompanied by the information of which siRNA was applied. Moreover, Asynch cells are missing on the immunoblots on the right.

Referee #3:

Marotta, Sabat-Pośpiech and Fielding et al., report that the deubiquitinase (DUB) OTUD6B positively regulates KIFC1 in triple-negative breast cancer (TNBC) cells with amplified centrosomes. The authors first carried out an siRNA screen to identify DUBs required for KIFC1-dependent centrosome clustering in TNBC cell line BT549, and OTUD6B emerged as a candidate for further investigation. By immunolabelling, the authors suggested that exogenous OTUD6B localizes to centrosomes and colocalizes with KIFC1 during mitosis, and by immunoprecipitation, OTUD6B interacts with KIFC1. The authors present evidence suggesting OTUD6B regulates KIFC1 deubiquitylation during mitosis, and catalytic activity of OTUD6B is required for this process. Lastly, by cell survival and clonogenic assays, the authors show that the depletion of KIFC1 or OTUD6B decreased colony formation in TNBC cells lines, but not in immortalized breast epithelial cells line. Based on these observations, the authors proposed a mechanism by which OTUD6B antagonizes APC/C-Cdh1 and deubiquitinates KIFC1 to promote centrosome clustering and limit multipolar spindle formation in triple-negative breast cancer (TNBC) cells with amplified centrosomes.

Overall assessment:

While this work reports a substantial amount of work on a topic of interest there are many gaps in the study and the major conclusions are not well supported. Four OTUD6B siRNAs inconsistently reduce KIFC1 expression levels without convincing OTUD6B rescue validation, raising concerns about potential off-target effects of siRNAs. Additional inconsistencies are observed in assays with similar experimental settings, adding another layer of concern about the reliability of the results. The sole reliance on siRNAs is a major concern given that systematic genome-wide CRISPR screens (e.g. DepMap) in the cell lines studied here did not highlight OTUD6B as a dependency in TNBC cells with amplified centrosomes.

In terms of the proposed mechanism where OTUD6B antagonizes APC/C-Cdh1 and deubiquitinates KIFC1 to promote centrosome clustering and limit multipolar spindle formation in TNBC cells with amplified centrosomes, there are also concerns. KIFC1 is required starting early in mitosis to organize the pseudo-bipolar spindle with amplified centrosomes while the proposed antagonism between OTUD6B and APC/C-Cdh1 would operate very late in mitosis, after anaphase onset when APC/C-Cdh1 is activated (APC/C-Cdh1 then remains active until the G1-S transition but is not active during mitosis). The disconnect between when APC/C-Cdh1 is active versus when KIFC1 is needed to cluster multiple centrosomes raises questions as to the validity of the proposed antagonism.

Collectively, these major concerns limit enthusiasm for the study, despite the important topic, and make it not suitable for publication in its current form.

Major points:

1) In Figure EV2 A-G, the authors tested four OTUD6B siRNAs, and all four efficiently depleted OTUD6B. However, KIFC1 expression upon OTUD6B depletion varied among siRNAs and cell lines - only OTUD6B depletion by siRNA#3 consistently reduced KIFC1 expression. The authors attempted a rescue experiment show in Figure 6A, but: (1) the KIFC levels in rescued sample (lane 5) seems unchanged compared to the siOTUD6B sample (lane 4); (2) the KIFC level in siOTUD6B sample (lane

4) does not show a significant reduction compared to siC sample (lane 1) as is shown in Figure EV2F. These data are insufficiently robust to support the central conclusion made by the authors. The authors must validate the KIFC1 reduction phenotype by generating OTUD6B knockout in TNBC cell lines, where the effect on spindle assembly is also observed, instead of U2OS cells. Given that OTUD6B knockouts in systematic CRISPR/Cas9 screens do not reduce cell fitness (see point 2 below), and the relative technical ease of generating CRISPR/Cas9-based knockouts, the work needs to include generation and analysis of OTUD6B knockouts in the relevant TNBC cell lines. If this approach revealed increased multipolar spindle formation and reduced KIFC1 protein levels following OTUD6B knockout, then both of these phenotypes must be rescued by expression of OTUD6B from a transgene at near-endogenous levels.

2) According to the CRISPR/Cas9 screening data collated in the Cancer Dependency Map (DepMap), KIFC1 is required for proliferation of breast cancer cell lines with amplified centrosomes (e.g., BT549 and MDA-MB-231) while OTUD6B is dispensable, which is in contrast to the synthetic lethal mechanism between amplified centrosomes and loss of OTUD6B proposed in this manuscript. While systematic screens may include false negatives, in the collective experience of many research groups the DepMap data is robust and there are numerous examples of true cancer dependencies that have emerged from this dataset. This discrepancy further enhances the importance of the authors employing CRISPR/Cas9 knockout approaches to support their major conclusions.

3) The spindle phenotypes reported in Figure 6E are the opposite of the claim "... overexpression of siOTUD6B-3 resistant wild type GFP-OTUD6B significantly increased the number of bipolar spindles in BT549 cells, whilst the catalytic dead C158S mutant could not rescue this phenotype". Potentially the order of the labels is wrong in this figure. Assuming the order is wrong, the percentage of multipolar BT549 cells in siC group (~40%, top bar in Figure 6E) is ~2 times higher than that of the siC group in Figure EV2J (~20%). These types of differences across experiments are concerning and need to be given greater attention by the authors.

4) For Co-IP experiments shown in Figure 3E and 4C, endogenous KIFC1 exhibited differences in binding to exogenous GFP-OTUD6B-C158S when compared to wild type GFP-OTUD6B - once again the reason for such differences is unclear.

5) If the authors are rigorously able to support the claim that OTUD6B acts by reducing KIFC1 levels, e.g. by generating, analyzing, and rescuing OTUD6B knockouts in TNBC cells, then an important prediction is that overexpression of KIFC1 should rescue OTUD6B loss-of-function. Such an experiment would provide support for their major claim provided that the conclusions that OTUD6B protects KIFC1 from ubiquitin-mediated degradation and this in turn makes cells with amplified centrosomes sensitive to OTUD6B loss are first rigorously established.

Minor comments:

1) In Figure EV3, the authors evaluated centrosome amplification in siOTUD6B hTERT-HME1 (normal breast epithelium cell line), and showed OTUD6B depletion did not induce centrosome amplification. However, the baseline centrosome amplification frequency in this assay is two times higher than that in Figure EV1A which is not mentioned. It is also important that the authors could perform the centrosome counting assay in TNBC cell lines most relevant to their work (MDA-MB-231 and BT549) and report the centrosome amplification frequency.

2) In Figure 3A and 3B, the authors showed that overexpressed GFP-OTUD6B localizes to centrosomes in U2OS cells. Is this subcellular localization also observed in TNBC cell lines (MDA-MB-231 and BT549)? Is there any insight into how OTUD6B is achieving this localization through analysis of truncations, etc.

3) The authors need to more thoroughly introduce OTUD6B, including information on domains in its primary structure, current knowledge about its functions, etc. in the manuscript text.

EMBO Reports – response to reviewers**Referee #1:**

In the manuscript entitled: "OTUD6B regulates KIFC1-dependent centrosome clustering and breast cancer cell survival", Marotta et al. identified OTUD6B as a positive regulator of KIFC1 expression that is required for centrosome clustering in triple-negative breast cancer (TNBC) cells with amplified centrosomes. Cancer cells with amplified centrosomes heavily rely on a process called as centrosome clustering, which ensures bipolar cell division. Thus, novel therapeutic strategies interfering with centrosome clustering have the potential to selectively kill aggressive cancer cells with amplified centrosomes. In the current study, Marotta et al. identified OTUD6B from an siRNA screen of deubiquitinase (DUB) enzymes as a positive regulator of centrosome clustering and KIFC1 protein expression. OTUD6B depletion increased multipolar divisions and reduced cell viability as well as colony formation ability of cancer cells with amplified centrosomes, while normal cells were unaffected. Mechanistically, the authors showed that OTUD6B interacts with KIFC1 at the centrosomes and inhibits KIFC1 ubiquitination, thus, causing an increase in protein levels, while it has no effect on the mRNA expression of KIFC1. Furthermore, clinical relevance has also been tested and a higher OTUD6B expression in basal-like cancers, and a positive correlation between OTUD6B and KIFC1 protein levels were observed. Although this study identified a potentially interesting interactor of KIFC1 in centrosome amplified cancers, there are major problems in some of the experimental settings and in the conclusions being drawn based on the available data. Furthermore, the mitosis-specific activity of OTUD6B on KIFC1 is not supported by sufficient experimental data. The therapeutic benefit of OTUD6B targeting compared to other available options for centrosome amplified cancers is also not clearly stated. Overall, the following major and minor concerns outlined below are needed to be addressed.

Thank you for recognising the potential interest of our study, and for your constructive suggestions to improve it further. We have addressed your comments, as described below, and believe this has considerably strengthened our study and better evidenced the conclusions.

Major Points:

1-In Fig. EV1C, the types of alterations need to be separately drawn in terms of their effects on patient survival. In the current form, patients with potentially opposing types of alterations (e.g., truncating mutations and amplifications) are included within the same group. Furthermore, other available patient datasets (e.g., METABRIC) should be analyzed to support the clinical relevance of the findings.

Thank you for this constructive comment. Although OTUD6B alterations are predominantly gene amplification or mRNA overexpression (Fig S3A), you are correct that a few other potentially opposing alterations were included in the original analysis. We have now extracted the TCGA data and stratified this based on high OTUD6B mRNA expression relative to diploid samples, which is seen in 28% of cases (updated Fig S3B). We focused on mRNA alteration, which is more directly relevant to potential function than gene amplification. The revised analysis is now shown in new Fig EV1C and demonstrates that high OTUD6B mRNA expression is significantly associated with worse overall survival ($P < 0.005$).

We have also looked at the METABRIC dataset as you suggested. We see OTUD6B gene amplification in 20% of cases (Figure 1A for reviewers), and a significant association with survival ($P < 0.0005$, Figure 1B for reviewers). However, the mRNA data from the METABRIC dataset appear less robust, with OTUD6B mRNA overexpression reported in only 4% of METABRIC samples (Figure 1C for reviewers). This is likely due to a combination of technical factors, firstly it is generated by DNA microarray rather

than RNAseq, and secondly data are normalised to all samples rather than to diploid samples, so the high frequency of OTUD6B gene amplification can obscure the mRNA overexpression. Given the sample size bias between altered/unaltered groups for high OTUD6B mRNA in METABRIC, it is not possible to predict survival outcomes with confidence, and we have therefore not been able to include these data in the revised manuscript.

Figure for reviewers removed

2-In Fig. 3E, there seems to be no overexpression of KIFC1 upon overexpressing the WT OTUD6B. What could be the reason for that?

Please note that in response to your comment 3 below, and major point 4 made by reviewer 3, we have repeated this experiment to confirm the relative ability of GFP-OTUD6B WT and CS to pulldown endogenous KIFC1. The original Fig 3E has now been replaced by new Fig 5C, where we see the same result for the input samples. These experiments are performed on a background of endogenous OTUD6B in asynchronous cells, and this observation suggests that there is already

sufficient endogenous OTUD6B to sustain KIFC1 levels under these conditions. We have now noted this in the revised manuscript on page 10/11.

3-In Fig. 3E, the binding of catalytically inactive OTUD6B to KIFC1 is stronger compared to WT OTUD6B. However, in Fig. 4C, FL CL158S construct only very weakly interacts with KIFC1. What is the reason for this discrepancy?

We are unsure of the reason for this discrepancy between the experiments in original Fig 3E and original Fig 4C. We have now repeated the experiment a third time to clarify the result. In this experiment, we saw very similar amounts of KIFC1 pulled down by GFP-OTUD6B WT and CS. We follow standardised laboratory protocols, but the experiments were performed by different researchers at different times, potentially introducing some technical variation. As the new experiment is most representative, generating an intermediate result, it is now shown in new Fig 5C, replacing the original figure panels.

4-Although it was claimed that the activity of OTUD6B against KIFC1 is specific to mitosis, in Fig. 5B, it looks like the KIFC1 ubiquitination increases upon OTUD6B depletion, not only in M phase but also in the G2/M phase. This experiment should also include the G1-synchronized cells.

Thank you for raising this point, we agree that OTUD6B depletion also effects KIFC1 ubiquitination early in mitosis. Indeed, in the original manuscript we said "importantly, the level of KIFC1 poly-ubiquitylation increased in OTUD6B depleted cells arrested at prometaphase and this persisted 30 to 45 minutes into pseudo-mitotic exit in the absence of proteasome inhibition (Fig 5B, Appendix Fig S6E)" - please note these figures are now Fig 6B and Appendix Fig S7E. We believe that the confusion arose due to the labelling on original Fig 5B (now Fig 6B) and apologise for this, the G2/M label was misleading, as the cells are arrested in prometaphase. This finding for the timing of KIFC1 ubiquitination during mitosis is consistent with additional data we have now included in the revised manuscript to address major point 6 that you raised, examining the dynamics of KIFC1 stability during mitosis (shown in new Fig 6F and new Fig 6G). These data confirm that OTUD6B depletion is already affecting KIFC1 stability in prometaphase, as described below. This observation is consistent with OTUD6B promoting KIFC1 centrosome clustering activity early in mitosis to form a pseudo-bipolar spindle, along with an ongoing requirement to maintain KIFC1 levels and prevent declustering during anaphase. These points are now discussed in more detail on pages 12 and 13 of the revised manuscript.

Unfortunately, we are not able to perform the ubiquitination experiment with cells synchronised in G1 as you suggested, as KIFC1 is not present as cells exit mitosis and enter G1 (new Fig S7F-G) and levels remain too low for this type of analysis during G1. However, we do already show the same experiment performed in asynchronous cells in Fig EV5I and have now included quantification of the ubiquitin smear normalised to the GFP-KIFC1 pulldown for this experiment (n=2, new Fig EV5J). Comparison with these asynchronous cell data shows that the effect of OTUD6B depletion on KIFC1 ubiquitination is enriched during early and late mitosis (Fig 6B) and we now highlight this on page 12 of the revised manuscript.

5-Related to Fig. 5C, it was mentioned that overexpressing the WT but not catalytic dead OTUD6B decreases KIFC1 polyubiquitination. However, the blots do not reflect this.

We apologise if the reviewer was not able to appreciate the differences in KIFC1 poly-ubiquitin between the last 3 lanes of the IP blot in original Fig 5C, which we believe show the effect of WT OTUD6B (centre lane) in reducing this level, compared to either no exogenous OTUD6B (lane to the left) or the catalytic dead OTUD6B (lane to the right). In the revised manuscript this is now Fig 6C, and we have improved the contrast and labelled the area of the ubiquitin smear to make this effect

easier to appreciate. We have also quantified the amount of poly-ubiquitinated KIFC1 relative to the total GFP-KIFC1 pulled down in each lane, as suggested by reviewer 2 (major point 5), and indicated these numbers beneath the blot. This shows that OTUD6B WT but not CS could reduce the KIFC1 ubiquitin smear by ~50% during mitotic exit, even at the time when the APC/C, as highly processive E3, is most active.

6-The proteasome-dependent degradation and protein stability testing shown in Fig. EV5 should be performed in mitotic vs. interphase-synchronized cells.

Performing these types of biochemical experiments in mitotic cells is difficult as, for example, proteasome inhibition will block mitotic exit that is inherently dependent upon protein degradation. We therefore attempted to address your suggestion by holding MDA-MB-231 cells +/- OTUD6B depletion in prometaphase with STLC whilst inhibiting the proteasome for 6 hours (Figure 2 for reviewers). As described above, we see effects on KIFC1 polyubiquitination (Fig 6B) at this point in mitosis. OTUD6B depletion was effective, with reduced KIFC1 levels evident in prometaphase arrested cells (higher Cyclin B and lower Cyclin A2 than asynchronous cells). However, proteasome inhibition under these conditions had no effect on levels of either KIFC1 or the positive control Cyclin A2. Due to the technical limitations of these biochemical endpoint assays, we instead addressed the question through an alternative experiment.

Figure for reviewers removed

We used live cell imaging of Venus-tagged KIFC1 (as reported in Min et al, 2014) to assess how KIFC1 degradation progresses in OTUD6B depleted compared to control cells during mitosis (methods page 23). The KIFC1-Venus-Dmut, where the N-terminal D-box that is recognised by the APC/C is mutated (R5A, L8A), did not degrade over the experimental timecourse (new Fig S7F – S7G), and this was unaffected by OTUD6B depletion (new Fig 6F and 6G). In contrast, degradation of KIFC1-Venus WT began around anaphase onset in control cells. However, in OTUD6B depleted cells, degradation of KIFC1-Venus WT was evident from prophase, prior to anaphase onset (new Fig 6F and 6G). These data are consistent with the ubiquitination results (Fig 6B) and suggest that OTUD6B could help maintain KIFC1 stability between prophase and anaphase. Although OTUD6B depletion results in significantly lower levels of KIFC1 until 16-24 minutes into anaphase, the results suggest that OTUD6B is less able to oppose the APC/C once the spindle assembly checkpoint is satisfied. Thus, OTUD6B may counteract the APC/C early in mitosis and so, in the absence of OTUD6B, this positions KIFC1 as a relatively early APC/C substrate, more akin to cyclin A and Nek2, which can be degraded prior to satisfying the spindle assembly checkpoint. These data and implications are now discussed in more detail on page 13 of the revised manuscript. We hope that together these complementary data address your general concern that mitosis-specific activity of OTUD6B on KIFC1 was not supported by sufficient experimental data in the original version of the manuscript.

7-The quantification graph in S7B does not reflect what is shown in S7A. Also, what happens to the mitotic, centrosomal levels of KIFC1 upon OTUD6B depletion? Since the interaction between OTUD6B and KIFC1 was hypothesized to be mitosis-specific, showing the immunofluorescence signal from interphase cells is somewhat confusing.

We apologise that the presentation of data in original Fig S7A & S7B was confusing, and that these figure panels did not appear to be reflective of each other. OTUD6B depletion affects KIFC1 levels throughout the cell cycle (Fig 6A), perhaps because OTUD6B depleted cells struggle to recover KIFC1 levels after mitotic exit, as they arrest, or there is some ongoing role for OTUD6B. However, the dynamic effect at mitosis is most acute, and hence easiest to demonstrate biochemically. We have now rearranged the figure panels to improve the flow and to make the point more clearly in the text on page 9 that these experiments relate to the immunoblotting in interphase cells. Those panels that you queried are now Fig S5B & S5C with the same data represented on a log2 scale in the graph, which makes it much easier to appreciate the differences in KIFC1 mean pixel intensity (now Fig S5B).

Thank you for the additional suggestion to extend this question to mitosis, we have now also examined what happens to the mitotic, centrosomal levels of KIFC1. Indeed, OTUD6B depletion led to a significant reduction in centrosome-associated KIFC1 at mitotic spindles (new Fig S5F-S5G) described on page 10. Although loss of KIFC1 at centrosomes is not complete, this is consistent with biochemical assessment of OTUD6B depletion by immunoblotting.

8-A rescue experiment by overexpressing KIFC1 in the absence of OTUD6B in the mitotic- and interphase-synchronized cells and analyzing centrosome clustering and cell survival would be needed to support the hypothesis that KIFC1 is the major downstream of OTUD6B in mitotic cells.

Thank you for this suggestion that we agree is an important, if experimentally challenging, question. OTUD6B depletion is deleterious to cells, and we predict this is particularly relevant at mitosis if cancer cells cannot cluster super-numerary centrosomes. Therefore, following multiple transfections and synchronisation it can be hard to find surviving mitotic cells in rescue experiments. However, we were able to do this using an experimental configuration where OTUD6B was depleted in an MDA-MB-231 cell line with stably inducible KIFC1 and the cells examined by microscopy. We confirmed that OTUD6B depletion reduced KIFC1 in the population of uninduced cells, but that this was

overcome by acute induction of KIFC1 with doxycycline (new Fig 3H, please note the log₁₀ scale). Under the conditions of OTUD6B knockdown and KIFC1 overexpression, we also scored mitotic cells for the presence of multipolar spindles (new Fig 3I). As in our previous experiments OTUD6B depletion induced multipolar spindle formation, which we could now show was partially rescued by induction of KIFC1. These data, described on pages 9 & 10 of the revised manuscript, support our model where OTUD6B is acting via KIFC1 to modulate centrosome clustering.

We also looked at the effects on cell survival in this experiment, using nuclei per field of view as a proxy. We saw a significant reduction in nuclei on siOTUD6B treatment compared to siC for the uninduced cells, but this was not the case in cells where KIFC1 overexpression was induced by doxycycline (Figure 3 for reviewers). This is consistent with KIFC1 rescue of OTUD6B-induced effects on survival. However, we are cautious about this interpretation, as nuclei number was lower in both conditions following doxycycline treatment, therefore we have chosen not to include these data in the revised manuscript.

Figure for reviewers removed

9-The effects of depletion of OTUD6B on viability or colony formation is much less effective than KIFC1 depletion. However, the authors claim that OTUD6B could be a potentially better target for

cells with centrosome amplification. Given also the moderate downregulation of KIFC1, targeting OTUD6B may not be a better option. Can authors comment on this?

We did not intend to claim that OTUD6B is a better target than KIFC1, but rather that it could provide an alternative target for the same cancer-associated phenotype. Indeed, on page 18, we said “identifying a DUB for KIFC1 might present an alternative way to therapeutically target centrosome clustering”. We then speculated that “As kinesins are relatively difficult to target pharmacologically, and complete inhibition of KIFC1 may have non-cancer cell specific effects (Xiao & Yang, 2016), targeting OTUD6B has the potential to be developed as a more tolerable therapeutic route”. This relates to the issues highlighted in the introduction on page 5 that for KIFC1 inhibitors “their pharmacological specificity for cancer over normal cells remains limited (Patel *et al*, 2018; Wu *et al*, 2013; Yukawa *et al*, 2018) and next-generation KIFC1 inhibitors or novel ways to target centrosome-clustering are needed.” Unfortunately, there are currently no therapeutically relevant KIFC1 inhibitors, whereas DUB enzymes are very tractable therapeutic targets, with specific DUB inhibitors being developed and showing promise as cancer therapeutics, as we also mentioned on page 5 “Specific inhibitors are already available for DUBs including USP7 (Turnbull *et al*, 2017) and, most notably USP1, with KSQ-4279 in phase 1 clinical trials (<https://www.clinicaltrials.gov/study/NCT05240898>).” We therefore believe we put forward a considered argument for OTUD6B as a potential alternative therapeutic target in cancer cells with centrosome amplification.

In addition to KIFC1, other proteins that play roles in centrosome clustering may of course also present druggable, cancer specific targets (Rhys & Godinho, 2017; Sabat-Pospiech *et al*, 2019). To give just two examples, blocking Hsp72 or Nek6 function induces multipolar spindles and reduces cancer cell growth (Sampson *et al*, 2017) whilst a large-scale screen identified an inhibitor of STAT3 that causes centrosome declustering and inhibits viability in cancer cells and *in vivo* (Morris *et al*, 2017). To our knowledge inhibiting these, or any other centrosome clustering proteins, therapeutically is not yet an approach that is being tested in clinical trials. Therefore, exploring novel modulators of KIFC1 such as DUBs, remains an important area of research. We now highlight these examples of other potential options for centrosome amplified cancers on page 18 of the revised manuscript.

In the context of our data, we agree that the knockdown of OTUD6B had a milder viability phenotype than knockdown of KIFC1. This may reflect several properties of these two proteins and does not necessarily imply that OTUD6B would be a worse therapeutic target for small molecule inhibition. For example, the half-life of KIFC1 is only 9.4 hours (and we know that it is acutely degraded during every mitosis), whereas the half-life of OTUD6B is 71 hrs (Schwanhausser *et al*, 2011). For siRNA depletion this will mean that the effect on levels of OTUD6B is slower than it is for KIFC1, which would not be the case for approaches that blocked OTUD6B catalytic activity. The outcome of our attempts to knockout *OTUD6B* by CRISPR/Cas9 in BT549 lends additional weight to the suggestion that it is legitimate target, as over time reduced OTUD6B levels/activity are lethal to cancer cells through a phenotype indicative of mitotic catastrophe (see page 14, Fig 7I-J and response to reviewer 3, major point 1). We now provide this clearer context for our discussion on page 18 of the revised manuscript.

10-Most experiments compare de novo centrosome amplified TNBC cell line with a non-TNBC cell line (MCF7) for the role of OTUD6B in centrosome clustering. Therefore, more controlled experiments with centrosome amplification induction using pharmacological or genomic methods are needed.

You are mistaken, we have not made any functional comparisons to MCF7. Our aim is ultimately to understand how OTUD6B functions in this role in cancer cells that naturally display centrosome

amplification, against heterogenous genetic backgrounds, to be able to translate our findings. Hence, we have confirmed our key experiments in two TNBC cell lines with different degrees of centrosome amplification: BT549 (high CA) and MDA-MB-231 (moderate CA) that both rely on centrosome clustering to survive. We have now clarified this on pages 7 and 13/14 in the revised manuscript. Some experiments, e.g. the clonogenic assays used hTERT-HME1 cells as a comparator; the rationale here is that they provide a model of non-transformed breast cells, which would be present in a clinical setting. We do not believe that demonstrating OTUD6B has an impact on centrosome clustering in cells with artificially generated centrosome amplification would significantly add to our study. Whilst potentially an interesting approach, we feel it is outside the scope of this manuscript and have not addressed it experimentally due to limited capacity in the lab, and the need to prioritise the most relevant experiments. However, to address point 1 raised by reviewer 3, we have now attempted to generate CRISPR knockouts of *OTUD6B* in TNBC cells (new Fig 7I-7J, page 14). Whilst this does not directly address your point, it has demonstrated in a genomic model that OTUD6B is essential for survival of cells with natural centrosome amplification that struggle to complete mitosis (Appendix Videos S3-6).

Minor Points:

1-In the Abstract, the mechanistic relation between OTUD6B and KIFC1 is not very clearly stated. It is better not to use vague terms, such as 'regulate', at least in the Abstract.

Thank you for this suggestion, the abstract has been modified to reflect the new data we have incorporated and reworded, changes are highlighted in the revised manuscript. For example, "In OTUD6B-deficient cells, we see **increased** KIFC1 poly-ubiquitination and premature KIFC1 degradation during mitosis."

2-In Fig. 5A, the labels are missing (i.e., siC and siOTUD6B).

We apologise for this omission; these labels have now been added (now Fig 6A).

3-There might be a problem with the data shown in Fig. 6E in terms of labeling. It seems that OTUD6B depletion reduces multipolar mitosis while its overexpression increases it, which contradicts with what is being proposed.

We apologise for this error; the labels have now been corrected (now Fig 3G).

4-The y-axis of Figure 2C lowest panel may be incorrect.

The y-axis in Fig 2C lower panel is correct, with the data for Relative Protein expression shown normalised to the mean of the two controls (mock and siC) as in other figures. In Fig 2D lower panel, no good JOSD2 antibody was available, so in this case we determined depletion of JOSD2 by qRT-PCR and the y-axis is Relative mRNA. I hope that this answers your question.

Interestingly, in Fig 2C lower panel KIFC1 depletion (lane 3) induces OTUD6B expression in BT549 cells, and we also see the same response in MDA-MB-231 cells (original Fig S7C, now Fig S8D). We speculate on page 7 of the revised manuscript that this may reflect a feedback mechanism where cells with less KIFC1 upregulate OTUD6B to attempt to restore KIFC1 levels by protein stabilisation.

Referee #2:

In this manuscript Marotta et al. carry out a DUB targeted siRNA screen in a triple negative breast cancer cell line, scoring both the percentage of multipolar mitosis and the KIFC1 protein expression. KIFC1/HSET is long known as a factor promoting the clustering of amplified centrosomes and is thus required for bipolar spindle formation in the presence of an abnormally high number of centrosomes. As supernumerary centrosomes are often present in tumors, it is evident that reducing the cellular propensity to form a bipolar spindle and thus promoting catastrophic outcomes of cell division is an attractive mean to target tumor cell fitness specifically. The screen elegantly identifies two DUBs, one of which, OTUD6B, is followed up in greater detail.

While the findings reported here are interesting and novel, however, several shortcomings prevent publication without adequately addressing the following points:

Thank you for your positive comments about the elegance of the screens as well as the interesting and novel nature of our study. Thank you also for your constructive suggestions to improve the manuscript further. We have addressed all your comments, as described below, and believe this has considerably strengthened our study.

Major points:

1. The point covered in Figure EV3 (the impact of OTU6DB knockdown on centrosome/centriole number) is important and has not been sufficiently addressed. Firstly, what has been carried out in EV3A should not be limited to hTERT-HME1 as this cell line shows no phenotype upon knockdown of the protein. Centrioles should also be counted in interphase cells of a line displaying multipolar spindles during mitosis. If the authors' hypothesis holds true, the cells should say the same penetrance of centrosome amplification +/- OTU6DB kd. Moreover, representative pictures should be shown.

Thank you for these suggestions, which we have incorporated into the revised manuscript. New Fig EV3B shows scoring of centrosome amplification in response to OTUD6B depletion in mitotic MDA-MB-231 cells. The two siRNAs that most efficiently reduce KIFC1 expression (siOTUD6B-3 and siOTUD6B-iso1) do not increase centrosome amplification in these TNBC cells. New Fig EV3D and EV3E show example images and scoring of both centrosomes and centrioles in interphase BT549 cells with pre-existing centrosome amplification, which display multipolar spindles during mitosis as already shown in original Fig EV3B (now Fig EV3C). Most interphase cells have 1 centrosome and 2 centrioles, as expected for cells in G1. The remainder have either 2 centrosomes and 2 centrioles, representing a mixed population of normal cells in G2 and centrosome amplified cells in G1, or more than 2 centrosomes and more than 4 centrioles, representing centrosome amplification irrespective of interphase stage. Depletion of KIFC1 or OTUD6B does not significantly alter the distribution of cells between these groups. Importantly, OTUD6B depletion also does not induce centriole splitting, as pairs of centrioles were observed in centrosomes of all conditions, and in cells of different centrosome amplification status (both for interphase cells in Fig EV3E and mitotic cells in Fig EV3C). These data are described on page 9 of the revised manuscript.

2. Many of the microscopy pictures in Figures 3A, 3B, and 4D are not of high quality. Appreciating the differences described in the accompanying text is hard when looking at the figures. If the related phenomena are not striking, they might be quantified to allow for solid conclusions to be drawn.

Thank you for making this suggestion regarding quantification to aid in interpretation of the microscopy images. We have now quantified the co-localisation in original Figs 3A, 3B, and 4D (now Figs 4A, 4B and 5D) for OTUD6B co-localisation with pericentrin during interphase and prometaphase, and for OTUD6B co-localisation with KIFC1 during prometaphase. These data confirm significant enrichment of co-localisation for GFP-KIFC1, compared to GFP alone, with pericentrin throughout the cell cycle and with KIFC1 in prophase (new Fig 4C). They also confirm that the catalytic dead CS and C-terminal truncated GFP-OTUD6B mutants retain co-localisation with pericentrin, but that this is lost for the N-terminal mutants (new Fig 5E). The methodology used is described on page 23 of the revised manuscript.

We have also included some additional data in the supplementary figures as new Fig S6B, which shows clear co-localisation of GFP-KIFC1 with dTOMATO-OTUD6B on the nascent mitotic spindle. This is described on page 10 of the revised manuscript.

3. At the end of page 9, the use of U2OS cells and how they are discussed is disturbing. The rationale of using this cell line is not evident, and results obtained in this line are contrasted with results obtained in breast cancer lines with centrosome amplification. Reading this section, the authors seem to credit the idea that different results can be justified by the presence/absence of amplified centrosomes, but this conclusion cannot be drawn. U2OS cells have nothing in common with the other lines covered in the manuscript, not even the tissue of origin.

We apologise if the way this section was written did not clearly convey our message, we did not mean to imply there was a causal effect of centrosome amplification on KIFC1 stability, but mentioned the centrosome status of BT549 only as a reminder that these are the cells where KIFC1 levels are important to understand. We chose to use U2OS cells for mechanistic experiments, as they are often used in the literature for biochemistry experiments, because they are easy to transfect and amenable to synchronisation, both of which were more difficult in TNBC cells. Most importantly, U2OS were the cell line originally selected to demonstrate KIFC1 ubiquitination by the APC/C as they are highly synchronisable (Min et al, 2014). In the revised manuscript, we have justified the use of U2OS cells more clearly on pages 8 & 12 and have revised the text you refer to at the bottom of page 11.

4. The chosen strategy to synchronize cells in Figure 5 is not optimal. Thymidine-arrested cells are released in the presence of an Eg5 inhibitor. The purity of the mitotic component in those assays is not clear (figures define this data point as G2/M, which suggests a mixture; the text, on the other hand, refers to prometaphase arrest, which is not shown anywhere). Later mitotic stages are even more problematic as they are obtained by overriding the mitotic arrest promoted by the Eg5 inhibitor by pharmacologic inhibition of Aurora kinase B. Why not simply use a reversible mitotic blocker (e.g., low nocodazole), shake off the cells, and then release them in the fresh medium? This would warrant greater purity of the mitotic sample and less artefactual mitotic exit.

We apologise for the incorrect G2/M labelling on original Fig 5 (now Fig 6), this has now been changed in the figure and the description of the method. The cells are indeed arrested in prometaphase as we describe in the text, as shown below (Figure 4 for reviewers), and are then collected by mitotic shake-off prior to STLC washout and release with ZM to increase purity of the mitotic component, as described in the methods section (page 21/22).

We selected this synchronisation and release protocol to directly replicate that previously used to demonstrate KIFC1 ubiquitination by the APC/C in U2OS cells. Min et al (2014) reasoned that highly synchronized cell populations would be required to identify substrates of the UPS specific to mitotic exit. They “chose a ZM-release protocol over a protocol for the release of SAC-arrested cells using mitotic cyclin-dependent kinase (Cdk1) inhibitor treatment, as described elsewhere because the anaphase degradation rate of APC/C substrates that we have tested is reduced when Cdk1 is inhibited (but not when Aurora B is inhibited), suggesting that the timing of Cdk1 inactivation during anaphase is critical for optimal targeting of substrates at this time.” Min et al (2014) rigorously validated both the specificity of pulldowns and robustness of their synchronization protocol, e.g. ubiquitinated cyclin B1 was recovered only from cells in which the SAC had been inactivated and unmodified cyclin B1 was undetectable in the pulldown fractions, even when loaded in massive excess. We were therefore confident that this was a very robust protocol to monitor mitotic KIFC1 ubiquitination in U2OS cells. In our hands, we found that the washout worked much better for their protocol, than it did from nocodazole arrested cells, to give synchronous mitotic progression. We have now more clearly stated our rationale for selecting this method for the ubiquitination experiments on page 12 in the revised manuscript. Please note, that we did not use this synchronisation method for the mitotic cell imaging experiments, e.g. new Fig 6F and 6G.

Figure for reviewers removed

5. Data in 5B and C should be quantified as differences are not striking.

Thank you for this helpful suggestion, we have now quantified the amount of poly-ubiquitinated KIFC1 (Flag-ub signal) relative to the GFP-KIFC1 pulled down in each lane and indicated these numbers beneath the blot in original Fig 5B and 5C (now Fig 6B and 6C). The quantification shows that OTUD6B depletion could increase the ubiquitin smear for KIFC1 by 2.5- to 3-fold (Fig 6B) and that OTUD6B WT but not CS could reduce the KIFC1 ubiquitin smear by ~50% during mitotic exit, at the time when the APC/C is most active (Fig 6C). We have also improved the contrast for the presented images to make these differences easier to appreciate visually.

6. Figure 6: while dealing with a critical point of the paper, figure 6 is very confusing. Panel E shows the opposite of what one would expect. Moreover, panel B is very uninformative, and the accompanying quantification is unclear (the only clear thing is the asterisks). As for panels A and D, it

is not clear to this reviewer what is the purpose of transfecting cells both with a plasmid and a siRNA targeting the mRNA expressed by the plasmid. Less data should be shown possibly reflecting multiple replicates and with the accompanying quantification.

We have moved this rescue figure to improve the flow of the revised manuscript and have addressed your comments with revisions to what is now Fig 3, and to its description in the figure legend and text on page 9. We hope that these adjustments and improved explanation (see also response to reviewer 3, major point 1, paragraph 2) have resolved any unintended confusion we caused.

We had included the additional siRNAs (siOTUD6B-iso1 in panel A, siOTUD6B-2 in panel D) that can target exogenous OTUD6B as additional controls to contrast with siOTUD6B-3, which cannot target exogenous OTUD6B. We have now simplified original Fig 6A (now Fig 3A) to remove the siOTUD6B-iso1 data and original Fig 6D (now Fig 3F) to remove the siOTUD6B-2 data, as you suggest. The immunoblot in new Fig 3A is a representative example from three independent experiments, and we have now moved the graphs showing quantification for those experiments from original Fig S7D-S7E so that these are now included as new Fig 3B and new Fig 3C.

We are sorry that you found panel B uninformative. Now Fig 3D this shows example cells transfected with OTUD6B WT where KIFC1 levels are preserved, or OTUD6B CS where they are not. We have now included arrows to indicate these cells and improved the description of the figure on page 9 of the revised manuscript.

We apologise for the error in original Fig 6E; these labels have now been corrected (now Fig 3G).

Minor points:

1. The upper part of the graphical abstract is not only superfluous but also misleading: aberrantly high centrosomes should be already present in interphase and not only in mitosis.

We apologise that the upper part of the graphical abstract appeared misleading. It was not intended to infer that aberrant centrosome number was unique to mitosis, rather to illustrate the normal centrosome cycle in the context of the cell cycle, whilst summarising the KIFC1 and OTUD6B levels through the cell cycle. However, we absolutely agree that this is superfluous and hard to understand. We have now simplified the graphical abstract as you suggest, by removing the upper section.

2. In page 4 of the introduction, it is stated that centrosome clustering is superfluous in non-cancer tissues. This is not always true, for instance, for hepatocytes.

Thank you for picking up this omission, which is now corrected on page 4 of the revised manuscript.

3. How many times was the immunoblot in Figure 1E repeated? The quantification should also make the experimental variability evident (SD or SEM).

The immunoblot in Fig 1E was only performed once, hence no error bars are included. It was undertaken as an intermediary step to eliminate any non-reproducible candidates from the full screen (Fig 1D) before following up the most promising candidates (n=3) in Fig 2 and Fig S2. We have now made this clearer in the description of Fig 1E in the figure legend and page 6 of the revised manuscript.

4. Page 6-7: "Overall, OTUD6B emerged as the lead candidate DUB that may regulate KIFC1-dependent centrosome clustering, as the same two OTUD6B siRNAs markedly induced both

phenotypes (Fig 2E)." It is not clear why OTUD6B should be the lead candidate at this stage, as JOSD2 is excluded from the validation only at a later stage.

Our reasoning was that for JOSD2 although two individual siRNAs affected each phenotype, only one siRNA was common to both phenotypes. This is now more clearly explained in the revised manuscript on page 7. "Overall, OTUD6B emerged as the lead candidate DUB that may regulate KIFC1-dependent centrosome clustering, as the same two OTUD6B siRNAs markedly induced both phenotypes (Fig 2A, 2C & 2E). **In contrast, for JOSD1 whilst siJOSD2-3 significantly altered both readouts, siJOSD2-2 induced multipolar spindles (Fig 2B), whereas siJOSD2-4 reduced KIFC1 expression (Fig 2D).**"

In addition, as described in response to reviewer 1 minor point 4, in Fig 2C KIFC1 depletion induces OTUD6B expression in BT549 cells, and we also see the same response in MDA-MB-231 cells (now Fig S8D). We speculate on page 7 of the revised manuscript that this may reflect a feedback mechanism where cells with less KIFC1 upregulate OTUD6B to attempt to restore KIFC1 levels by protein stabilisation.

5. hTERT-HME are often referred to as "transformed". Shouldn't they be referred to as "immortalized"?

We apologise for this oversight, which has now been corrected in the revised manuscript.

6. Figure 4C: what is the band appearing in the KIFC1 immunoblot upon overexpression of OTUD6B-deltaC? Monoubiquitylation? It seems plausible that the presence of this OTUD6B fragment hinders the activity of endogenous OTUD6B on KIFC1.

Yes, we agree that this was a potentially interesting observation, however when we repeated this experiment (new Fig 5C), the additional band was not evident. For this reason, we have not investigated further.

7. Figure 5A, the immunoblot is not accompanied by the information of which siRNA was applied. Moreover, Asynch cells are missing on the immunoblots on the right.

We apologise for this omission; these labels have now been added (now Fig 6A). Although we did not have an asynchronous sample for siOTUD6B in the experiment originally shown, we have now included the blot from a different experiment that did include that sample (please note, n=2 quantification for this sample; n=3 for the other samples).

Referee #3:

Marotta, Sabat-Pośpiech and Fielding et al., report that the deubiquitinase (DUB) OTUD6B positively regulates KIFC1 in triple-negative breast cancer (TNBC) cells with amplified centrosomes. The authors first carried out an siRNA screen to identify DUBs required for KIFC1-dependent centrosome clustering in TNBC cell line BT549, and OTUD6B emerged as a candidate for further investigation. By immunolabelling, the authors suggested that exogenous OTUD6B localizes to centrosomes and colocalizes with KIFC1 during mitosis, and by immunoprecipitation, OTUD6B interacts with KIFC1. The

authors present evidence suggesting OTUD6B regulates KIFC1 deubiquitylation during mitosis, and catalytic activity of OTUD6B is required for this process. Lastly, by cell survival and clonogenic assays, the authors show that the depletion of KIFC1 or OTUD6B decreased colony formation in TNBC cells lines, but not in immortalized breast epithelial cells line. Based on these observations, the authors proposed a mechanism by which OTUD6B antagonizes APC/C-Cdh1 and deubiquitinates KIFC1 to promote centrosome clustering and limit multipolar spindle formation in triple-negative breast cancer (TNBC) cells with amplified centrosomes.

Overall assessment:

While this work reports a substantial amount of work on a topic of interest there are many gaps in the study and the major conclusions are not well supported. Four OTUD6B siRNAs inconsistently reduce KIFC1 expression levels without convincing OTUD6B rescue validation, raising concerns about potential off-target effects of siRNAs. Additional inconsistencies are observed in assays with similar experimental settings, adding another layer of concern about the reliability of the results. The sole reliance on siRNAs is a major concern given that systematic genome-wide CRISPR screens (e.g. DepMap) in the cell lines studied here did not highlight OTUD6B as a dependency in TNBC cells with amplified centrosomes.

In terms of the proposed mechanism where OTUD6B antagonizes APC/C-Cdh1 and deubiquitinates KIFC1 to promote centrosome clustering and limit multipolar spindle formation in TNBC cells with amplified centrosomes, there are also concerns. KIFC1 is required starting early in mitosis to organize the pseudo-bipolar spindle with amplified centrosomes while the proposed antagonism between OTUD6B and APC/C-Cdh1 would operate very late in mitosis, after anaphase onset when APC/C-Cdh1 is activated (APC/C-Cdh1 then remains active until the G1-S transition but is not active during mitosis). The disconnect between when APC/C-Cdh1 is active versus when KIFC1 is needed to cluster multiple centrosomes raises questions as to the validity of the proposed antagonism.

Collectively, these major concerns limit enthusiasm for the study, despite the important topic, and make it not suitable for publication in its current form.

Thank you for recognising that our study represents a substantial amount of work, and that the topic is of interest, as well as for your constructive suggestions to improve it further. We have addressed all your comments, as described below, and believe our new data have considerably strengthened our study and should answer your concerns.

With reference to the specific point you raise above about mechanism, our mitotic declustering phenotype, mitotic ubiquitination data, and new data we include in the revised manuscript on the timing of KIFC1 degradation during mitosis (Fig 6) are all consistent with OTUD6B promoting KIFC1 centrosome clustering activity early in mitosis to form a bipolar spindle before anaphase. Please also see responses to reviewer 1 major points 4 and 6 above. We appreciate that this early effect of OTUD6B on KIFC1 is before the main phase of APC/C-CDH1 activity would be relevant. However, there is also a requirement to maintain KIFC1 levels to prevent declustering until anaphase is complete. As such, we would expect OTUD6B activity towards KIFC1 to begin early in mitosis but continue into mitotic exit, when KIFC1 is rapidly degraded as an established target of the APC/C-CDH1. We believe the activity of OTUD6B towards KIFC1 decorated with K11-linked ubiquitin is also persuasive that this is antagonism of the APC/C. We had already covered e.g. the potential for earlier APC/C activity, ahead of the spindle assembly checkpoint, either towards early substrates, or because of mitotic slippage in cells with amplified centrosomes, in the original manuscript discussion (page 16/17). We have now integrated our new findings and further clarified our interpretation and

model on pages 12, 13, 16 and 17, which we hope you will agree has improved the revised manuscript.

Major points:

1) In Figure EV2 A-G, the authors tested four OTUD6B siRNAs, and all four efficiently depleted OTUD6B. However, KIFC1 expression upon OTUD6B depletion varied among siRNAs and cell lines - only OTUD6B depletion by siRNA#3 consistently reduced KIFC1 expression. The authors attempted a rescue experiment show in Figure 6A, but: (1) the KIFC levels in rescued sample (lane 5) seems unchanged compared to the siOTUD6B sample (lane 4); (2) the KIFC level in siOTUD6B sample (lane 4) does not show a significant reduction compared to siC sample (lane 1) as is shown in Figure EV2F. These data are insufficiently robust to support the central conclusion made by the authors. The authors must validate the KIFC1 reduction phenotype by generating OTUD6B knockout in TNBC cell lines, where the effect on spindle assembly is also observed, instead of U2OS cells. Given that *OTUD6B* knockouts in systematic CRISPR/Cas9 screens do not reduce cell fitness (see point 2 below), and the relative technical ease of generating CRISPR/Cas9-based knockouts, the work needs to include generation and analysis of OTUD6B knockouts in the relevant TNBC cell lines. If this approach revealed increased multipolar spindle formation and reduced KIFC1 protein levels following OTUD6B knockout, then both of these phenotypes must be rescued by expression of OTUD6B from a transgene at near-endogenous levels.

Thank you for highlighting the first point, we agree that the data as originally presented in Fig EV2 did not accurately represent the reproducibility that we see for the effects of OTUD6B siRNAs on KIFC1 levels. The original experiments had been performed by different researchers at different times, and crucially did not show all the siRNAs for each of the cell lines. We have now repeated these experiments with four siRNAs for all the four cell lines that we use in the paper and these data are shown as new Figs EV2D–EV2G. Both siOTUD6B-3 (from the original siRNA library pool) and siOTUD6B-iso1 (which we designed to be specific for the transcript variant encoding the major OTUD6B isoform-1) both effectively deplete OTUD6B and consistently reduce KIFC1 protein levels in all 4 cell lines. Although, we see some context-specific differences in the degree of effect on KIFC1 for the other two siRNAs (siOTUD6B-1 and siOTUD6B-2) between cell lines, each of these siRNAs replicate the effect of siOTUD6B-3 and siOTUD6B-iso1 in at least two cell lines. Although we cannot mechanistically explain the higher variability with these two siRNAs, efficient RNA interference with specific sequences depends on many variables, including the global context of the target sequence (e.g. conformation, abundance (Hong *et al*, 2014; Pancoska *et al*, 2004), and interplay with other regulatory mechanisms for the protein of interest, combinations of which may vary between cell lines. Importantly, we see a correlation between the efficiency of OTUD6B depletion and the reduction in KIFC1 levels in both the TNBC cell lines BT549 (Fig EV2B-EV2D) and MDA-MB-231 (Fig EV2E). Overall, we are confident that data generated using siRNAs in our study are robust, reproducible and support our conclusions, and hope that this is now clearer for the readers. Together with evidence that both OTUD6B and KIFC1 overexpression can rescue OTUD6B depletion phenotypes, and that attempts to knockout *OTUD6B* using CRISPR also generate consistent phenotypes (as described below), we hope you will now agree that our data support our central conclusions.

We apologise if the reviewer was not convinced by our rescue experiment for KIFC1 levels, and we have now included additional quantification and explanation (page 9) in the revised manuscript to facilitate interpretation of this experiment. The immunoblot in original Fig 6A (now Fig 3A) is primarily intended to show that the siRNA depletion of endogenous OTUD6B was effective, and to

demonstrate overexpression of the non-targetable form of dTOMATO-OTUD6B following transient transfection. Given the efficiency limitations of transient plasmid transfection, the primary readout for the KIFC1 rescue is not the immunoblot, but quantification by microscopy of KIFC1 levels in individual dTOMATO-transfected cells, shown in Fig 6B–6C (now Fig 3D–3E). Despite the technical limitation of immunoblotting analysis of total KIFC1 levels across a heterogeneous population of cells, some of which do and some of which do not overexpress dTOMATO-OTUD6B, we believe the overall effect of the rescue is discernible on the immunoblot, although more moderate when assessed by this method compared to single cell analysis, as one might expect. We have now moved the quantification of the corresponding immunoblots (original Fig S7D–S7E), for the three independent experiments used for the cell scoring, to include these histograms as new Fig 3B and 3C. They show that KIFC1 levels in lane 4 (siOTUD6B-3, dTOMATO) are significantly reduced relative to lane 1 (sic, dTOMATO). They also show that in lane 6 (siOTUD6B-3, dTOMATO-OTUD6B-CS) there is no evidence of rescue by the catalytic dead mutant as levels remain significantly different to lane 1, whilst in lane 5 (siOTUD6B-3, dTOMATO-OTUD6B-WT) there is no significant difference compared to the KIFC1 levels in lane 1, indicative of rescue.

As you requested, we have also attempted to validate the KIFC1 reduction phenotype by generating *OTUD6B* knockouts in a TNBC cell line that has centrosome amplification and generates multipolar spindles. We had not attempted this previously, as our *OTUD6B* depletion experiments indicated that, despite the DepMap data you refer to (please also see response to point 2 below), TNBC cells with reduced levels of *OTUD6B* do not thrive (Fig 7A-B, 7E-H). We therefore initially validated our methods (page 21) and CRISPR guides by attempting to knockout *OTUD6B* in a non-relevant cell line. We could not do this in hTERT-HME cells as they already carry puromycin resistance used to select for CRISPR, so chose to use HEK-293T as an amenable workhorse cell line that lacks centrosome amplification (Bose *et al*, 2021). We generated several homozygous *OTUD6B*^{-/-} HEK293T clones (new Fig S8C), which demonstrated no growth defects compared the parental *OTUD6B*^{+/+} cells (new Fig S8D, Appendix Videos S1-S2), despite lacking *OTUD6B* expression and having reduced KIFC1 expression (new Fig S8E).

We then attempted the same CRISPR/Cas9 protocol in the BT549 TNBC cells. However, despite two attempts, we were only able to generate a few heterozygous *OTUD6B*^{+/-} knockouts but no homozygous knockouts in BT549 cells (new Fig 7I). Single clones were monitored for 4 weeks post-transfection, but no colonies were observed. Although individual cells were visible in some wells, these remained extremely sparse and did not appear healthy and a lot of cell death was also evident, suggesting other cells had divided but subsequently died (new Fig 7J). On closer examination by live cell imaging, there was evidence that the remaining cells were trying and failing to complete mitosis, as they rounded up on the plate, before flattening out again as a single cell, and in some cases subsequently dying (Appendix Videos S3-S6). This phenotype would be consistent with mitotic failure and catastrophe, that are common outcomes when cells with centrosome amplification fail to cluster their supernumerary centrosomes during mitosis. As we were unable to grow any of these clones, it was impossible to recover any for analysis of *OTUD6B*/KIFC1 expression, centrosome number or spindle phenotypes, we could only lyse the few remaining cells after imaging to confirm their genotype by PCR (new Fig 7I). These results are described on page 14 of the revised manuscript.

Overall, these outcomes provide additional support to our siRNA data (i) in the context of the requirement of *OTUD6B* to maintain KIFC1 levels, evidenced in TNBC and other cell lines with siRNA depletion of *OTUD6B* and also in *OTUD6B*^{-/-} HEK293T, and (ii) reinforcing the argument that *OTUD6B* is essential for long term survival of TNBC cells, and providing evidence that attempts to genetically knockout *OTUD6B* result in phenotypes consistent with the inability of cells to cluster supernumerary centrosomes.

2) According to the CRISPR/Cas9 screening data collated in the Cancer Dependency Map (DepMap), *KIFC1* is required for proliferation of breast cancer cell lines with amplified centrosomes (e.g., BT549 and MDA-MB-231) while *OTUD6B* is dispensable, which is in contrast to the synthetic lethal mechanism between amplified centrosomes and loss of *OTUD6B* proposed in this manuscript. While systematic screens may include false negatives, in the collective experience of many research groups the DepMap data is robust and there are numerous examples of true cancer dependencies that have emerged from this dataset. This discrepancy further enhances the importance of the authors employing CRISPR/Cas9 knockout approaches to support their major conclusions.

We agree with the reviewer that Depmap is a valuable resource for high throughput overview of gene essentiality profiles. However, as mentioned, systematic genome-wide CRISPR dropout screens may include false negatives. Indeed, a recent study shows the Depmap screen is non-saturating and estimates the false negative rate as approximately 20% (Dede & Hart, 2023). As described above, taking a targeted approach, and with knowledge of the biology, we have now shown that *OTUD6B* isoform 1 knockout is highly deleterious to BT549 cells (new Fig 7H-7I). We now briefly discuss this discrepancy on page 18 of the revised manuscript.

Depmap collates data from two CRISPR screens performed by the Sanger and Broad Institutes. Their methodology varies slightly, as do the cell lines covered. For example, sequencing for dropouts was done after 14 days (Sanger) or 21 days (Broad), and whilst MDA-MB-231 were included in both screens (with data compiled from both timepoints), BT549 were only included in the shorter Sanger screen. Related to these timelines, the half-life of *KIFC1* is 9.4 hours (and we know it degrades during every mitosis), whereas the half-life of *OTUD6B* is much longer at 71 hours (Schwanhausser *et al.*, 2011).

Plausible explanations for why *OTUD6B* was not picked up in the Depmap screens, and was apparently less essential for TNBC than *KIFC1*, may follow from this information, the biology of the system we are investigating, and our experimental CRISPR/Cas9 data.

(i) Even in BT549 (where centrosome amplification is highest so they may be expected to have a more severe phenotype) only 32% of cells within the population demonstrate centrosome amplification at any one time. Therefore, on *OTUD6B* knockout we would predict 68% of the population are initially impervious. Centrosome amplification is a dynamic process and will accumulate over time in these surviving cells (Lee *et al.*, 2014). As such, we predict that if fitness were tightly linked to the ability to cluster centrosomes, the population would only gradually become susceptible to knockout of *OTUD6B* (or indeed *KIFC1*) and die off. Therefore, the profile of dropout is likely much slower than would be expected for essential genes that affect 100% of the cell population from the start of the experiment.

(ii) BT549 (expected to have a severe phenotype) were sequenced after 2 weeks, whereas the phenotypes we recorded for *OTUD6B* CRISPR/Cas9 were still evident after 4 weeks. Thus, the timeline of experiments may contribute to *OTUD6B* being a false negative in Depmap.

(iii) Overlaid with these points, the longer half-life of *OTUD6B* protein (versus *KIFC1*) and the indirect effect (via *KIFC1*) on centrosome clustering, will delay the initial viability phenotype, even in the 32% of BT549 cells with centrosome amplification.

(iv) The essentiality of the *KIFC1* gene according to Depmap, may in part be conferred by its other cellular roles. If this were the case, a slower centrosome clustering-dependent effect could be masked.

(v) In line with these ideas, the effect of *OTUD6B* depletion on both cell proliferation and colony formation was more moderate compared to *KIFC1* depletion (Fig 7A-7H). Thus, *OTUD6B* may better classed as a fitness gene, with a cumulative effect on growth of a cell population, rather than an essential gene with widespread and rapid lethality.

(vi) Finally, the guides used in the Depmap screens were not specific for the major OTUD6B isoform. The literature suggests potentially opposing roles for an OTUD6B isoform (Sobol *et al*, 2017), which based on our data (Fig 5, Fig EV4) may not contain centrosome targeting or nuclear exclusion signals.

The emerging literature also support a role for OTUD6B in promoting cancer cell growth in other targeted studies. For example, in lung adenocarcinoma OTUD6B knockdown with shRNA inhibited proliferation of cells and xenografts (Yang *et al*, 2024). To date there is only one report of *OTUD6B* knockout in cancer cells, for a DUB-focused CRISPR/Cas9 screen of vulnerabilities in multiple myeloma that identified OTUD6B (Paulmann *et al*, 2022). Although these studies focus on other substrates/pathways for OTUD6B, many cancer types exhibit centrosome amplification (Sabat-Pospiech *et al.*, 2019). Overall, there is accumulating evidence for OTUD6B as a fitness gene, despite not being identified as essential in Depmap.

3) The spindle phenotypes reported in Figure 6E are the opposite of the claim "... overexpression of siOTUD6B-3 resistant wild type GFP-OTUD6B significantly increased the number of bipolar spindles in BT549 cells, whilst the catalytic dead C158S mutant could not rescue this phenotype". Potentially the order of the labels is wrong in this figure. Assuming the order is wrong, the percentage of multipolar BT549 cells in siC group (~40%, top bar in Figure 6E) is ~2 times higher than that of the siC group in Figure EV2J (~20%). These types of differences across experiments are concerning and need to be given greater attention by the authors.

We apologise for this error in original Fig 6E; the labels have now been corrected (now Fig 3G).

Regarding the difference in baseline centrosome spindle frequency in the experiments shown in original Fig 6E (now Fig 3G) and Fig EV2J, the experimental setting was different. In Fig EV2J none of the cell lines are transfected, so these reflect the true basal levels, i.e. the BT549 data are not for an siC group. In contrast, in original Fig 6E (now Fig 3G) all cells (including the siC group) have been transfected with siRNAs using RNAiMAX. Although RNAiMAX is generally regarded as a low toxicity option for efficient siRNA transfection, and the reasons for this were unclear, we consistently saw an increase in multipolar spindles in transiently transfected cells. We now mention this observation on page 9 of the revised manuscript. We do not believe that this detracts from the conclusions that we independently draw based on data in each of these figures, where conditions (transfection or not) are consistent between all samples that we directly compare within an experiment.

4) For Co-IP experiments shown in Figure 3E and 4C, endogenous KIFC1 exhibited differences in binding to exogenous GFP-OTUD6B-C158S when compared to wild type GFP-OTUD6B - once again the reason for such differences is unclear.

We are unsure of the reason for this discrepancy between the experiments in original Fig 3E and original Fig 4C. We have now repeated the experiment a third time to clarify the result. In this new experiment, we saw very similar amounts of KIFC1 pulled down by GFP-OTUD6B WT and CS. Although we follow standardised laboratory protocols, the experiments were performed by different researchers at different times, potentially introducing some technical variation. The new experiment is most representative, as it generates an intermediate result, and is now shown in new Fig 5C, replacing the original figure panels.

5) If the authors are rigorously able to support the claim that OTUD6B acts by reducing KIFC1 levels, e.g. by generating, analyzing, and rescuing OTUD6B knockouts in TNBC cells, then an important prediction is that overexpression of KIFC1 should rescue OTUD6B loss-of-function. Such an experiment would provide support for their major claim provided that the conclusions that OTUD6B protects KIFC1 from ubiquitin-mediated degradation and this in turn makes cells with amplified centrosomes sensitive to OTUD6B loss are first rigorously established.

As described in our response to point 1 above, we were unable to generate OTUD6B knockouts in TNBC cells, as this proved lethal. However, as requested by reviewer 1 (major point 8), we have performed this rescue experiment in a different experimental configuration to demonstrate that re-expressing KIFC1 can partially rescue OTUD6B dependent phenotypes. In summary, OTUD6B was deleted in an MDA-MB-231 cell line with stably inducible KIFC1 (methods page 20/21) and the cells examined by microscopy. We confirmed that OTUD6B depletion reduced KIFC1 in the population of uninduced cells, but that this was overcome by acute induction of KIFC1 with doxycycline (new Fig 3H, please note the log10 scale). Under the conditions of OTUD6B knockdown and KIFC1 overexpression, we also scored mitotic cells for the presence of multipolar spindles (new Fig 3I). As in our previous experiments OTUD6B depletion induced multipolar spindle formation, which we could show was partially rescued by induction of KIFC1. These data, described on pages 9 & 10 of the revised manuscript, support our model where OTUD6B is acting via KIFC to modulate centrosome clustering.

We also looked at the effects on cell survival in this experiment, using nuclei per field of view as a proxy. We saw a significant reduction in nuclei on siOTUD6B treatment compared to siC for the uninduced cells, but this was not the case in cells where KIFC1 overexpression was induced by doxycycline (Figure 3 for reviewers – reproduced again below for ease). This is consistent with KIFC1 rescue of OTUD6B-induced effects on survival, however we are cautious about this interpretation, as nuclei number was lower in both conditions following doxycycline treatment, therefore we have chosen not to include these data in the revised manuscript.

Figure for reviewers removed

Minor comments:

1) In Figure EV3, the authors evaluated centrosome amplification in siOTUD6B hTERT-HME1 (normal breast epithelium cell line), and showed OTUD6B depletion did not induce centrosome amplification. However, the baseline centrosome amplification frequency in this assay is two times higher than that in Figure EV1A which is not mentioned. It is also important that the authors could perform the centrosome counting assay in TNBC cell lines most relevant to their work (MDA-MB-231 and BT549) and report the centrosome amplification frequency.

As for point 3 that you raised above, regarding original Fig 6E, we believe this difference arises because of a difference in the way the cells in the two figures were treated before scoring centrosomes – in Fig EV1A the cells were untransfected, whereas in Fig EV3A cells have been transfected. Indeed, we can show that transfection of these cells hTERT-HME1 significantly increased centrosome number (Figure 5 for reviewers). This will however have no impact on our interpretation of the experiment shown in Fig EV3, as all the cells were treated in the same way so that data within the experiment are internally consistent.

Figure for reviewers removed

It was also requested that we show an experiment examining the effect of OTUD6B depletion on centrosome amplification in TNBC cells (reviewer 2, major point 1). We have addressed this in the revised manuscript and reproduce the response to reviewer 2 below for convenience.

Thank you for these suggestions, which we have incorporated into the revised manuscript. New Fig EV3B shows scoring of centrosome amplification in response to OTUD6B depletion in mitotic MDA-MB-231 cells. The two siRNAs that efficiently reduce KIFC1 expression (siOTUD6B-3 and siOTUD6B-iso1) do not increase centrosome amplification in these TNBC cells. New Fig EV3D and Fig3E show example images and scoring of both centrosomes and centrioles in interphase BT549 cells with pre-existing centrosome amplification, which display multipolar spindles during mitosis as already shown in Fig EV3C. Most interphase cells have 1 centrosome and 2 centrioles, as expected for cells in G1.

The remainder have either 2 centrosomes and 2 centrioles, representing a mixed population of normal cells in G2 and centrosome amplified cells in G1, or more than 2 centrosomes and more than 4 centrioles, representing centrosome amplification irrespective of interphase stage. Depletion of KIFC1 or OTUD6B does not significantly alter the distribution of cells between these groups. Importantly, OTUD6B depletion also does not induce centriole splitting, as pairs of centrioles were observed in centrosomes of all conditions, and in cells of different centrosome amplification status (both for interphase cells in Fig EV3E and mitotic cells in Fig EV3C). These data are described on page 8 and 9 of the revised manuscript.

2) In Figure 3A and 3B, the authors showed that overexpressed GFP-OTUD6B localizes to centrosomes in U2OS cells. Is this subcellular localization also observed in TNBC cell lines (MDA-MB-231 and BT549)? Is there any insight into how OTUD6B is achieving this localization through analysis of truncations, etc.

Thank you for this suggestion, which we have incorporated into the revised manuscript and is described on page 10. New Fig S6B shows localisation of GFP-OTUD6B or GFP alone in BT549 cells. GFP-OTUD6B co-localises with centrosomes in interphase cells. During mitosis, in cells with both bipolar (or pseudo-bipolar) and multipolar spindles, GFP-OTUD6B is evident colocalised with tubulin at the nascent spindle poles in the vicinity of the centrosomes.

We agree that the signal for this centrosomal location of OTUD6B would be very interesting to establish. So far, we know from the truncation mutants that the N-terminal of OTUD6B is required for co-localisation with pericentrin (Fig 5D and New Fig 5E) and this same region is required to keep OTUD6B out of the nucleus of interphase cells (Fig EV4). Interestingly, this N-terminal region is truncated in a shorter isoform in NSCLC that was proposed to have opposing properties to isoform 1 (Sobol *et al.*, 2017). Centrosome localisation signals have been identified in some substrates, such as Cyclin E and Cyclin A, however these appear to be substrate-specific, and no consensus sequence has been identified to allow bioinformatic prediction. Preliminary predictions for nuclear export signals (NES) do suggest though that the only three predicted motifs meeting the minimum stringency cutoff are all within the N terminal region. Whilst further investigation is outside the experimental scope of the current study, this will be interesting to consider in future, and we have added some brief discussion on page 17 of the revised manuscript.

3) The authors need to more thoroughly introduce OTUD6B, including information on domains in its primary structure, current knowledge about its functions, etc. in the manuscript text.

Thank you for this suggestion, we have now concisely introduced these basic aspects of OTUD6B biology earlier in the revised manuscript (page 7), which we already talk about in more detail in the description of results using OTUD6B constructs, and in the discussion.

References

- Bose A, Modi K, Dey S, Dalvi S, Nadkarni P, Sudarshan M, Kundu TK, Venkatraman P, Dalal SN (2021) 14-3-3gamma prevents centrosome duplication by inhibiting NPM1 function. *Genes Cells* 26: 426-446
- Curtis C, Shah SP, Chin SF, Turashvili G, Rueda OM, Dunning MJ, Speed D, Lynch AG, Samarajiwa S, Yuan Y *et al* (2012) The genomic and transcriptomic architecture of 2,000 breast tumours reveals novel subgroups. *Nature* 486: 346-352

Dede M, Hart T (2023) Recovering false negatives in CRISPR fitness screens with JLOE. *Nucleic Acids Res* 51: 1637-1651

Hong SW, Jiang Y, Kim S, Li CJ, Lee DK (2014) Target gene abundance contributes to the efficiency of siRNA-mediated gene silencing. *Nucleic Acid Ther* 24: 192-198

Lee MY, Moreno CS, Saavedra HI (2014) E2F activators signal and maintain centrosome amplification in breast cancer cells. *Mol Cell Biol* 34: 2581-2599

Morris EJ, Kawamura E, Gillespie JA, Balgi A, Kannan N, Muller WJ, Roberge M, Dedhar S (2017) Stat3 regulates centrosome clustering in cancer cells via Stathmin/PLK1. *Nat Commun* 8: 15289

Pancoska P, Moravek Z, Moll UM (2004) Efficient RNA interference depends on global context of the target sequence: quantitative analysis of silencing efficiency using Eulerian graph representation of siRNA. *Nucleic Acids Res* 32: 1469-1479

Patel N, Weekes D, Drosopoulos K, Gazinska P, Noel E, Rashid M, Mirza H, Quist J, Braso-Maristany F, Mathew S *et al* (2018) Integrated genomics and functional validation identifies malignant cell specific dependencies in triple negative breast cancer. *Nat Commun* 9: 1044

Paulmann C, Spallek R, Karpiuk O, Heider M, Schaffer I, Zecha J, Klaeger S, Walzik M, Ollinger R, Engleitner T *et al* (2022) The OTUD6B-LIN28B-MYC axis determines the proliferative state in multiple myeloma. *EMBO J* 41: e110871

Pereira B, Chin SF, Rueda OM, Vollan HK, Provenzano E, Bardwell HA, Pugh M, Jones L, Russell R, Sammut SJ *et al* (2016) The somatic mutation profiles of 2,433 breast cancers refines their genomic and transcriptomic landscapes. *Nat Commun* 7: 11479

Rhys AD, Godinho SA (2017) Dividing with Extra Centrosomes: A Double Edged Sword for Cancer Cells. *Adv Exp Med Biol* 1002: 47-67

Sabat-Pospiech D, Fabian-Kolpanowicz K, Prior IA, Coulson JM, Fielding AB (2019) Targeting centrosome amplification, an Achilles' heel of cancer. *Biochem Soc Trans* 47: 1209-1222

Sampson J, O'Regan L, Dyer MJS, Bayliss R, Fry AM (2017) Hsp72 and Nek6 Cooperate to Cluster Amplified Centrosomes in Cancer Cells. *Cancer Res* 77: 4785-4796

Schwanhausser B, Busse D, Li N, Dittmar G, Schuchhardt J, Wolf J, Chen W, Selbach M (2011) Global quantification of mammalian gene expression control. *Nature* 473: 337-342

Sobol A, Askonas C, Alani S, Weber MJ, Ananthanarayanan V, Osipo C, Bocchetta M (2017) Deubiquitinase OTUD6B Isoforms Are Important Regulators of Growth and Proliferation. *Mol Cancer Res* 15: 117-127

Turnbull AP, Ioannidis S, Krajewski WW, Pinto-Fernandez A, Heride C, Martin ACL, Tonkin LM, Townsend EC, Buker SM, Lancia DR *et al* (2017) Molecular basis of USP7 inhibition by selective small-molecule inhibitors. *Nature* 550: 481-486

Wu J, Mikule K, Wang W, Su N, Petteruti P, Gharahdaghi F, Code E, Zhu X, Jacques K, Lai Z *et al* (2013) Discovery and mechanistic study of a small molecule inhibitor for motor protein KIFC1. *ACS chemical biology* 8: 2201-2208

Xiao YX, Yang WX (2016) KIFC1: a promising chemotherapy target for cancer treatment? *Oncotarget* 7: 48656-48670

Yang M, Wei Y, He X, Xia C (2024) The deubiquitinating protein OTUD6B promotes lung adenocarcinoma progression by stabilizing RIPK1. *Biol Direct* 19: 46

Yukawa M, Yamauchi T, Kurisawa N, Ahmed S, Kimura KI, Toda T (2018) Fission yeast cells overproducing HSET/KIFC1 provides a useful tool for identification and evaluation of human kinesin-14 inhibitors. *Fungal Genet Biol* 116: 33-41

Dear Prof. Coulson,

Thank you for the submission of your revised manuscript to our editorial offices. I have now received the reports from the referees that I asked to re-evaluate the study, you will find below. As you will see, all referees now support publication of the study in EMBO reports. However, referees #1 and #3 have remaining concerns and suggestions to improve the study, I ask you to address in a final revised manuscript. Please also provide a final p-b-p-response addressing the remaining points.

- Please provide the abstract written in present tense throughout.
- Please remove the synopsis image, the synopsis blurb and the bullet points from the main manuscript text. Please upload these separately. Moreover, please provide the schematic summary figure as separate file with the exact width of 550 pixels and a height of not more than 400 pixels.
- We now use CRediT to specify the contributions of each author in the journal submission system. CRediT replaces the author contribution section. Please use the free text box to provide more detailed descriptions and do NOT provide your final manuscript text file with an author contributions section. See also our guide to authors: <https://www.embopress.org/page/journal/14693178/authorguide#authorshipguidelines>
- Please make sure that all the funding information is also entered into the online submission system and that it is complete and similar to the one in the acknowledgement section of the manuscript text file. Please remove the text from the Comments box in the submission system. Moreover, Wellcome Trust [109307/Z/15/Z] needs to be entered as a separate entry. Please check.
- Please add scale bars of similar style and thickness to all microscopic images (main, EV and Appendix figures), using clearly visible black or white bars (depending on the background). Please place these in the lower right corner of the images themselves. Please do not write on or near the bars in the image but define the size in the respective figure legend. Presently, some images miss scale bars (e.g. 7J or S8) and for some (e.g. 3D) the scale bars are too small. Please check.
- Please make sure that microscopic images are clearly separated by white lines or spaces. They should not touch each other, creating the impression that one larger image is shown (see e.g. Figure EV4A).
- Please make sure that the number "n" for how many independent experiments were performed, their nature (biological versus technical replicates), the bars and error bars (e.g. SEM, SD) and the test used to calculate p-values is indicated in the respective figure legends (main, EV and Appendix figures). Please also check that all the p-values are explained in the legend, and that these fit to those shown in the figure. Please provide statistical testing where applicable. Please avoid the phrase 'independent experiment', but clearly state if these were biological or technical replicates. Please also indicate (e.g. with n.s.) if testing was performed, but the differences are not significant. In case n=2, please show the data as separate datapoints without error bars and statistics. See also: <http://www.embopress.org/page/journal/14693178/authorguide#statisticalanalysis>

If $n < 5$, please show single datapoints for diagrams. It seems that presently in many diagrams the 'ns' is missing. Moreover:

- Please note that information related to n is missing in the legend of figure 6g.
- Is $n=2$ or $n=1$ in figure 1c? Why is there one error bar shown (control)? Please check.
- Please note that the error bars are not defined in the legends of figures 3b-c; 6a; 7d, f, h.
- Please note that the measure of center for the error bars needs to be defined in the legends of figures 7a-b; EV 5k.
- Please note that the legend for figure 7h is mislabeled as 7g for p-values in the manuscript. This needs to be rectified.
- Please note that the exact p values are not provided in the legends of figures 2c-d; 3b, e, h; 4c; 5e; 6g; 7a, h; EV 1f-g; EV 2i-j; EV 5a.
- Please note that in figure 2c; there is a mismatch between the annotated p values in the figure legend and the annotated p values in the figure file that should be corrected.
- Please note that the scale bar needs to be defined for figure EV 3e.
- Please note that scale bar and its definition are missing for figure 7j.
- Please note that the asterisk is not defined in the legend of figure 5b. This needs to be rectified.
- Please note that the white dotted structures are not defined in the legend of figure EV 4a. This needs to be rectified.
- Please add to each legend (main, EV figures, Appendix figures, where applicable) a 'Data Information' section explaining the statistics used or providing information regarding replicates and scales. See:

- Please note that the data citation (Data Refs: Breast Invasive Carcinoma, TCGA, PanCancer Atlas (2018), Data refs: Berger et

al., 2018; Mertins et al., 2016, Data Ref: CPTAC Breast Cancer (2024)) does not refer to deposited experimental data.

- Please note that the data citations Data refs: Berger et al., 2018; Mertins et al., 2016 are not tagged with the label "DATASET" in the reference list.

- Please note that the URLs for Data refs: Berger et al., 2018; Mertins et al., 2016 data citations are not provided.

- Please note that generic URLs are provided for Data Refs: Breast Invasive Carcinoma, TCGA, PanCancer Atlas (2018), Data Ref: CPTAC Breast Cancer (2024) data citations.

- Please make sure that the Appendix items are named AND called out as 'Appendix Table Sx' and 'Appendix Figure Sx'. Many callouts don't have the word "Appendix" in them (see e.g.: Fig S8C, Fig S8D, Fig S8E, Fig S7D, Fig S3).

- Please name the movies (source file names, titles and callouts) Movie EV1-EV6. Please remove their titles/legends from the Appendix file. Each legend then needs to be in a readme.txt file and ZIPped together with its movie and uploaded as one folder per movie.

- The Data Availability section (DAS) is restricted to datasets generated in this study and deposited. Thus please remove the sentence 'New materials and reagents are available on request from the corresponding authors' from the DAS.

- All materials and methods used need to be described in the main text using our 'Structured Methods' format, which is required for all research articles. According to this format, the Methods section should include a Reagents and Tools Table (listing key reagents, experimental models, software, and relevant equipment and including their sources and relevant identifiers), uploaded as separate file, and a Methods section (called 'Methods') in which we encourage the authors to describe their methods using a step-by-step protocol format with bullet points, to facilitate the adoption of the methodologies across labs. More information on how to adhere to this format as well as downloadable templates (.doc or .xls) for the Reagents and Tools Table can be found in our author guidelines (section 'Structured Methods'):

- Please remove the word files "Main figures with legends" and "Expanded view figures with legends" from the manuscript files. These are not needed.

Please use this link to submit your revision: <https://embor.msubmit.net/cgi-bin/main.plex>

Yours sincerely,

Referee #1:

The authors have addressed most of my major concerns. The points below could help the authors further strengthen their manuscript:

1- The effect of OTUD6B on KIFC1 expression beyond mitosis is still not clearly addressed. The IF images in Appendix Fig. S5A show that KIFC1 is strongly reduced in interphase cells upon OTUD6B depletion. The WB in Fig. 6A also shows that KIFC1 is downregulated upon OTUD6B depletion even in cells in S phase. Furthermore, while answering comment #7, the authors mentioned that OTUD6B depletion affects KIFC1 levels throughout the cell cycle by referring to Fig. 6A. Despite these, it was stated in answer to comment #4 that an interphase synchronization cannot be performed as KIFC1 is not present in G1 cells. These contradictions make it unclear if OTUD6B has a more general effect on KIFC1 throughout the cell cycle or has a mitosis-specific effect.

2- The reduction in KIFC1 mean intensity upon OTUD6B depletion in Supplementary Fig. S5B is much stronger compared to Fig. 3H. What is the reason for that difference?

3- Significance analysis for 3I is needed.

4- The newly added sgOTUD6B data shown in Fig. 7I and J seem incomplete. The downregulation of OTUD6B is not shown at protein level. Furthermore, a quantification is needed to comment on the effects of OTUD6B depletion on cell survival instead of showing cell pictures in Fig. 7J.

Referee #2:

The only remaining issue is that I was unable to properly view the uploaded movies provided as supplementary .mp4 files. While this may be due to a technical issue specific to my computer, I recommend that the editorial team double-check this with the authors to ensure the files are accessible before publication.

Aside from this, the authors have satisfactorily addressed all my previous comments, and I fully support the publication of this manuscript in EMBO Reports in its present form.

Referee #3:

The authors have included a significant amount of new data supporting the claim that OTUD6B deubiquitylates and stabilizes KIF1C, thereby facilitating centrosome clustering. They also clarified inconsistencies that were present in the original manuscript.

The authors spend a lot of energy justifying the interpretation of their results. Instead, they could have addressed the raised issues with a complementary approach. For example, establishing inducible OTUD6B knockout cell lines in BT549 to verify the siRNA-based phenotype. The inability to generate OTUD6B knockout cell lines does not conclusively indicate that OTUD6B is essential; it may also highlight technical challenges. Similarly, the authors were unable to generate OTUD6B knockout in the control hTERT-HME1 cell line due to the missing availability of puromycin as a selection marker, which could be easily circumvented using an alternative selection marker.

Below are specific points the authors should address before acceptance:

1. The observation that siRNA-mediated OTUD6B knockdown reduces cell viability in BT549 does not substantiate the claim that OTUD6B is essential across all TNBC cancers. Please tone down the claim: "Notably, OTUD6B is amplified and overexpressed in breast cancer and is required for the survival of TNBC cell lines with amplified centrosomes."
2. Although the authors show that OTUD6B deubiquitylates KIF1C, they provide no direct evidence that it counteracts APC/C. Moreover, the timing of deubiquitylation suggests that additional ubiquitin ligases might be involved. Consequently, references to APC/C should be removed from the synopsis figure, and the discussion should be moderated accordingly.
3. In Figure 3D, cells with degraded KIF1C (second and fourth rows) appear to have recently divided and entered G1, contrasting with the other cells shown. This would be a problem for the claim that WT-OTUD6B but not the C158C mutant stabilizes KIF1C since KIF1C is degraded in G1-phase in a cell cycle-specific manner. To ensure that transgene-specific effects, rather than cell cycle-specific effects, are accurately represented, the authors should select, show, and quantify cells in the same cell cycle phase.
4. Figure 3H: Representative images should be provided.
5. Please add error bars in Figures 2A, 2B, 3G and 3I.
6. In Figure 3C, indicate the statistical significance between WT and C158S constructs. Is there a significant difference?

EMBO Reports – response to reviewers

Dear Prof. Coulson,

Thank you for the submission of your revised manuscript to our editorial offices. I have now received the reports from the referees that I asked to re-evaluate the study, you will find below. As you will see, all referees now support publication of the study in EMBO reports. However, referees #1 and #3 have remaining concerns and suggestions to improve the study, I ask you to address in a final revised manuscript. Please also provide a final p-b-p-response addressing the remaining points.

Thank you for your feedback and the comments from the three reviewers, who we are pleased now support publication of the study in EMBO Reports.

Referee #1:

The authors have addressed most of my major concerns. The points below could help the authors further strengthen their manuscript:

Thank you for your feedback.

1- The effect of OTUD6B on KIFC1 expression beyond mitosis is still not clearly addressed. The IF images in Appendix Fig. S5A show that KIFC1 is strongly reduced in interphase cells upon OTUD6B depletion. The WB in Fig. 6A also shows that KIFC1 is downregulated upon OTUD6B depletion even in cells in S phase. Furthermore, while answering comment #7, the authors mentioned that OTUD6B depletion affects KIFC1 levels throughout the cell cycle by referring to Fig. 6A. Despite these, it was stated in answer to comment #4 that an interphase synchronization cannot be performed as KIFC1 is not present in G1 cells. These contradictions make it unclear if OTUD6B has a more general effect on KIFC1 throughout the cell cycle or has a mitosis-specific effect.

We believe dynamic activity of OTUD6B towards KIFC1 occurs predominantly during mitosis, when KIFC1 is acutely degraded through addition of K11-linked ubiquitin. This is supported by data showing enriched mitotic co-localisation of OTUD6B and KIFC1 (Fig 4B-4E), and OTUD6B depletion increasing K11 ubiquitination of KIFC1 (Fig 6D) and decreasing mitotic KIFC1-Venus stability (Fig 6F-6G). In point 4 of the original review, you specifically asked for ubiquitination experiments to be repeated in **G1-synchronised cells**. We responded that we are not able to perform the ubiquitination experiment with **cells synchronised in G1**, as KIFC1 is not present as cells exit mitosis and enter G1 and that levels remain too low for this type of analysis during G1. We stand by this point. In Fig EV5I-5J we had already shown OTUD6B depletion did not significantly affect KIFC1 stability or ubiquitination in asynchronous interphase cells (a mixture of G1, S and G2). This supports our conclusion that the dynamic activity of OTUD6B towards ubiquitylated KIFC1 is most marked during mitosis.

Although predominant in mitosis, we do not claim that the effect is exclusive to mitosis. Being transparent about our data, we see reduced levels of KIFC1 by immunoblotting in cells where OTUD6B is depleted prior to analysis of an asynchronous population, or synchronisation and analysis in S-phase as well as throughout mitosis (Fig 6A). This is consistent with data in Fig S5. The image in old Fig S5A was an example of individual interphase cells, so to address your concerns we have now replaced this with a full field of view image (new Fig S5A) that better summarises the data to reflect variability within the cell population (please also see response to point 2 below regarding quantification of this experiment in Fig S5B).

The effect of OTUD6B on KIFC1 levels in interphase cells may reflect antagonism of an alternative E3, or of premature APC/C activity in cancer cells with defective SAC, or undergoing DNA-repair in G2, the latter potentially a legacy of OTUD6B-deficient division during the previous cell cycle. These ideas were already discussed in the manuscript and have now been further clarified (page 16). Irrespective of interphase cell

activity, the effect of OTUD6B at mitosis is clear, and has physiologically relevant consequences during division of cancer cells with centrosome amplification.

2- The reduction in KIFC1 mean intensity upon OTUD6B depletion in Supplementary Fig. S5B is much stronger compared to Fig. 3H. What is the reason for that difference?

Thank you for highlighting this. On rechecking, we unfortunately found an error in the log2 conversion for some data in Fig S5B. We apologise for this, and it has now been corrected in the revised figure. We also noted that the comparison of data between these figures was hampered by the different styles and y-axes, which have now been harmonised. Although the experimental configurations differ, as Fig 3H monitors stably expressed GFP-KIFC1 and Fig S5B endogenous KIFC1, in both cases KIFC1 is significantly reduced by OTUD6B depletion, consistent with a post-translational role in regulating OTUD6B stability.

3- Significance analysis for 3I is needed.

As stated in the legend, old Fig 3I (new Fig 3J) shows the distribution of cells from one biological repeat, with $n = \text{approx. } 40$ cells in each condition, hence we had not included statistical analysis. For information, when we performed statistical analysis as you request (Chi-squared test as used in similar figures) this confirmed a significant increase in multipolar spindles on OTUD6B depletion ($P=0.0053$). Rescue of this phenotype by KIFC1 induction approaches significance ($P = 0.09$). Unfortunately, we do not currently have resource for additional biological replicates and would interpret these p-values with caution given the relatively low cell numbers, so have not included them in the manuscript.

4- The newly added sgOTUD6B data shown in Fig. 7I and J seem incomplete. The downregulation of OTUD6B is not shown at protein level. Furthermore, a quantification is needed to comment on the effects of OTUD6B depletion on cell survival instead of showing cell pictures in Fig. 7J.

As we explained in the previous point-by-point letter, the BT549 did not tolerate CRISPR editing of OTUD6B and so we were unable to grow out sufficient cells for these types of analyses. We have now made this clearer on page 13. We managed to get genetic data by direct lysis of the very few remaining cells for PCR, but we could not analyse protein levels. We were also unable to quantify growth using the Incucyte, as so few cells were present, and substantial cell debris precluded the use of a mask to determine cell confluence.

Referee #2:

The only remaining issue is that I was unable to properly view the uploaded movies provided as supplementary .mp4 files. While this may be due to a technical issue specific to my computer, I recommend that the editorial team double-check this with the authors to ensure the files are accessible before publication.

The editorial assistant at EMBO Reports has confirmed that they were able to open and play the movies.

Aside from this, the authors have satisfactorily addressed all my previous comments, and I fully support the publication of this manuscript in EMBO Reports in its present form.

Thank you for your feedback and support.

Referee #3:

The authors have included a significant amount of new data supporting the claim that OTUD6B deubiquitylates and stabilizes KIF1C, thereby facilitating centrosome clustering. They also clarified inconsistencies that were present in the original manuscript.

Thank you for your positive feedback on the new data and clarifications.

The authors spend a lot of energy justifying the interpretation of their results. Instead, they could have addressed the raised issues with a complementary approach. For example, establishing inducible OTUD6B knockout cell lines in BT549 to verify the siRNA-based phenotype. The inability to generate OTUD6B knockout cell lines does not conclusively indicate that OTUD6B is essential; it may also highlight technical challenges. Similarly, the authors were unable to generate OTUD6B knockout in the control hTERT-HME1 cell line due to the missing availability of puromycin as a selection marker, which could be easily circumvented using an alternative selection marker.

We were surprised by these comments and, although we are not being asked to address these points, we will respond briefly. We do not have unlimited resources and had already used complementary or alternative techniques to address the experimental questions raised, thus establishing inducible OTUD6B knockout cell lines was not essential for the study. The inability to generate knockouts on specific cellular backgrounds is the fundamental basis of the DepMap essentiality screen, that was relied upon by this reviewer when initially critiquing our study. We would like to re-iterate that we did not have technical challenges in generating OTUD6B knockouts. BT549 cells did not tolerate OTUD6B gene-editing, demonstrating phenotypes consistent with division without clustering amplified centrosomes (movies EV4-EV6). Alongside this, we clearly showed that the identical system and protocol used in BT549, generated OTUD6B knockout cell lines in HEK293T that lack centrosome amplification (Appendix Fig S8C-S8E). Direct comparison was important, hence the need to use the same selection marker in an alternative cell line.

Below are specific points the authors should address before acceptance:

1. The observation that siRNA-mediated OTUD6B knockdown reduces cell viability in BT549 does not substantiate the claim that OTUD6B is essential across all TNBC cancers. Please tone down the claim: "Notably, OTUD6B is amplified and overexpressed in breast cancer and is required for the survival of TNBC cell lines with amplified centrosomes."

Thank you for highlighting this sentence, which we now realise could potentially be mis-interpreted as overstating our claims. We have modified this to read "Notably, OTUD6B is amplified and overexpressed in breast cancer, and is required for survival of TNBC cell lines with either high (BT549) or moderate (MDA-MB-231) levels of amplified centrosomes."

2. Although the authors show that OTUD6B deubiquitylates KIF1C, they provide no direct evidence that it counteracts APC/C. Moreover, the timing of deubiquitylation suggests that additional ubiquitin ligases might be involved. Consequently, references to APC/C should be removed from the synopsis figure, and the discussion should be moderated accordingly.

We provide several lines of experimental evidence that, taken together, strongly support the notion that activity of OTUD6B can counteract the addition of ubiquitin chains to KIFC1 by the APC/C, which is well established as the predominant E3 ligase that adds K11 chains to substrates during mitosis (page 11/12).

- (i) OTUD6B depletion has a very profound effect on the decoration of KIFC1 with K11-only chains during mitotic exit (Fig 6E), which is more marked than its effect towards ubiquitin chains with no linkage restriction (Fig 6B). This would only be achieved by opposing an E3 with a strong mitotic preference for generating K11 linkages.

- (ii) Linked to this, endogenous OTUD6B was co-immunoprecipitated with KIFC1 from mitotic cells when K11-only, but not K63-only ubiquitin was over-expressed (Fig 6D), indicating it interacts with KIFC1 decorated with ubiquitin chains containing K11 linkages.

- (iii) OTUD6B depletion affects the stability of KIFC1-Venus during mitosis. Importantly however, it does not affect the stability of KIFC1-Venus-Dmut, which has a mutation in the D-box that is required to identify it as

substrate for the APC/C (Fig 6F-6G). Thus, OTUD6B opposes E3 activity that relies on the D-box to target KIFC1 for mitotic ubiquitin-mediated degradation by the APC/C.

(iv) OTUD6B depletion affects both KIFC1 ubiquitination (Fig 6B) and the stability of KIFC1-Venus (Fig 6F-6G) over a timeframe encompassing both the activity of the APC/C towards early substrates (like cyclin A), and classical substrates degraded during anaphase once the SAC is satisfied.

As the synopsis figure accurately reflects these data, we have not modified it. We had already thoroughly addressed in the previous manuscript versions, the points you raise again about timing of APC/C activity, which is well known to differ for different substrates (pages 12, 15, 16), as well as the potential for antagonism of other E3s by OTUD6B (page 16) but have endeavoured to make these points even clearer.

3. In Figure 3D, cells with degraded KIF1C (second and fourth rows) appear to have recently divided and entered G1, contrasting with the other cells shown. This would be a problem for the claim that WT-OTUD6B but not the C158C mutant stabilizes KIFC1 since KIFC1 is degraded in G1-phase in a cell cycle-specific manner. To ensure that transgene-specific effects, rather than cell cycle-specific effects, are accurately represented, the authors should select, show, and quantify cells in the same cell cycle phase.

We took a completely unbiased approach to this experiment, as we had already shown by flow cytometry that OTUD6B depletion does not alter cell cycle phase distribution (Fig EV5K). Rather than scoring KIFC1 in cells according to specific cell cycle phases (as you suggest), we scored all cells that were visible in multiple, random fields of view across all slides (Fig 3E). The populations of >250 cells per condition from 3 biological replicates provided an overview of KIFC1 expression across all cell cycle phases. In control cells the population had a bimodal distribution, most likely reflecting the cells in G1 in the lower KIFC1 expressing group. Importantly, there was a highly significant shift of the mean expression level with OTUD6B depletion, which was rescued by WT but not CS OTUD6B. At a population level, it was evident that this was driven by a reduction in the level of KIFC1 in the higher expressing cells (i.e. not G1 cells). Thus, we strongly believe that the data we presented accurately reflect the effects of OTUD6B depletion, rather than any change or effect of cell cycle. Thus, the experiment you proposed is not essential and would not bring anything additional to the manuscript. However, in response to your comment we have now included alternative representative images in Fig 3D, of cells which do not appear to have recently divided.

4. Figure 3H: Representative images should be provided.

Thank you for this suggestion, representative images for 3H have been included as new Fig 3I.

5. Please add error bars in Figures 2A, 2B, 3G and 3I.

We do not believe it would be correct to add error bars to these figures. They represent populations of mitotic cells scored in an unbiased fashion across biologically replicated experiments in unsynchronised cells. By their nature the proportion of mitotic cells is low and therefore equal numbers for each experimental condition could not be scored in each biological replicate. In line with the unequal group sizes and relatively small n numbers, we have used non-parametric tests in these figure panels (Chi-square in 2A, 2B, 3G and 3I). As these are distribution-free tests, standard deviation error bars are not applicable.

6. In Figure 3C, indicate the statistical significance between WT and C158S constructs. Is there a significant difference?

We have added the statistical comparison across all samples in Fig 3C. There is not a significant difference between WT and CS in the bulk western blotting analysis, as this represents a heterogeneous population of cells, some with and some without overexpression of dTOMATO-OTUD6B, as described on page 8. As explained in the previous P-B-P letter, the immunoblots are intended to show the depletion of OTUD6B and its effect on KIFC1, whereas single cell quantification of KIFC1 level (Fig 3E) was performed to assess rescue.

Prof. Judy Coulson
University of Liverpool
Molecular and Clinical Cancer Medicine
Institute of Systems Molecular & Integrative Biology
University of Liverpool
Liverpool L69 3BX
United Kingdom

Dear Prof. Coulson,

Thank you for the submission of your further revised manuscript to our editorial offices. I now went through this and your further p-b-p-response and consider the remaining referee points as adequately addressed.

I thus am very pleased to accept your manuscript for publication in the next available issue of EMBO reports. Thank you for your contribution to our journal.

Yours sincerely,
